# Isoform-level transcriptome-wide association uncovers genetic risk mechanisms for neuropsychiatric disorders in the human brain

**Arjun Bhattacharya**[1,2,3] ✉, **Daniel D. Vo** [4,5], **Connor Jops**[4,5], **Minsoo Kim**[6,7], **Cindy Wen**[6,7,8], **Jonatan L. Hervoso**[8], **Bogdan Pasaniuc** [3,8,9,11] & **Michael J. Gandal** [4,5,6,7,10,11] ✉

Methods integrating genetics with transcriptomic reference panels prioritize risk genes and mechanisms at only a fraction of trait-associated genetic loci, due in part to an overreliance on total gene expression as a molecular outcome measure. This challenge is particularly relevant for the brain, in which extensive splicing generates multiple distinct transcript-isoforms per gene. Due to complex correlation structures, isoform-level modeling from *cis*-window variants requires methodological innovation. Here we introduce isoTWAS, a multivariate, stepwise framework integrating genetics, isoform-level expression and phenotypic associations. Compared to gene-level methods, isoTWAS improves both isoform and gene expression prediction, yielding more testable genes, and increased power for discovery of trait associations within genome-wide association study loci across 15 neuropsychiatric traits. We illustrate multiple isoTWAS associations undetectable at the gene-level, prioritizing isoforms of *AKT3*, *CUL3* and *HSPD1* in schizophrenia and *PCLO* with multiple disorders. Results highlight the importance of incorporating isoform-level resolution within integrative approaches to increase discovery of trait associations, especially for brain-relevant traits.

Recently, the number of genetic associations with complex traits identified by genome-wide association studies (GWAS) has increased considerably[1,2]. However, translating these associations into concrete molecular mechanisms remains a great obstacle for the field. As GWAS hits predominantly localize within non-coding regions, often within large blocks of linkage disequilibrium (LD), a major challenge is prioritizing the underlying causal variant(s) and identifying their putative functional impact on nearby target genes. Numerous methods,

[1]Department of Epidemiology, University of Texas MD Anderson Cancer Center, Houston, TX, USA. [2]Institute for Data Science in Oncology, University of Texas MD Anderson Cancer Center, Houston, TX, USA. [3]Department of Pathology and Laboratory Medicine, David Geffen School of Medicine, University of California, Los Angeles, CA, USA. [4]Department of Psychiatry, Perelman School of Medicine, University of Pennsylvania, Philadelphia, PA, USA. [5]Lifespan Brain Institute at Penn Med and the Children's Hospital of Philadelphia, Philadelphia, PA, USA. [6]Department of Psychiatry and Biobehavioral Sciences, Semel Institute, David Geffen School of Medicine, University of California, Los Angeles, CA, USA. [7]Department of Human Genetics, David Geffen School of Medicine, University of California, Los Angeles, CA, USA. [8]Bioinformatics Interdepartmental Program, University of California, Los Angeles, CA, USA. [9]Department of Computational Medicine, David Geffen School of Medicine, University of California, Los Angeles, CA, USA. [10]Department of Genetics, Perelman School of Medicine, University of Pennsylvania, Philadelphia, PA, USA. [11]These authors contributed equally: Bogdan Pasaniuc, Michael J. Gandal. ✉e-mail: abhattacharya3@mdanderson.org; michael.gandal@pennmedicine.upenn.edu

including transcriptome-wide association studies (TWAS), have been developed to integrate population-level transcriptomic reference panels with GWAS summary statistics to prioritize genes at trait-associated loci[3–15]. TWASs impute the *cis*-component of gene expression predicted by common variants into an association cohort, thereby reducing multiple comparisons and increasing interpretability by identifying a set of genes that may underlie the genetic association[3,4].

Previous integrative analyses have largely focused on total gene expression as the molecular outcome, and not the distinct transcript isoforms of a gene generated through alternative splicing, a tissue-specific gene regulatory mechanism present in ~90% of human genes that vastly expands the genome's coding and regulatory potential[16–19]. Compared with other tissues, brain-expressed genes are longer, contain more exons, and exhibit the most complex splicing pattern, contributing to the evolutionary and phenotypic complexity of the human brain[20–23]. While Gencode v40 annotates 4.0±7.28 isoforms per gene (mean ± standard deviation), specific neuronal genes are individually known to have >1000 unique isoforms[24,25]. Independent of gene expression, splicing dysregulation has been implicated in disease[20–22,26–28], especially for neuropsychiatric disorders[10,20,22,29]. Local splicing events can be difficult to measure and integrate across multiple large-scale datasets. Splicing is often coordinated across a gene, yielding many non-independent features that increases multiple testing burden. In contrast, transcript-isoform abundance can be rapidly estimated across large-scale RNA-sequencing (RNA-seq) datasets using pseudoalignment methods[30,31]. Furthermore, in the brain, isoform-level expression changes have shown greater enrichment for schizophrenia (SCZ) heritability than gene or local splicing changes[20,29,32–34]. However, to fully integrate transcript-isoform quantifications with GWASs, innovative computational methods are needed that jointly model the highly correlated isoforms of the same gene.

Here, we present isoform-level TWAS (isoTWAS), a flexible approach for complex trait mapping by integrating genetic effects on isoform-level expression with GWAS. Using simulations and data from the Genotype-Tissue Expression (GTEx) Project[35] and the PsychENCODE Consortium[20,22], we show that isoTWAS provides several advantages compared with gene-level methods. First, for transcriptomic prediction, the correlation between isoforms provides additional information unavailable when only gene-level expression is modeled. This leads to improved prediction accuracy[36] of >80% of individual isoforms, with a median of ~1.8- to 2.4-fold improvement, and of total gene expression by 25–70%. Consequently, this doubles the number of testable features in the trait mapping step. Third, divergent patterns of genetic effects across isoforms can be leveraged to provide a more granular hypothesis for a mechanism underlying the single-nucleotide polymorphism (SNP)–trait relationship. Finally, the isoTWAS framework jointly captures expression and splicing disease mechanisms while maintaining a well-controlled false discovery rate. Using GWAS data for 15 neuropsychiatric traits, isoTWAS greatly increases discovery of gene-level trait associations, uncovering associations at ~60% more GWAS loci compared to traditional gene-level TWAS. These results stress the need to shift focus to transcript isoforms to increase discovery of transcriptomic mechanisms underlying genetic associations with complex traits.

## Results

### The isoTWAS framework

isoTWAS prioritizes genes with transcript isoforms whose *cis*-genetic component of expression is significantly associated with a complex trait. We first jointly model the expression of distinct isoforms of a gene as a matrix while accounting for their pairwise correlation structure[3,4,24,35]. Here, we assume that (1) local genetic variants directly modulate expression of an isoform and (2) the abundance of a gene is the sum of the abundance of its isoforms, computed as transcripts per million (TPM) (Extended Data Fig. 1a)[30,31,37,38]. Integrating isoform-level

expression into trait mapping may prioritize discoveries in disease mapping missed by gene-level integration, as in a setting where a gene has multiple isoforms but only one is associated with the trait (Fig. 1a). By modeling the genetic architectures of isoforms of a gene simultaneously, isoTWAS provides a deeper understanding of potential transcriptomic mechanisms that underlie genetic associations.

The isoTWAS framework contains three steps (Fig. 1b). First, we build multivariate predictive models of isoform-level expression from all SNPs within 1 Mb in well-powered functional genomics training datasets (for example, GTEx[35] and PsychENCODE[20,22]) using one of four multivariate penalized predictive frameworks[39–42]. As a baseline for comparison, we modeled each individual isoform independently with univariate regularized regressions[4,41,43,44] (Methods). Model performance was assessed via 5-fold cross-validation (CV).

Second, we use these models to impute isoform expression into an external GWAS cohort and quantify the association with the target GWAS phenotype. If individual-level genotypes are available, isoform expression can be directly imputed as a linear combination of the SNPs in the models, and these associations can be estimated through appropriate regression analyses. If only GWAS summary statistics are available, imputation and association testing is conducted simultaneously through a weighted burden test[4].

Third, isoTWAS performs stepwise hypothesis-testing procedure to account for multiple comparisons and control for local LD structure. Isoform-level *P* values are first aggregated to the gene-level using the aggregated Cauchy association test (ACAT)[45], where false discovery rates are controlled, and then individual isoforms of prioritized genes are subjected to post-hoc family-wise error control[46] (Extended Data Fig. 1b and Methods). After this step, a set of isoforms are identified whose *cis*-genetic components of expression are associated with the trait of interest[4]. For these isoforms, we apply a rigorous permutation test by permuting the SNP-to-isoform effects to generate a null distribution. This permutation test assesses how much signal is added by isoform expression, given the GWAS architecture of the locus, and controls for large LD blocks[4]. Lastly, we can perform isoform-level Bayesian fine mapping at loci with significant trait associations to identify the minimal credible set of isoforms that contains the 'causal' isoform and to assign individual posterior inclusion probabilities (Methods). isoTWAS is available as an R package[47].

### Improved isoform and gene expression prediction

Previous work demonstrates that isoform-level quantifications from short-read RNA-seq, when propagated to the gene-level, can lead to more accurate gene expression estimates and differential expression inference[37,38]. We therefore hypothesized that our multivariate SNP-based imputation of isoform expression, when aggregated to the gene level, would outperform traditional gene-level (for example, TWAS) models. To evaluate total gene expression predictions of TWAS and isoTWAS models across multiple genetic architectures, we conducted simulations across 22 different gene loci using European-ancestry reference data[48]. At each gene locus, we controlled expression heritability and simulated 2–10 distinct isoforms, varying the proportion of causal isoform-level quantitative trait loci (isoQTLs; $p_{causal}$) and their sharing between isoforms ($p_{shared}$) (Methods and Fig. 2a).

For isoTWAS, multivariate elastic net[41] demonstrated the greatest CV prediction of isoform expression across most simulation settings (Fig. 2b, Extended Data Fig. 2a and Supplementary Data 1). For total gene expression prediction, the optimal isoTWAS models in sum outperformed the optimal TWAS model, particularly at sparser isoQTL architectures, with median absolute increase in adjusted $R^2$ of 0.6–3.5% (Fig. 2c, Extended Data Fig. 2b and Supplementary Data 2). Performance gains decreased with denser isoQTL architectures, although we expect approximately 0.1–1% quantitative trait locus (QTL) sparsity (that is, 1–10 causal expression, or e-, and isoQTLs per gene or isoform)[35]. In simulations, isoTWAS prediction of gene expression also increases

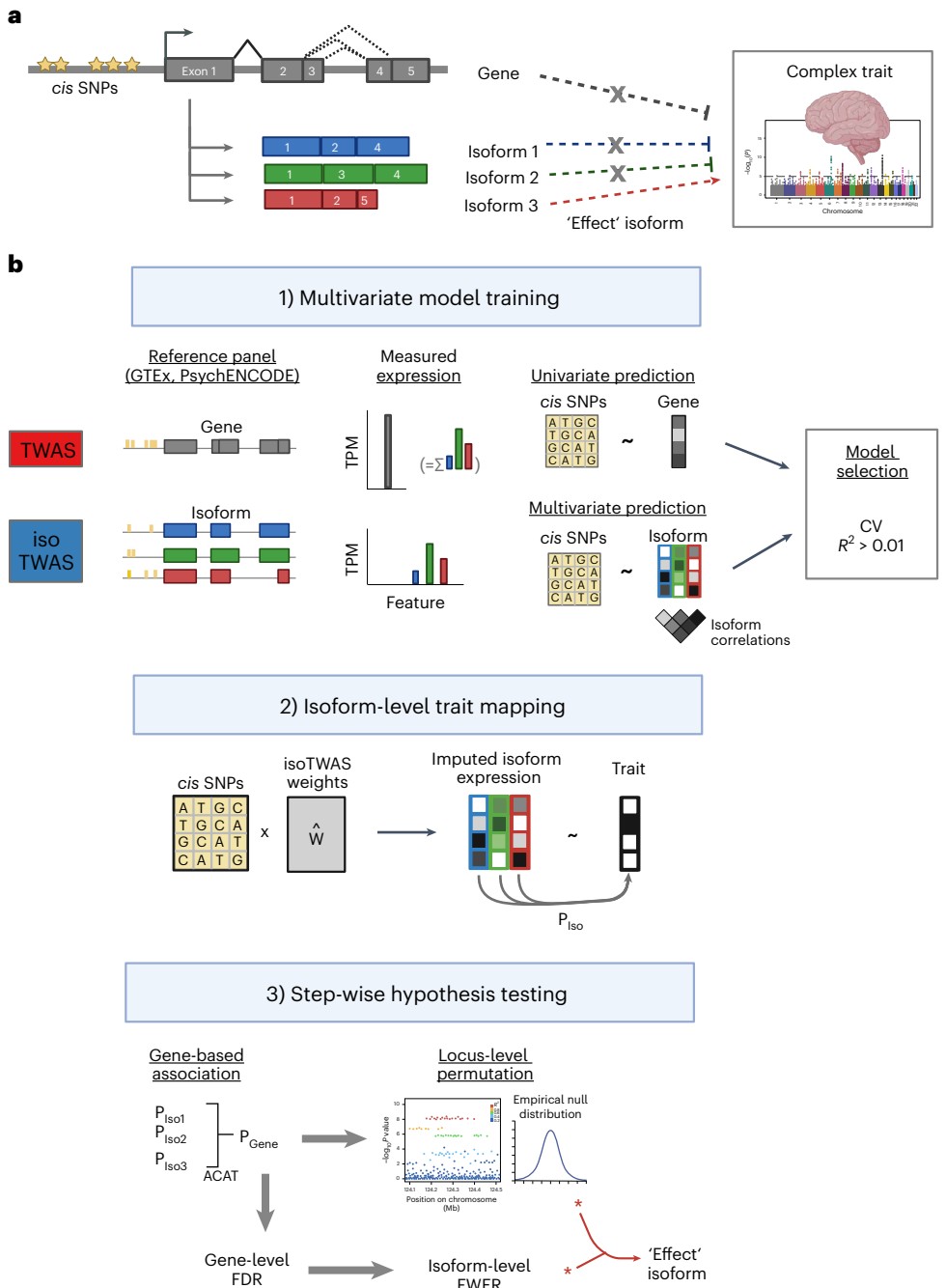

**Fig. 1 | Isoform-centric approach for complex trait mapping and prioritization of disease mechanisms at a genetic locus. a**, Motivation for isoTWAS. Gene G has three isoforms but only one has an effect on the trait. Gene G itself does not show an association with the trait. Studying genetic associations with an isoform-centric perspective will prioritize gene G, but not with a gene-centric perspective. **b**, Schematic comparison of isoTWAS and TWAS. First, using functional genomics reference panels, isoTWAS trains a multivariate model to predict isoform expression from *cis*-window SNPs, compared to a univariate model of total gene expression in TWAS. Second, predictive models with CV $R^2 > 0.01$ are then imputed into an association cohort to generate a nominal *P* value. Third, isoTWAS maps isoform–trait associations through a stepwise hypothesis-testing framework that provides gene-level false discovery rate (FDR) control and isoform-level family-wide error rate (FWER) control. Finally, locus-level permutation testing is performed to control for GWAS architecture and LD structure at the locus. An optional Bayesian fine-mapping step can be additionally applied at loci with multiple associations.

as the proportion of shared non-zero effect SNPs across isoforms decreases (Fig. 2b,c, Extended Data Fig. 2b and Supplementary Data 2).

Next, we assessed predictive performance in GTEx data from 48 tissues (13 brain) with sufficient sample sizes ($N > 100$) for all genes with multiple expressed isoforms (Supplementary Table 1 and Methods). Altogether, we built predictive models for 50,000 to 80,000 isoforms across 8,000 to 12,000 unique genes per tissue that met CV cutoffs (Methods, Extended Data Figs. 3–5 and Supplementary Table 2).

We considered three criteria to evaluate the prediction of both the multivariate and isoform-centric approaches of isoTWAS: (1) the number of isoforms imputed using multivariate/univariate models with CV $R^2 > 0.01$, (2) the number of unique genes with >1 isoform imputed at CV $R^2 > 0.01$ and (3) the number of unique genes with total gene expression imputed at CV $R^2 > 0.01$ using isoTWAS (summed) or TWAS models. At the isoform level (criterion 1), through multivariate modeling, we trained 2.3- to 2.5-fold more models at CV $R^2 > 0.01$ across

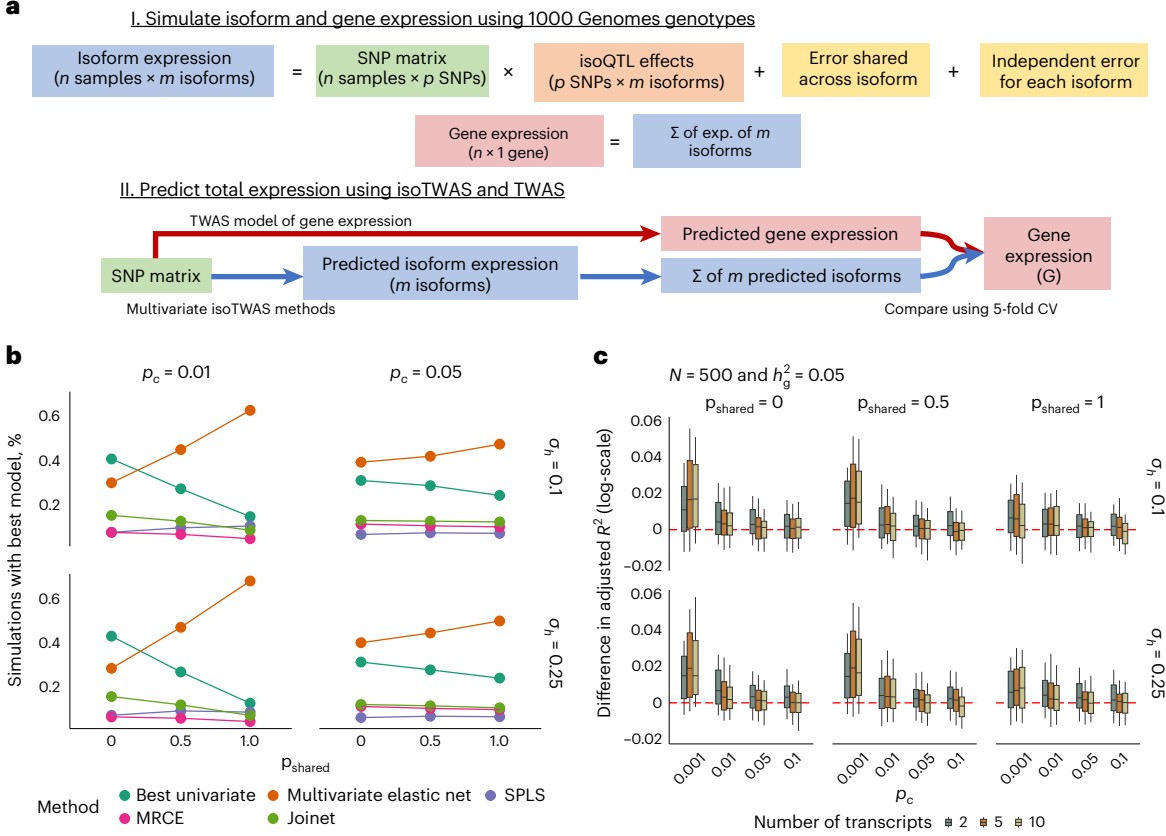

**Fig. 2 | IsoTWAS models predict gene expression with more accuracy than TWAS models in simulated data. a**, Simulation setup to generate isoform expression with specified isoQTL architecture, controlled expression heritability, number of isoforms and inter-isoform correlation structure. **b**, Proportion of simulations where the isoTWAS model exhibited the maximal adjusted $R^2$ for marginal isoform prediction ($y$ axis) across varying proportions of shared isoQTLs between isoforms ($x$ axis; $p_{shared}$), proportions of causal isoQTLs ($p_c$; top margin) and isoform expression variance attributed to shared non-*cis*-genetic effects ($\sigma_h$; right margin).

The multivariate elastic net was the best performing model across most simulated architectures. **c**, Boxplots show the difference in adjusted $R^2$ in predicting total gene expression between isoTWAS and TWAS models from simulations with sample size 500 where isoform and gene expression heritability are set to 0.05, across varying causal isoQTL proportions ($x$ axis), number of transcripts per gene, proportion of shared isoQTLs (top margin) and proportion of variance explained by shared non-*cis*-genetic effects (right margin). All boxplots represent the median, 25% and 75% quantiles, and whiskers correspond to the 10% and 90% quantiles.

the 48 tissues, compared to univariate approaches (Fig. 3a). isoTWAS improved prediction for 79–82% of isoforms with a median increase of ~1.8- to 2.4-fold increase in adjusted $R^2$ (Extended Data Fig. 3a,b and Supplementary Table 2). Concordant with simulations, multivariate elastic net outperformed other methods, indicating that leveraging the shared genetic architecture between isoforms aids in marginal prediction of each isoform (Extended Data Fig. 3c and Supplementary Table 2). Additionally, multivariate models were particularly powerful in brain tissues compared to other tissues in GTEx, showing significantly improved performance compared to univariate models (Fig. 3b; $P = 0.011$ from ordinary least squares regression of median percent increase in CV $R^2$ across tissue, adjusted for sample size). This suggests more shared isoQTL architecture in brain tissues than others, which isoTWAS leverages for improved prediction. These gains in prediction accuracy translate into increased power in trait association[49].

At the gene level (criteria 2 and 3), isoTWAS increased the number of genes with testable models in the trait mapping step and improved prediction of total gene expression. The number of unique genes with >1 isoTWAS model at CV $R^2 > 0.01$ (inclusion criterion for isoTWAS trait mapping) was 1.9–2.5 times larger than the number of unique genes with TWAS models achieving CV $R^2 > 0.01$ for gene expression prediction (Fig. 3c, Extended Data Fig. 4a and Supplementary Table 2). For a given gene, isoTWAS models (summed) outperformed TWAS models in prediction of total gene expression by a median of 25–70% in CV (Extended Data Fig. 4b) with a 50–80% increase in the number

of genes that are predicted at CV $R^2 > 0.01$ (Fig. 3d and Extended Data Fig. 5). We replicated these gains in total gene expression prediction using an independent, out-of-sample QTL dataset of adult cortex from PsychENCODE/AMP-AD (Methods). Multivariate isoTWAS models outperformed univariate TWAS models in predicting total gene expression, with a 15.2% median percent increase in adjusted $R^2$ when training in GTEx and testing in PsychENCODE/AMP-AD and 23.9% vice versa (Fig. 3e and Supplementary Table 3).

As genes differ in the number and expression patterns of their constituent isoforms, gene length, SNP density, quantification accuracy, and other relevant factors, we characterized their impact on isoTWAS performance (Methods, Supplementary Note, Extended Data Fig. 6 and Supplementary Data 3 and 4). We also evaluated the impact of reference transcriptome annotation fidelity by generating a synthetic dataset quantified using a reference annotation masking the dominant isoforms for a set of genes (Extended Data Fig. 3d). We discuss these evaluations in detail in Supplementary Note.

In total, as predictive performance is positively related to power to detect trait associations[49], both the increased number and accuracy of trainable imputation models using isoTWAS have strong implications for increased discovery[49].

## Calibrated null and improved power across architectures

We next introduced GWAS data for complex traits into our simulation framework to benchmark the false positive rate (FPR) and power of

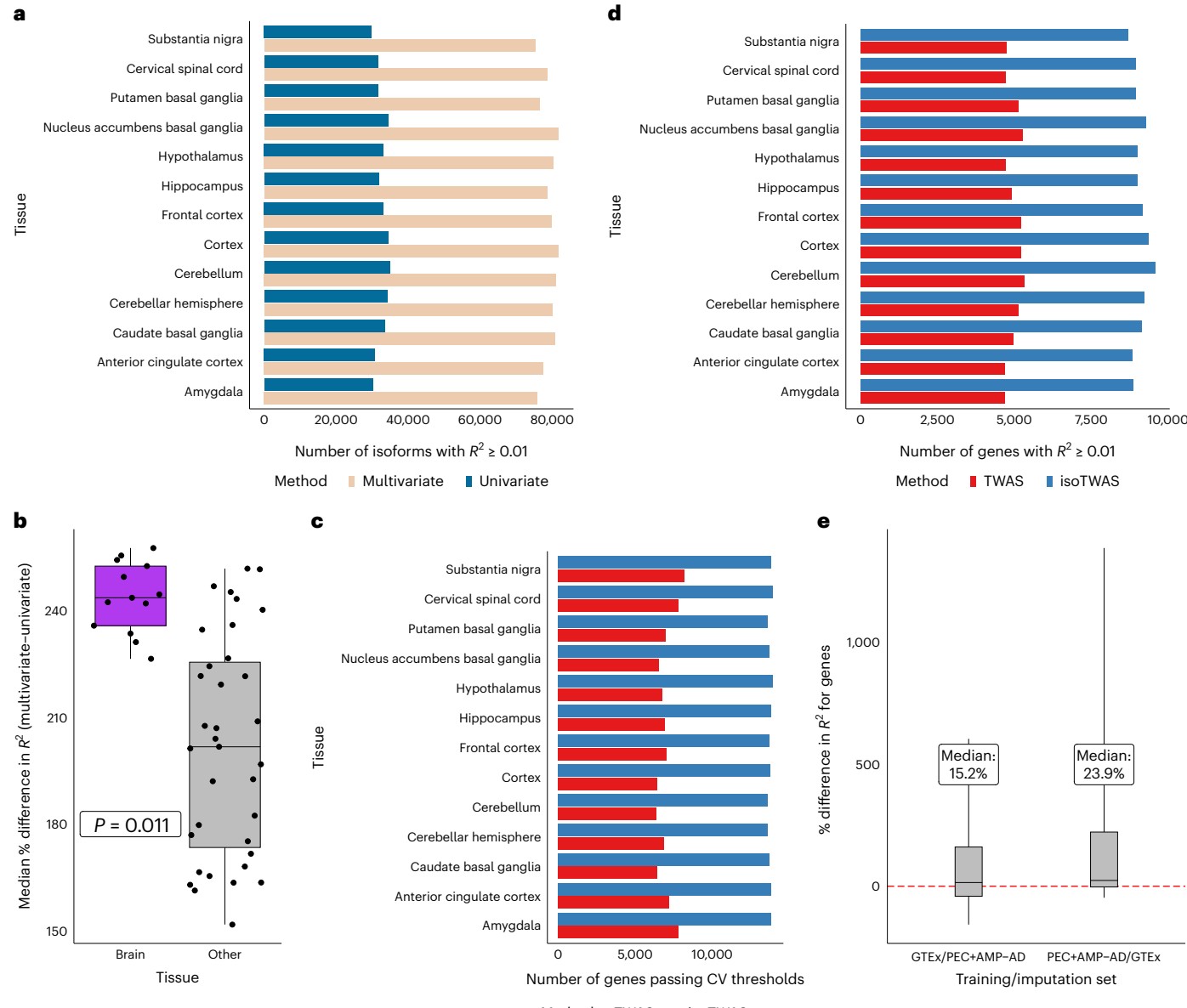

**Fig. 3 | Multivariate isoform-level models overperform gene-level models in predicting total gene expression. a**, Barplot showing the number of isoforms with CV $R^2 > 0.01$ ($y$ axis) using multivariate (cream) and univariate (blue) modeling methods across brain tissues ($x$ axis). **b**, Boxplot of median percent difference in predicting isoform expression ($y$ axis) using multivariate compared to univariate method by tissue type ($x$ axis), colored by brain (purple) or non-brain (gray) tissue. $P$ value derived from two-sided Wald-type $t$-test, adjusted for sample size ($n = 13/35$ brain/other tissues, respectively). All boxplots represent the median, 25% and 75% quantiles, and whiskers correspond to minimum and

maximum. **c**, Barplot showing the number of genes passing CV thresholds ($y$ axis) using TWAS (red) and isoTWAS (blue) across brain tissues. **d**, Barplot showing the number of genes with CV $R^2 > 0.01$ ($y$ axis) using TWAS (red) and isoTWAS (blue) across brain tissues. **e**, Boxplot of percent difference in $R^2$ ($y$ axis) for out-of-sample prediction of total gene expression (isoTWAS−TWAS) using external datasets. $x$ axis shows the training and imputation datasets. The median percent difference is labeled ($n = 255$ GTEx, 2115 PEC + AMP/AD). All boxplots represent the median, 25% and 75% quantiles, and whiskers correspond to the 10% and 90% quantiles.

isoTWAS (Methods). First, the FPR is controlled at 0.05 for isoform-level mapping using ACAT (Extended Data Fig. 7a and Supplementary Data 5). For a simulated trait, we modeled causal effect architectures for a genomic locus with 2–10 isoforms under three scenarios (Methods, Fig. 4 and Extended Data Fig. 7b): (1) where the true trait effect is from only total gene expression, (2) where there is only one 'effect isoform' with a non-zero effect on the trait and (3) where there are two effect isoforms with varying magnitudes of association. Scenario 1 showed clear increases in power for TWAS over isoTWAS, but this advantage decreased with increased causal proportion of isoQTLs and proportion of shared isoQTLs (Fig. 4a and Supplementary Data 6). For scenarios

2 and 3, as effects on the trait varied across isoforms of the same gene (Fig. 4b,c and Supplementary Data 7 and 8), isoTWAS showed clear increases in power over TWAS across most scenarios and causal effect architectures and particularly in settings with one effect isoform or two divergent effect isoforms. However, when the effect sizes of two effect isoforms converged, TWAS and isoTWAS demonstrated similar power (Fig. 4c).

Finally, we assessed the performance of probabilistic fine mapping in identifying the true effect isoform in our simulation framework of genes with 5 or 10 isoforms (Methods, Extended Data Fig. 7c and Supplementary Data 9). The sensitivity of 90% credible sets (proportion

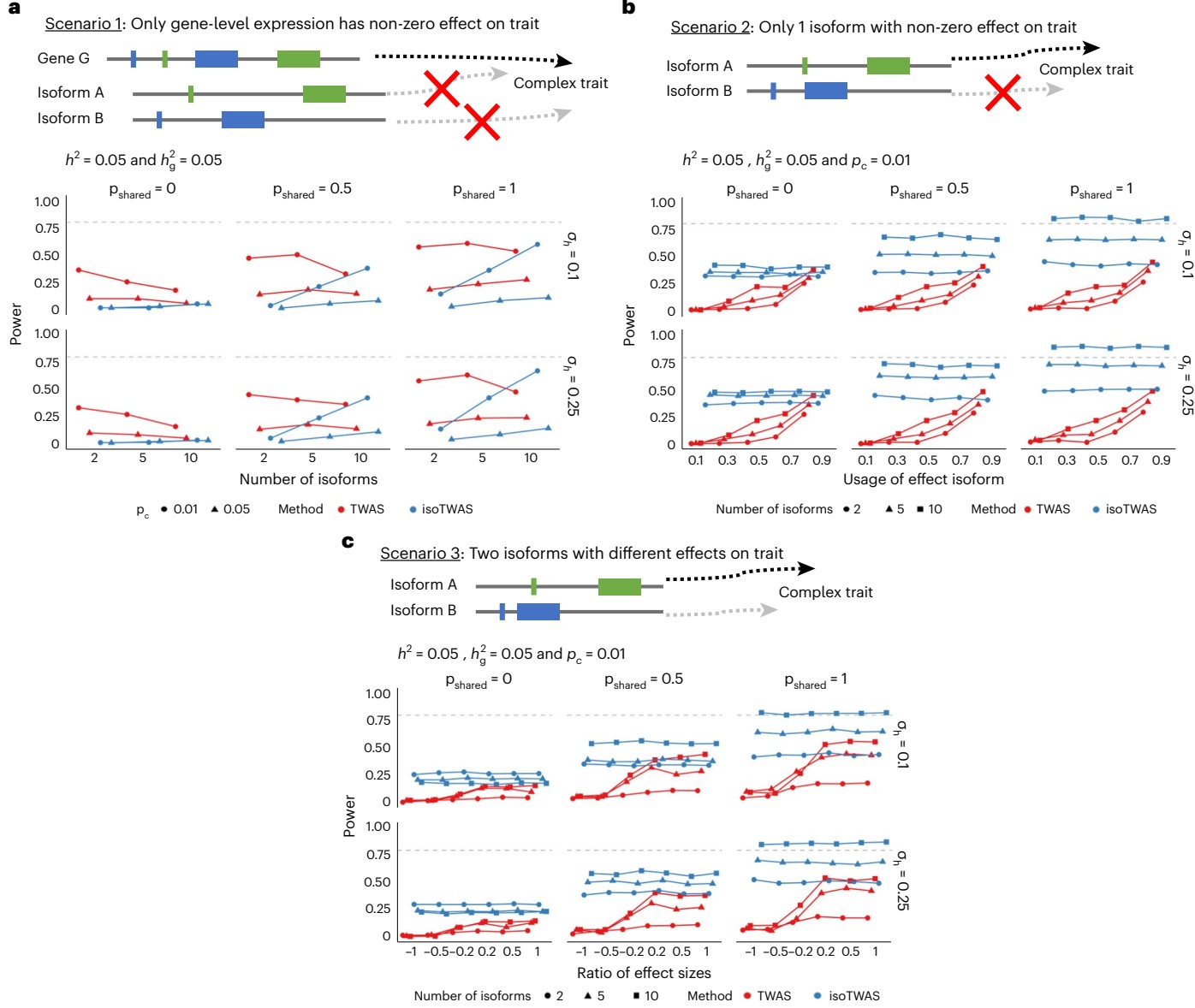

**Fig. 4 | IsoTWAS improves power to detect gene-trait associations in simulations, especially when genetic effects differ across isoforms.**
**a**, Schematic for only gene-level expression affecting the complex trait (top). Power to detect gene-trait association (proportion of tests with $P < 2.5 \times 10^{-6}$ using weighted burden test, $y$ axis) across varying numbers of isoforms per gene, causal proportion of isoQTLs ($p_c$), proportion of shared isoQTLs ($p_{shared}$) and proportion of variance explained by shared non-*cis*-genetic effects ($\sigma_h$).
**b** Schematic for only one isoform of a gene affecting the complex trait, whereas the others do not (top). Power to detect gene-trait association (proportion of tests with $P < 2.5 \times 10^{-6}$ using weighted burden test, $y$ axis) across proportion of gene expression explained by effect isoform ($x$ axis). **c**, Schematic for two isoforms of the same gene affecting the complex trait with varied effect sizes (top). Power to detect gene-trait association (proportion of tests with $P < 2.5 \times 10^{-6}$ using weighted burden test, $y$ axis) across ratio of effect sizes for two effect isoforms. Across all plots, isoform and total gene expression heritability is set to 0.05 and causal proportion of 0.01 in **b** and **c**. Dashed lines in **a**–**c** represent 80% power.

of credible sets containing the true effect isoform) was undercalibrated, likely due to difficulties in fine mapping when QTL horizontal pleiotropy is high[50]. With increasing proportions of shared isoQTLs, the sensitivity of 90% credible sets decreased and the mean set size increased. Our simulation results suggest that varied isoQTL architectures and isoform–trait effects for isoforms of the same gene are key features that influence power gains in isoform-centric modeling.

**Improved trait mapping across 15 neuropsychiatric GWAS**
To explore our central hypothesis that isoform-centric multivariate prediction improves discovery for complex trait mapping, particularly for brain relevant traits, we next compared isoTWAS/TWAS trait mapping across 15 neuropsychiatric traits. To maximize discovery,

we trained both isoTWAS and TWAS models using a large adult brain functional genomics reference panel ($N = 2,115$), composed of frontal cortex samples from PsychENCODE and AMP-AD Consortia[20,51], and using a developmental[22] prefrontal cortex ($N = 205$) dataset (Methods, Fig. 5 and Extended Data Fig. 8). In the adult cortex, we trained models for 15,127 genes using isoTWAS passing the CV $R^2 > 0.01$ cutoff, compared to 14,283 genes using gene-level TWAS. In the developing cortex, despite a smaller sample size, 16,504 and 10,535 models for genes were successfully trained using isoTWAS and TWAS, respectively (Methods and Supplementary Table 1).

We applied these models to perform trait mapping using summary statistics from 15 brain-related GWAS[52–66] (Methods, Fig. 5a and Extended Data Fig. 8a) using the stepwise hypothesis-testing procedure

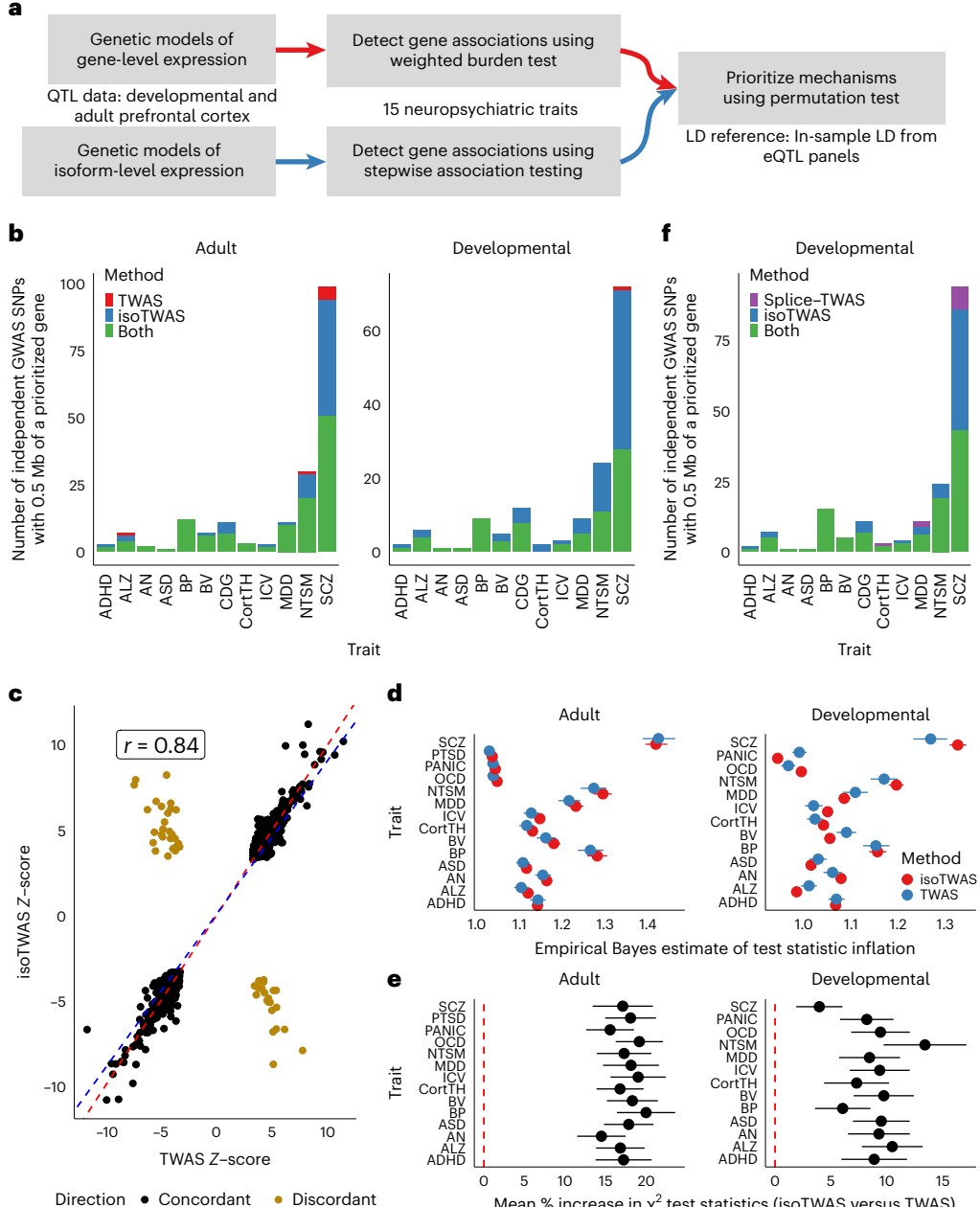

**Fig. 5 | Isoform-level trait mapping increases discovery of genetic associations over gene-level trait mapping. a**, Schematic diagram for trait mapping using gene-level TWAS and isoTWAS using PsychENCODE data. **b**, Number of gene-trait associations overlapping GWAS risk loci using gene-level TWAS (red), isoTWAS (blue) or either (green) at conservative permutation-based significance thresholds. **c**, Scatterplot of standardized effect sizes (Z-scores) for significant associations ($P_{nominal} < 0.05$) using isoTWAS and gene-level TWAS. **d**, Empirical Bayes estimate with jackknifed 95% confidence interval of test

statistic inflation (x axis) for TWAS (blue) and isoTWAS (red) gene-level associations across 15 traits (y axis) ($n = 1,000$ independent draws from posterior). **e**, Mean percent increase in approximate $\chi^2$ test statistic (squared Z-score) with jackknifed 95% confidence interval, which is proportional to increase in effective sample size, for isoTWAS trait associations over TWAS trait associations ($n = 1,624$-6,982 gene-level test statistics). **f**, Number of gene-trait associations overlapping GWAS loci using splicing-event-based TWAS (purple), isoTWAS (blue) or either (green) at conservative permutation-based significance thresholds.

(false discovery rate-adjusted $P < 0.05$ and within-locus permutation $P_{ACAT} < 0.05$). We detected more trait-associated genes with isoTWAS compared with TWAS, across adult (2,595 versus 1,589 genes) and developmental (4,062 versus 890 genes) reference panels, respectively (Extended Data Fig. 8b and Supplementary Data 10–13). Across both reference panels and all 15 traits, isoTWAS detected 3,436 unique gene and 5,377 unique isoform–trait associations (Extended Data Fig. 8c). Of the 1,335 genes with multiple isoform–trait associations, 661 genes exhibited distinct isoform-level associations in different directions.

We next compared the performance of isoTWAS/TWAS in prioritizing candidate mechanisms within independent, high-confidence GWAS-significant loci[67]. Across a combined 1,149 GWAS loci, isoTWAS identified significant associations within 323, compared with 201 detected by TWAS, a ~ 60% increase in discovery (Fig. 5b, Methods and Supplementary Table 4). Of the 287 GWAS loci identified for SCZ[68], isoTWAS prioritized genes within 70 and 86 unique loci across adult and developmental cortex, respectively, compared with 56 and 29 loci for TWAS (Fig. 5b). Furthermore, 96% of gene-level TWAS associations

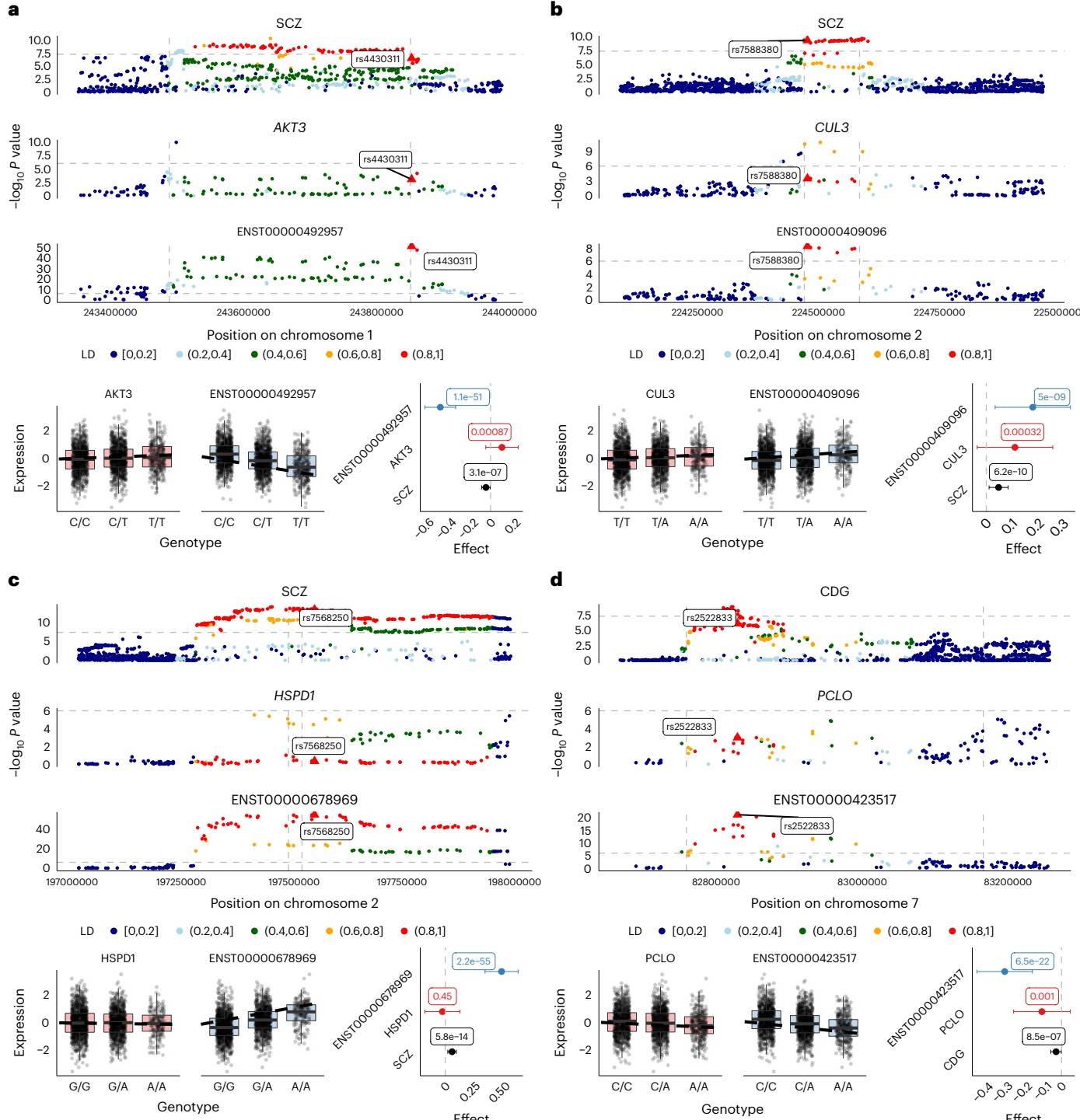

**Fig. 6 | isoTWAS implicates isoforms of AKT3, CUL3, HSPD1, and PCLO in genetic associations with psychiatric traits.** Top: Manhattan plots of GWAS loci, eQTLs and isoQTLs, colored by LD to the lead isoQTL (triangle shape). LD is based on the 1000 Genomes European reference. Vertical gray dashed lines indicate the transcription start and end sites for each gene, and the horizontal gray dashed lines indicate $P = 5 \times 10^{-8}$ for GWAS and $10^{-6}$ for QTLs. Bottom: Boxplots of gene and isoform expression by genotype of the lead isoQTL SNP and forest plot of the lead isoQTL's effect size and 95% confidence interval on the trait, gene and isoform, with $P$ values labeled for two-sided Wald-type $t$-test from linear regression ($n = 2{,}115$ biologically independent samples). All boxplots represent the median, 25% and 75% quantiles, and whiskers correspond to the 10% and 90% quantiles. **a**, SCZ GWAS, *AKT3* gene expression QTLs and ENST00000492957 isoform expression QTLs. **b**, SCZ GWAS, *CUL3* gene expression and ENST00000409096 isoform expression QTLs. **c**, SCZ GWAS, *HSPD1* gene expression and ENST00000678969 isoform expression QTL. **d**, CDG GWAS, *PCLO* gene expression and ENST00000423517 isoform expression QTLs.

(193/201) were concordantly identified by isoTWAS. Likewise, the standardized effect sizes for significant gene- and isoform-level associations were highly correlated ($r = 0.84$, $P < 2.2 \times 10^{-16}$; Fig. 5c). Finally, to explore whether these isoTWAS-specific associations were capturing true disease signal, we compared the rate at which each method prioritized constrained genes (probability of loss-of-function intolerance, pLI $\geq 0.9$; Supplementary Tables 5–8), which are known to be substantially enriched for disease associations[69]. Across adult

and developmental panels, respectively, isoTWAS prioritized 724 and 385 constrained genes compared to 106 and 200 with TWAS (Fisher's exact test, adult: $P = 0.048$, developmental: $P = 1.23 \times 10^{-5}$). Altogether, isoTWAS not only recovers the vast majority of TWAS associations but also increases discovery of candidate GWAS mechanisms, particularly for genes intolerant to protein-truncating variation[70].

To investigate whether this increase in trait mapping discovery reflected true biological signal rather than test statistic inflation due to the increased number of tests (~4-fold increase in number of tests), we next compared the null distributions across methods for results (Extended Data Fig. 9). As the genomic inflation factor is not a reliable measure in TWAS settings[71], we estimated inflation in gene-level test statistics using an empirical Bayes approach (Methods). There were no significant differences between TWAS and isoTWAS in the 95% credible intervals for test statistic inflation (Fig. 5d). Using a heuristic to estimate increases in effective sample size (Methods), we observed an approximate increase in effective sample size of 10–20% when using isoTWAS compared to TWAS (Fig. 5e and Supplementary Table 9). These analyses indicate that isoTWAS discovery is well-calibrated to the null and facilitates increased discovery in real data compared to gene-level TWAS.

We empirically compared probabilistic fine mapping[50] of results from isoTWAS and gene-level TWAS (Methods and Extended Data Fig. 8d). Here, we conducted fine mapping in loci with one or more significant trait-associated genes/isoforms (adjusted $P < 0.05$ and permutation $P < 0.05$) within 1 Mb of one another, termed risk regions. Overall, the mean number of genes in a risk region using TWAS was 3.15 compared to 3.90 using isoTWAS; the mean number of genes in a 90% credible set using TWAS was 1.33 compared to 1.25 using isoTWAS. On average, there were 1.54 isoforms per gene in a risk region and 1.27 isoforms per gene in a 90% credible set. Isoform-centric modeling presents unique challenges for fine mapping due to potentially high levels of horizontal pleiotropy and remains an important and open question for the field. Nevertheless, isoTWAS identified a comparable number of genes in risk regions compared with TWAS, and the combination of two-step trait mapping, permutation testing, and probabilistic fine mapping maintained narrow credible set sizes.

Lastly, we compared discovery using isoTWAS to discovery using local splicing-event-based trait mapping. For the developmental brain dataset, we calculated intron usage using LeafCutter[72] and transformed these usage percentages to M-values[73]. Then, for all introns mapped to a given gene, we used all SNPs within 1 Mb of a splicing event to predict its usage and mapped trait associations for these splicing events using isoTWAS's multivariate framework (Methods). Overall, when aggregated to the gene-level, across 15 traits, we found that isoTWAS prioritized features at ~40% more independent GWAS loci (167 loci) than splicing-event-based trait mapping (119 loci), with 108 loci (90.7%) jointly identified (Fig. 5f), using the same developmental brain reference panel. Taken together, isoTWAS's specific focus on modeling isoforms of a gene provided gains in trait association discovery over considering only total gene expression or intron usage.

### isoTWAS identifies trait associations undetectable by TWAS

Overall, isoTWAS prioritized dozens of candidate risk genes and mechanisms in the developing and adult brain for 15 neuropsychiatric traits. These isoTWAS-prioritized genes were enriched for relevant pathways consistent with the biology of the underlying trait: cell proliferation for brain volume (BV), calcium channel activity for SCZ and neuroticism (NTSM), and proteasome regulation in Alzheimer's disease (ALZ) (Extended Data Fig. 10a). In the Supplementary Note, we discuss several examples of trait associations for which isoTWAS prioritized a highly constrained gene within a GWAS locus (Supplementary Tables 5–8)[74–79].

A main advantage of isoTWAS over TWAS is the identification of trait associations for isoforms of genes, where the gene itself is not associated with the trait. We illustrate several examples of

isoTWAS-prioritized isoforms, all in the adult cortex, for genes with limited or distinct expression QTLs (Fig. 6, Extended Data Fig. 10b and Supplementary Data 14), with exon/intron structure shown in Supplementary Figs 1–4. First, we detected a SCZ association with ENST00000492957, an isoform of *AKT3* (1q43-144, pLI = 1), which encodes a serine/threonine-protein kinase that regulates cell life cycle (e.g., growth, proliferation and survival). *AKT3* has shown effects on anxiety, spatial-contextual memory, and fear extinction in mice, and loss-of-function of *AKT3* causes learning and memory deficits[80,81]. Within the GWAS locus, there was a strong overlapping isoQTL signal ($P < 10^{-50}$) but only one eQTL with $P < 10^{-6}$, which is in low LD with the GWAS-significant SNPs (Fig. 6a). The lead isoQTL (rs4430311) showed a significant, negative association with ENST00000492957, but a nominally significant positive association with *AKT3* expression. Interestingly, a different isoform of *AKT3* (ENST00000681794) was prioritized in an association with BV, which also has a GWAS association at this locus (Extended Data Fig. 10b). The two distinct isoforms of *AKT3* have distinct 3' transcript structures, close to the lead isoQTL of ENST00000681794. These results suggest a complex role of *AKT3* isoforms with brain-related traits to be explored further.

Similarly, we found a strong isoQTL signal for ENST00000409096 but a weak eQTL signal of its gene *CUL3* in the 2q36.2 locus (pLI = 0.99), in another association with SCZ (Fig. 6b). *CUL3* is involved in cell cycle regulation, protein trafficking and signal transduction, and its dysregulation is a potential mechanism for both SCZ and autism spectrum disorder (ASD) risk[82]. Next, isoform ENST00000678969 of *HSPD1*, encoding a mitochondrial heat shock protein, was associated with SCZ risk (pLI = 0.99, 2q33.1) and showed a similar pattern across GWAS, eQTL and isoQTL signals (Fig. 6c). *HSPD1* is among multiple non-MHC immune genes implicated in SCZ and has roles in brain hypomyelination[83]. Lastly, ENST00000423517, an isoform of *PCLO*, was associated with multiple traits in the cross-disorder (CDG) GWAS (meta-analysis of attention deficit hyperactivity disorder, bipolar disorder, major depression and SCZ, pLI = 1, 7q21.11). Again, we found a strong isoQTL but not eQTL signal, with the CDG risk allele negatively associated with isoform expression. *PCLO* is involved in the presynaptic cytoskeletal matrix, establishing active synaptic zones, and synaptic vesicle trafficking; rare variants of *PCLO* in diverse populations have been recently implicated in risk of SCZ and ASD[84,85]. Altogether, these results highlight the importance of incorporating isoform-level regulation for prioritizing novel candidate GWAS risk mechanisms, as implemented in our isoTWAS framework.

## Discussion

We present isoTWAS, a framework that integrates genetic and isoform-level transcriptomic variation with GWAS to identify gene expression-trait associations and prioritize a set of isoforms of the gene that best explains the association. We provide an extensive set of isoform-level predictive models[86–88] and software to train models and conduct isoform-level trait mapping with GWAS summary statistics[47].

isoTWAS presents several advantages over gene-level TWAS or univariate modeling of isoform expression. First, modeling expression at the isoform-level can detect isoQTL architectures that vary across isoforms and are not captured by gene-level eQTLs. Second, joint multivariate isoform-level modeling improved predictive accuracy of isoform and total gene expression. Third, aggregating isoform-level associations to the gene-level substantially increased power to detect trait associations. We attribute this increase in power to three features: (1) isoform-level modeling in isoTWAS increases the number of imputable genes by >2-fold, (2) isoTWAS models improve gene-level prediction up to 35% and (3) isoTWAS jointly models expression and splicing regulation, capturing underlying complex trait mechanisms. Finally, as genetic control of isoform expression is often more tissue- and cell-type-specific than eQTLs[26,35], we hypothesize that isoTWAS is more capable of uncovering context-specific trait associations.

Recent work has highlighted alternative splicing as a promising mechanism underlying complex traits not captured through eQTLs[20,22,26,89], as mapping genetic regulation at the exon- rather than gene-level often leads to more detected signal[90]. However, most of these analyses focused on local splicing events or exon-level inclusion, rather than different isoforms of the same gene, which reflect the combined consequences of these splicing events. Local splicing events can be difficult to systematically measure and integrate across multiple large-scale datasets, which is necessary for achieving sufficient sample sizes to interrogate population-level allelic effects[20,21]. Our results demonstrate that isoform-centric trait mapping with isoTWAS increases discovery by ~40% compared with a matched local splicing-event-based analysis, although these methods may recover some independent signal. Future work should integrate reference-guided and annotation-free approaches for isoform and local splicing quantification to develop nuanced mechanistic hypotheses for GWAS loci.

We conclude with limitations of and future considerations for isoTWAS. First, isoform-level expression quantifications are maximum-likelihood estimates, due to limitations of short-read RNA-seq. These estimates are guided by existing transcriptome annotations and thus are dependent on their completeness and accuracy. Further, dataset-specific sequencing factors will affect the accuracy of these estimates, especially sequencing depth, read length, and library preparation. The emergence of long-read sequencing platforms will be instrumental for improving tissue-specific reference transcriptome annotations, which, in turn, will improve isoTWAS. As these methods continue to gain scalability and cost-effectiveness, they will ultimately replace short-read sequencing and isoform estimation for population-scale datasets. isoTWAS is agnostic to the method of isoform expression quantification and will continue to be applicable as we approach the long-read sequencing era.

Second, although inferential replicates from RNA-seq quantification can provide measures of technical variation, they are not incorporated into the predictive models. Our analyses of prediction across inferential replicates suggest a methodological opportunity: leveraging these inferential replicates as a measure of quantification error to estimate the robustness of isoform prediction and the precision of SNP effects. A predictive model that estimates standard errors for SNP effects by model averaging across replicates may improve trait mapping by providing a prediction interval for imputed expression. Third, as isoform-level trait mapping is akin to differential transcript expression analysis, isoTWAS can be extended to analyses of genetically regulated transcript usage. However, it is unclear if the compositional nature of transcript usage data needs to accounted for during prediction or trait mapping[91]. Lastly, isoTWAS can suffer from reduced power, inflated false positives and reduced fine-mapping sensitivity in the presence of SNP horizontal pleiotropy[92,93]. For pathways that are not observed or accounted for in the reference expression panel and GWAS, accounting for horizontal pleiotropy may improve trait mapping. We motivate extensions of probabilistic fine mapping to reconcile pleiotropy for SNPs shared across models for multiple isoforms at the same genetic locus, as summary-statistic-based methods that control for horizontal pleiotropy are not yet effective[94].

isoTWAS provides a flexible framework to interrogate the transcriptomic mechanisms underlying genetic associations with complex traits and generate biologically meaningful and testable hypotheses about disease risk mechanisms. We emphasize a shift in focus from quantifications of the transcriptome on the gene-level to the transcript-isoform level to maximize discovery of transcriptome-centric genetic associations with complex traits.

## Online content

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

## Methods

### Ethical approval
We use public data with previous ethical approval[20,22,35,51–66], and our study did not need any specific approval.

### Overview of isoTWAS
isoTWAS consists of three steps: (1) training predictive models of isoform expression, (2) imputing isoform-specific expression into a separate GWAS panel and (3) association testing between imputed expression and a phenotype (Fig. 1b). isoTWAS contrasts with TWAS as it models correlations between the expression of isoforms of the same gene. Further mathematical details are provided in Supplemental Methods.

### Training predictive models of isoform expression
**Model and assumptions.** Assume a gene $G$ has $M$ isoforms with expression levels across $N$ samples, with each sample having $R$ inferential replicates. Let $Y_G^*$ be the $N \times M$ matrix of mean isoform expression (log-scale TPM) for the $N$ samples and $M$ isoforms, using the expectation-maximization point estimates from a pseudo-mapping quantification algorithm, like Salmon or kallisto[30,31]. We can jointly model isoform expression with a system of $N \times M \times R$ equations. For sample $n \in \{1, \dots, N\}$, isoform $m \in \{1, \dots, M\}$ of gene $G$, and replicate $r \in \{1, \dots, R\}$, we have:

$$y_{nmr} = \mathbf{x_n}\boldsymbol{\beta_m} + \epsilon_{nmr}, \qquad (1)$$

where $y_{nmr}$ is the expression of isoform $m$ for the $r$th inferential replicate of sample $n$, $\mathbf{x_n}$ is the $P$-vector (vector of length $P$) of *cis*-genotypes in a 1 Mb window around gene $G$, $\boldsymbol{\beta_m}$ is the $P$-vector of genetic effects of the $P$ genotypes on isoform expression, and $\epsilon_{nmr}$ is normally distributed random noise with mean 0 and variance $\sigma_{nmr}^2$. We standardize both the genotypes and the isoform expression to mean 0 and variance 1. As the SNP vector $\mathbf{x_n}$ does not differ across replicates, we assume that $\epsilon_{nmr}$ are independent and identically distributed across samples $n \in \{1, \dots, N\}$ and identically distributed across replicates $r \in \{1, \dots, R\}$. Accordingly, the point estimates of the SNP effects on isoform expression are not influenced by differences in expression across replications. Therefore, in matrix form, we consider the following predictive model:

$$Y_G^* = X_G B_G + E_G. \qquad (2)$$

Here, $X_G$ is the $N \times P$ matrix of genotype dosages, $B_G$ is the $P \times M$ matrix of SNP effects on isoform expression and $E_G$ is a matrix of random errors, such that $vec(E_G) \sim N_{NM}(0, \Sigma = \Omega^{-1} \otimes I_N)$. $\Sigma$ represents the variance-covariance matrix in the errors (with precision matrix $\Omega = \Sigma^{-1}$), following the above independence assumptions.

**Estimating SNP effects on isoform expression.** We apply five methods to estimate $\hat{B}_G$, the matrix of SNP effects on isoform expression. The first four are multivariate methods that model the isoforms jointly; the last method models each isoform separately using univariate methods. The goal of this SNP effect estimation is marginal prediction, that is, leveraging the correlation between isoforms to improve prediction of each isoform separately. The $\hat{B}_G$ matrix that gives the largest adjusted R² in 5-fold CV across the five methods is selected as the final model to predict isoform expression for a given gene. When interested in predicting gene-level expression from these predicted isoforms, isoTWAS trains an elastic net penalized linear regression that predicts gene-level expression from genetically-predicted isoform-level expression; this model training is conducted across the same 5 folds to prevent data leakage[95]. We train 4 multivariate models and 1 univariate model to marginally predict isoform expression (Supplemental Methods):

(1) *Multivariate elastic net (MVEnet) regression*: This is an extension of elastic net, where the response is a matrix of correlated responses[41]. The absolute penalty is imposed on each coefficient by a group-lasso penalty on each vector of SNP effects across isoforms (rows of $B_G$). Accordingly, a SNP can only have a non-zero effect on an isoform if it has a non-zero effect on all isoforms.

(2) *Multivariate LASSO regression with covariance estimation (MRCE)*: We adapt Rothman et al's proposed procedure to simultaneously and iteratively estimate both $\hat{B}_G$, the SNP effects matrix, and $\hat{\Omega}$, the precision matrix[40]. This procedure accounts for the correlation between isoforms but does not impose the group-lasso penalty as in MVEnet.

(3) *Multivariate elastic net with stacked generalization (joinet)*: We use Rauschenberger and Glaab's joinet method that uses a two-step prediction[42]: first, the design matrix of SNPs is used to generate a cross-validated prediction of each isoform, and second, the matrix of predicted isoform expression is used to predict each isoform.

(4) *Sparse partial least squares (SPLS)*: This is an implementation of partial least squares with a sparsity penalty, that attempts to find an optimal latent decomposition for the linear relationship between the matrix of isoform expression and the design matrix of SNPs. We use the Chun and Keles's implementation from the spls R package[39].

(5) *Univariate FUSION*: We disregard the correlation structure between isoforms and train a univariate elastic net[41], estimation of the best linear unbiased predictor (BLUP) in a linear mixed model[44], and SuSiE[43] predictive model for each isoform separately. The model with the largest adjusted $R^2$ out of these three models is outputted. This approach serves as a baseline measurement for prediction of each isoform independently.

### Trait association and stepwise hypothesis testing
The tests of association in isoTWAS are like tests in differential transcript expression analyses, as TWAS tests of association are analogous to tests in differential gene expression analyses. isoTWAS and TWAS are distinct, as these methods consider imputed isoform and gene expression, respectively, as predicted by the trained expression models. If individual-level genotypes are available in the external GWAS panel, isoform expression can be directly imputed by multiplying the SNP weights from the predictive model with the genotype dosages in the GWAS panel. If only summary statistics are available, we adopt the weighted burden test from Gusev et al. with an ancestry-matched LD panel[4,93]. Compared to TWAS, isoTWAS association testing involves an increased number of tests (~4 isoforms per gene)[24] and potential correlation in test statistics for isoforms of the same gene.

We perform a two-step hypothesis-testing framework (Extended Data Fig. 1b)[46]. In the first step, for every isoform with a trained model, we generate a test statistic using either linear regression for GWAS with individual-level genotypes or the weighted burden test for GWAS with only summary statistics[4]. Given the $t$ test statistics $T_1, \dots, T_t$ for isoforms for a gene, an omnibus test aggregates the $t$ test statistics into a single $P$ value for a gene. We benchmarked different omnibus tests in simulations, but the default omnibus test in isoTWAS is ACAT[45]. We control for false discovery across all genes via the Benjamini-Hochberg procedure, but the Bonferroni procedure can also be applied for more conservative false discovery control. In the second step, for isoforms for genes with an adjusted omnibus $P < 0.05$, we employ Shaffer's modified sequentially rejective Bonferroni procedure to control the within-gene family-wide error rate. At the end of these two steps, we identify a set of genes and their isoforms that are associated with the trait.

## Control for false positives within GWAS loci

In TWAS and related methods, association statistics have been shown to be well-calibrated under the null of no GWAS association. However, within loci harboring significant GWAS signal, false positive associations can result when eQTLs and GWAS coincide within overlapping LD blocks. To address this, we adopt two conservative approaches to control for type 1 error within GWAS loci, namely (1) permutation testing and (2) probabilistic fine mapping. The permutation testing approach, adopted from Gusev et al[4], is a highly conservative test of the signal added by the SNP-transcript effects from the predictive models, conditional on the GWAS architecture of the locus. Briefly, we permute the SNP-transcript effects in the predictive models 10,000 times and generate a null distribution for the isoform test statistic. We use this null distribution to generate a permutation-based P value for the original test statistic for each isoform. Finally, we can use isoform-level probabilistic fine mapping using methods from FOCUS[50] to generate credible set of isoforms that explain the trait association at a locus. We only run isoform-level fine mapping for significantly associated isoforms in overlapping 1-Mb windows.

## Simulation framework

We adopt techniques from Mancuso et al's *twas_sim* protocol[96] to simulate multivariate isoform expression based on randomly simulated genotypes and environmental random noise. First, for $n$ samples, we generate a matrix of genotypes for the SNPs within 1 Mb of 22 different genes (1 per chromosome) using an LD reference panel of European subjects from 1000 Genomes Project[48].

Next, we generate a matrix of SNP-isoform effects across different causal SNP proportions $p_c$, numbers of isoforms $t$, and $p_s$ proportion of the SNP-isoform effects being shared across isoforms of the same gene. We then add two matrices of random noise $U$ and $\epsilon$. The first matrix $U$ noise represents non-*cis*-genetic effects on isoforms that are correlated between samples and isoforms; we control the proportion of variance explained in isoform expression attributed to $U$ using a parameter $\sigma_h$. The second matrix $\epsilon$ is a matrix of random noise that is independent for each isoform, such that $\epsilon_i \sim N(0, \sigma_e^2 I)$ where $\sigma_e^2 = 1 - \sigma_h - h_g^2$. We generate 10,000 simulations for each configuration of the simulation parameters, varying $n \in \{200, 500\}$, $p_c \in \{0.001, 0.01, 0.05\}$, $h_g^2 \in \{0.05, 0.10, 0.25\}$, $p_s \in \{0, 0.5, 1\}$, and $\sigma_h \in \{0.1, 0.25\}$. Further details are provided in Supplementary Methods and summarized in Fig. 2.

We also generate traits under three distinct scenarios, with a continuous trait with heritability $h_t^2 \in \{0.01, 0.05, 0.10\}$ and a GWAS sample size of 50,000 (Supplementary Methods):

(1) *Only gene-level expression has a non-zero effect on trait.* Here, we sum the isoform expression to generate a simulated gene expression. We randomly simulate the effect size and scale the error to ensure trait heritability.

(2) *Only one isoform has a non-zero effect on the trait.* Here, we generate a multivariate isoform expression matrix with 2 isoforms and scale the total gene expression value such that one isoform (called the effect isoform) makes up $p_g \in \{0.10, 0.30, 0.50, 0.70, 0.90\}$ proportion of total gene expression. We then generate effect size for one of the isoforms and scale the error to ensure trait heritability.

(3) *Two isoforms with different effects on traits.* Here, we generate a multivariate isoform expression matrix with 2 isoforms that make up equal portions of the total gene expression. We then generate an effect size of $\alpha$ for one isoform and $p_e \alpha$ for the other isoform, such that $p_e \in \{-1, -0.5, -0.2, 0.2, 0.5, 1\}$. We then scale the error to ensure trait heritability.

To estimate the approximate FPR, we followed the same simulation framework to generate eQTL data and GWAS data. In the GWAS data, we set the effect of gene- and isoform-level imputed expression to 0 to generate a simulated trait under the null. We then estimated the FPR by calculating the proportion of gene-trait associations at $P < 0.05$ under this null across 20 sets of 1,000 simulated GWAS panels. We also assessed isoform-level fine mapping using FOCUS in a scenario with a gene with 5 or 10 isoforms and a single effect isoform. We computed the sensitivity of 90% credible sets of isoforms (proportion of credible sets that contain the effect isoform) and the number of isoforms in the 90% credible set.

## GTEx processing and model training

We quantified GTEx v8 (ref. [35]) RNA-seq samples for 48 tissues using Salmon v1.5.2 (ref. [30]) in mapping-based mode. We first built a Salmon index for a decoy-aware transcriptome consisting of GENCODE v38 transcript sequences and the full GRCh38 reference genome as decoy sequences[24]. Salmon was then run on FASTQ files with mapping validation and corrections for sequencing and GC bias. We computed 50 inferential bootstraps for isoform expression. We then imported Salmon isoform-level quantifications and aggregated to the gene-level using txmeta v1.16.1 (ref. [37]). Using edgeR, gene and isoform-level quantifications underwent TMM-normalization, followed by transformation into a log-space using the variance-stabilizing transformation using DESeq2 v1.38.3 (ref. [97,98]). We then residualized isoform-level and gene-level expression (as log-transformed CPM) by all tissue-specific covariates (clinical, demographic, genotype principal components (PCs), and expression PEER factors) used in the original QTL analyses in GTEx. We calculated the quantification variance across inferential replicates using the computeInfRV() function from fishpond v2.4.1 (ref. [99]). We computed the isoform fraction using the isoformToIsoformFraction() function from IsoformSwitchAnalyzeR v1.20.0 (ref. [100]).

SNP genotype calls were derived from Whole Genome Sequencing data for samples from individuals of European ancestry, filtering out SNPs with minor allele frequency (MAF) less than 5% or that deviated from HWE at $P < 10^{-5}$. We further filtered out SNPs with MAF less than 1% frequency among the European ancestry samples in 1000 Genomes Project[48].

Details of the model training pipeline for GTEx are similar to those in Extended Data Fig. 8a. Gene-level univariate models were trained using elastic net regression[41], BLUP in a linear mixed model[44], and SuSiE[43], using all SNPs within 1 Mb of the gene body[4,41,43,44]. For each gene, the best performing model was chosen based on McNemar's adjusted 5-fold CV $R^2$. We selected only genes with CV $R^2 \geq 0.01$. We applied multivariate modeling outlined in isoTWAS to train isoform-level predictive models, selecting only those isoform models with CV $R^2 \geq 0.01$. All isoTWAS models generated are publicly available (see Data availability).

## Developmental brain reference panel processing and model training

We quantified developmental frontal cortex[22] ($N = 205$) RNA-seq samples using Salmon v1.8.0[30] in mapping-based mode. We used the same indexed transcriptome as in the GTEx analysis and ran Salmon with mapping validation and corrections for sequencing and GC bias. We computed 50 inferential bootstraps for isoform expression using Salmon's Expectation-Maximization algorithm. We then imported Salmon isoform-level quantifications and aggregated to the gene-level using txmeta[37]. Using edgeR v3.40.2, gene and isoform-level quantifications underwent TMM-normalization, followed by transformation into a log-space using the variance-stabilizing transformation using DESeq2 v1.38.3[97,98]. We then residualized isoform-level and gene-level expression (as log-transformed CPM) by covariates (age, sex, 10 genotype PCs, 90 and 70 hidden covariates with prior (HCP), respectively). Typed SNPs with non-zero alternative alleles, MAF >1%, genotyping rate >95%, Hardy Weinberg equilibrium (HWE) $P < 10^{-6}$ were first imputed to TOPMed Freeze 5 using minimac4 and eagle v2.4 (refs. [101,102]). We then retained biallelic SNPs with imputation accuracy $R^2 > 0.8$, with

rsIDs. Finally, we filtered out SNPs with MAF < 0.05 or that deviated from Hardy–Weinberg equilibrium at $P < 10^{-6}$.

## Adult brain reference panel processing and model training

Matched genotype and RNA-seq data from adult brain cortex tissue from 2,365 individuals were compiled and processed from the PsychENCODE Consortium[20] and the Accelerating Medicines Partnership Program for Alzheimer's Disease (AMP-AD)[51], consisting of the individual studies BipSeq, BrainGVEX, CommonMind Consortium (CMC), CommonMind Consortium's National Institute of Mental Health Human Brain Collection Core (CMC HBCC), Lieber Institute for Brain Development-szControl (LIBD_szControl), UCLA-ASD, Religious Orders Study and the Memory and Aging Project (ROSMAP), Mount Sinai Brain Bank (MSBB) and MayoRNAseq.

Typed genotypes were lifted over to the GRCh38 build using CrossMap v.0.6.3 (ref. 103) and then filtered to remove variants where the reference allele matched any of the alternate alleles. Genotype data from whole genome sequencing (BrainGVEX, UCLA-ASD, ROSMAP, MSBB and MayoRNAseq) were further filtered to variants present on the Infinium Omni5-4 v1.2 array in order to satisfy the imputation server's maximum limit of 20,000 typed variants per 20 Mb. All genotype data were further processed with PLINK v1.90b6.21 (ref. 104), removing variants with HWE $P < 10^{-6}$, MAF < 0.01 or missingness rate > 0.05, and removing samples with missingness rate > 0.1 across typed variants or missingness rate > 0.5 on any individual chromosome. Genotype data was prepared for imputation using the McCarthy Group's HRC-1000G-check-bim-v4.3.0 tool against freeze 8 of the Trans-Omics for Precision Medicine (TOPMed) reference panel[105]. The tool removes A/T and G/C SNPs with MAF > 0.4, variants with alleles that differ from the reference panel, variants with an allele frequency difference > 0.2 from the reference panel and variants not in the reference panel. Additionally, the tool updates strand, position and reference/alternate allele assignment to match the reference panel.

Genotypes were then passed into the TOPMed Imputation Server by individual array batch[106]. The genotypes were phased with Eagle v2.4 and imputed with Minimac4 using the TOPMed reference panel[101,102]. Further QC was performed on the imputed genotypes using bcftools v1.11 and PLINK. The imputed genotypes were filtered to well-imputed variants with $R^2 > 0.8$. The arrays were merged after filtering to variants that were well imputed in all arrays to be merged. Only arrays with at least 400,000 variants after pre-imputation QC were merged in order to prevent too many variants from dropping out. The merged genotype data were then converted to PLINK 1 binary format and further processed with PLINK, removing variants with duplicates, HWE $P < 10^{-6}$, MAF < 0.01 or missingness rate > 0.05 and removing samples with missingness rate > 0.1. Samples from the same individual were identified by calculating the genetic relatedness matrix using SnpArrays.jl and finding sets of samples with relatedness > 0.75. From each set of replicates, only the genotyped sample from the array with the most variants after pre-imputation QC was kept. For model training, only SNPs annotated in HapMap3 were retained[107].

RNA-seq paired reads from each study were sorted by name and then converted to FASTQ format using samtools v1.14 (ref. 108). The reads were then quantified using salmon v1.8.0 in mapping-based mode using a full decoy indexed from GENCODE v38 transcriptome and GRCh38 patch 13 assembly[30]. Quantification was run using a standard EM algorithm with library type automatically inferred and estimates adjusted for sequence-specific and fragment-level GC biases. Bootstrapped abundance estimates were calculated using 50 bootstrap samples. Isoform-level expression was summarized to the gene-level using tximeta[37]. Only isoforms with 0.1 TPM for more than 75% of samples were retained. The resulting expression was normalized using the variance-stabilizing transformation from DESeq2 (ref. 98).

Samples with WGCNA network connectivity scores of less than -3 were removed as outliers, resulting in a total of 2,115 samples[109]. Isoform- and gene-level expression was then batch-corrected using ComBat (sva v3.46.0), using study site as the batch[110]. Lastly, age, age², sex, 10 genotype PCs and hidden covariates (200 for gene expression and 175 for isoform expression) were removed from the expression matrix[111,112]. The number of HCP were selected by optimizing the number of nominal *cis*-eQTLs and *cis*-isoQTLs at Bonferroni-corrected $P < 0.05$, respectively, on a grid from 100 to 300 HCPs, as detected by QTLtools v1.3.1 (ref. 90).

Details of the model training pipeline are summarized are equivalent to those used to train models in GTEx data.

## Gene- and isoform-level trait mapping

We conducted gene- and isoform-level trait mapping for 15 neuropsychiatric traits: attention-deficit hyperactivity disorder (ADHD, $N_{cases} = 20,183/N_{controls} = 35,191$)[53], ALZ (90,338/1,036,225)[54], anorexia nervosa (AN, 16,992/55,525)[66], ASD (18,381/27,969)[52], bipolar disorder (BP, 41,917/371,549)[55], BV ($N = 47,316$)[56], CDG (232,964/494,162)[57], cortical thickness (CortTH, $N = 51,665$)[58], intracranial volume (ICV, $N = 32,438$)[59], major depressive disorder (MDD, 246,363/561,190)[60], NTSM ($N = 449,484$)[61], obsessive compulsive disorder (OCD, 2,688/7,037)[62], panic and anxiety disorders (PANIC, 2,248/7,992)[63], post-traumatic stress disorder (PTSD, 32,428/174,227)[64] and SCZ (69,369/236,642)[65]. For gene-level trait mapping, we used the weighted burden test, followed by the permutation test, as outlined by Gusev et al[4]. For isoform-level trait mapping, we used the stage-wise testing procedure outlined in the isoTWAS method. In-sample LD from the QTL reference panels was used to calculate the standard error in the weighted burden test. For isoforms, irrespective of their corresponding genes, passing both stage-wise tests and the permutation test, we employed isoform-level probabilistic fine mapping using FOCUS with default parameters[50]. These methods are summarized in Extended Data Fig. 8a.

We estimated the percent increase in effective sample size by employing the following heuristic. We convert gene-level association $P$ values into $\chi^2$ test statistics with 1 degree of freedom. For $\chi^2 > 1$, we then calculate the percent increase for isoTWAS-based associations versus TWAS-based associations. As the mean of the $\chi^2$ distribution is linearly related to power and sample size[113], we can use this percent increase in test statistic as a measure of power or effective sample size. We defined independent genome-wide significant SNPs in GWAS by LD clumping with lead GWAS SNP < 5 ×$10^{-8}$ with $P$ value used for ranking and a $R^2$ threshold of 0.2.

## Statistics and reproducibility

For analysis of GTEx, PsychENCODE and AMP-AD data, no statistical method was used to predetermine sample size; the maximal sample size was determined by the number of individuals with both RNA-seq and genotype data. Exclusion criteria for these three datasets are included above, in detail. Briefly, as predetermined, GTEx data were restricted to individuals of European genetic ancestry to ensure portability of genetic predictions. PsychENCODE and AMP-AD individuals were removed if their WGCNA network connectivity scores based on isoform-level expression were less than −3; these low scores indicate that these samples may be plagued by technical biases that may affect the estimation of genetic effects on gene- and isoform-level expression. No data were collected directly in this work, and, as such, the investigators were blinded to allocation. Statistical analyses are summarized above and scripts to reproduce the analysis are listed in the code availability statement.

## Reporting summary

Further information on research design is available in the Nature Portfolio Reporting Summary linked to this article.

## Data availability

GTEx genetic, transcriptomic and covariate data were obtained through dbGAP approval at accession number phs000424.v8.p2 (ref. 35). LD reference data from the 1000 Genomes Project were obtained at https://www.internationalgenome.org/data-portal/sample (ref. 48). GENCODE reference transcriptome and assembly was downloaded from https://www.gencodegenes.org/human/release_38.html with GenBank assembly accession GCA_000001405.28 (ref. 24). GWAS summary statistics were obtained at the following links: ADHD (https://www.med.unc.edu/pgc/download-results/)[53], ALZ (https://ctg.cncr.nl/software/summary_statistics/)[54], AN (http://www.med.unc.edu/pgc/results-and-downloads)[66], ASD (https://www.med.unc.edu/pgc/download-results/)[52], BP (https://www.med.unc.edu/pgc/download-results/)[55], BV (https://ctg.cncr.nl/software/summary_statistics)[56], CDG (https://www.med.unc.edu/pgc/results-and-downloads)[57], CortTH (https://enigma.ini.usc.edu/research/download-enigma-gwas-results/)[58], ICV (https://enigma.ini.usc.edu/research/download-enigma-gwas-results/)[59], MDD (https://doi.org/10.7488/ds/2458)[67], NTSM (https://ctg.cncr.nl/software/summary_statistics/neuroticism_summary_statistics)[61], OCD (https://www.med.unc.edu/pgc/download-results/)[62], PANIC (https://www.med.unc.edu/pgc/download-results/)[63], PTSD (https://www.med.unc.edu/pgc/results-and-downloads/)[64] and SCZ (https://www.med.unc.edu/pgc/download-results/)[65]. The Developmental Brain RNA-seq and genotype dataset from Walker et al. is available at dbGAP with accession number phs001900 (ref. 22, accesible at https://www.ncbi.nlm.nih.gov/projects/gap/cgi-bin/study.cgi?study_id=phs001900.v1.p1). The subset of Adult Brain RNA-seq and genotype data from the PsychENCODE Consortium is available at https://psychencode.synapse.org/DataAccess and from AMP-AD is available at https://adknowledgeportal.synapse.org/Data%20Access (refs. 20,51). GWAS summary statistics and accession numbers to genotype and RNA-seq data are provided in Supplementary Table 10. isoTWAS models for 48 tissues from GTEx are available at https://zenodo.org/record/8047940 (ref. 86), adult brain cortex from PsychENCODE and AMP-AD are available at https://zenodo.org/record/8048198 (ref. 87), and developmental brain cortex from Walker et al. are available at https://zenodo.org/record/8048137 (ref. 88). All datasets used in this paper are listed here with no omissions.

## Code availability

isoTWAS is available as an R package at https://github.com/bhattacharya-a-bt/isotwas (ref. 47). Sample scripts for analyses are available at https://github.com/bhattacharya-a-bt/isotwas_manu_scripts (ref. 114). All relevant codes used in this paper are listed here and deposited online with no omissions or restrictions to access.

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

## Acknowledgements

We thank Kangcheng Hou, Tommer Schwarz, Vidhya Venkateswaran, Pan Zhang, Leanna Hernandez, Nathan LaPierre, Harold Pimentel, Mike Love and Achal Patel for engaging discussion during the research process. We thank Kanishka Patel for her aesthetic advice for figures. We thank the Psychiatric Genomics Consortium and Complex Trait Genomics Lab for their publicly available GWAS summary statistics. This work was supported by National Institutes of Health awards R01 HG009120, R01 MH115676, R01 CA251555, R01 AI153827, R01 HG006399, R01 CA244670 and U01 HG011715 (B.P.), as well as SFARI Bridge to Independence Award, NIMH R01-MH121521, NIMH R01-MH123922 and NICHD-P50-HD103557 (M.J.G.). The funders had no role in study design, data collection and analysis, decision to publish or preparation of the manuscript.

## Author contributions

A.B., B.P. and M.J.G. conceptualized the project. A.B., J.H., B.P. and M.J.G. developed the methodology. A.B. and D.D.V. wrote the software. A.B., C.J. and D.D.V. validated results. A.B., M.K., C.W., C.J., D.D.V. and J.H. contributed to formal analysis. A.B. and M.J.G. contributed to investigation. B.P. and M.J.G. provided resources. A.B., M.K., C.W., C.J., M.J.G. and DDV curated data. A.B. and M.J.G. wrote the original draft. All authors reviewed and edited the paper. A.B. visualized results. A.B., B.P. and M.J.G. supervised the project. B.P. and M.J.G. administered the project. B.P. and M.J.G. acquired funds for the project.

## Competing interests

The authors declare no competing interests.

## Additional information

**Extended data** is available for this paper at

**Supplementary information** The online version contains supplementary
material available at https://doi.org/10.1038/s41588-023-01560-2.

**Correspondence and requests for materials** should be addressed to
Arjun Bhattacharya or Michael J. Gandal.

**Peer review information** *Nature Genetics* thanks Eric Gamazon
and Pejman Mohammadi for their contribution to the peer review of
this work.

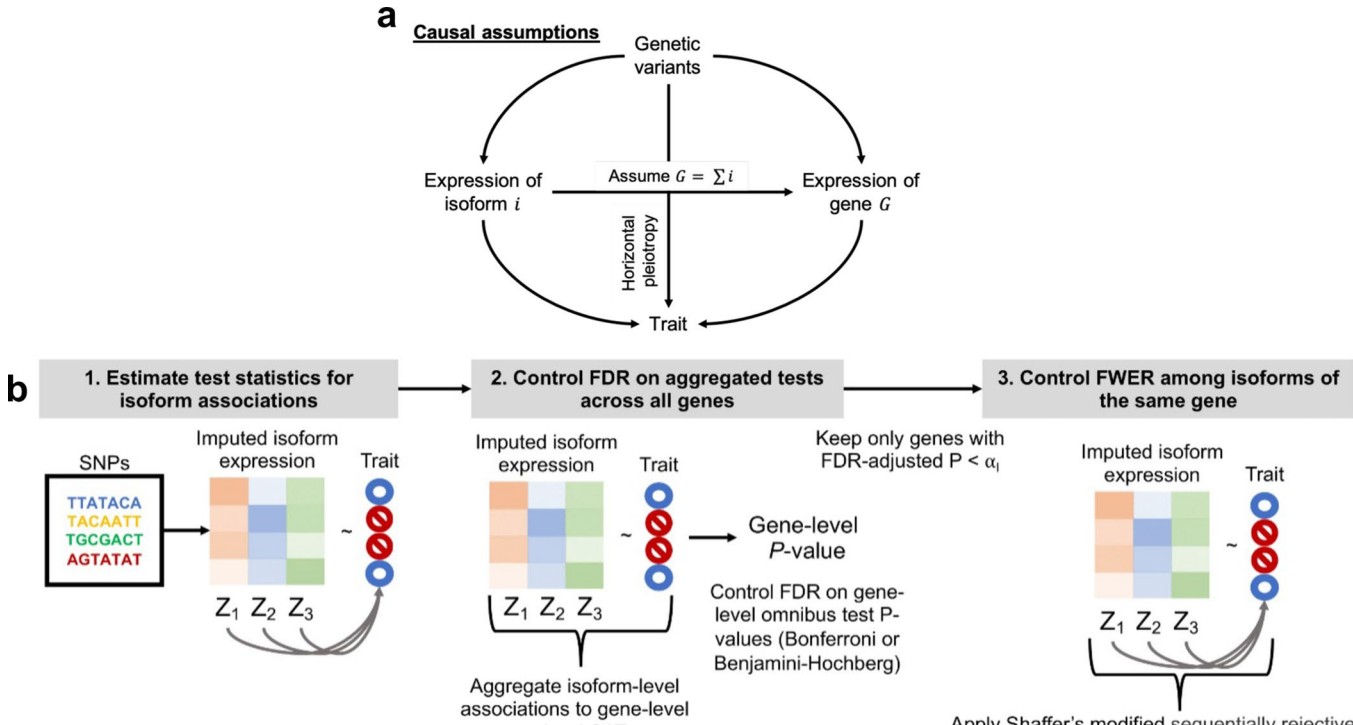

**Extended Data Fig. 1 | isoTWAS framework assumptions and testing framework. (a)** Directed acyclic graph (DAG) illustrating causal assumptions in isoTWAS: the local genetic variants within 1 Megabase of a gene have direct effects on the expression of a gene G and its isoforms; these genetic effects need not be shared across isoforms and the gene. Further, the abundance of a gene is the sum of abundances of its isoforms. Lastly, the isoform and gene need not affect the complex trait through the same path. Genetic variants may have effects on the trait through pathways independent of gene and isoform expression.

**(b)** Step-wise hypothesis testing in isoTWAS. First, isoform-trait associations are estimated Then, associations for isoforms are aggregated to the gene-level using the Aggregated Cauchy Association Test (ACAT). These aggregated gene-level associations are adjusted for multiple testing burden to control the false discovery rate (FDR). Lastly, for isoforms of genes that pass gene-level testing, we control the family-wide error rate (FWER) using Shaffer's modified sequentially rejective procedure.

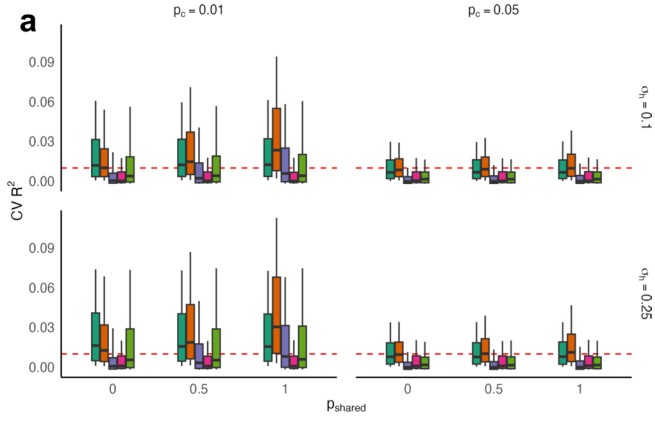

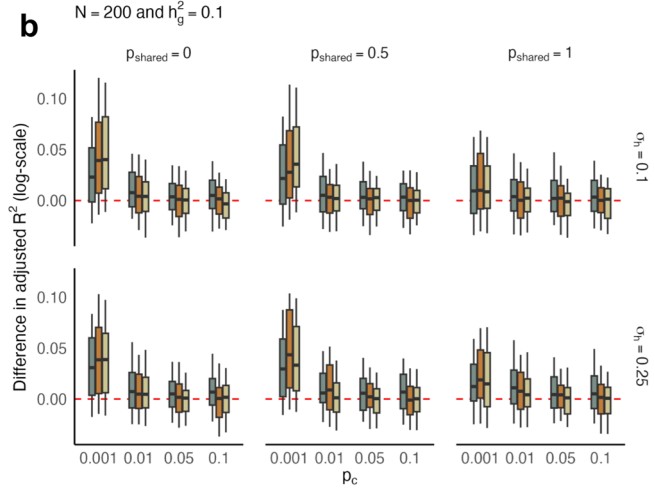

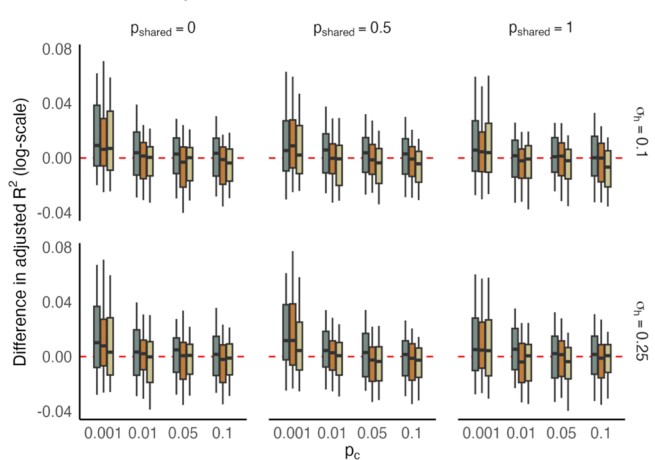

**Extended Data Fig. 2 | Prediction comparison in simulation. (a)** Boxplots of adjusted $R^2$ of prediction of isoform expression (Y axis) across shared isoQTL proportion (X axis), for 5 isoforms with isoform heritability ($h_i^2$) set to 0.05 or 0.10 (n = 1,000 independent simulations). **(b)** Boxplots of percent difference in adjusted $R^2$ in predicting gene expression between isoTWAS and TWAS models from simulations with sample size 200 (compared with sample size 500 in Fig. 2), where isoform and gene expression heritability are set to (top) 0.05 and (bottom) 0.10. For (a-b), all boxplots represent the median, 25% and 75% quantiles, and whiskers correspond to the 10% and 90% quantiles.

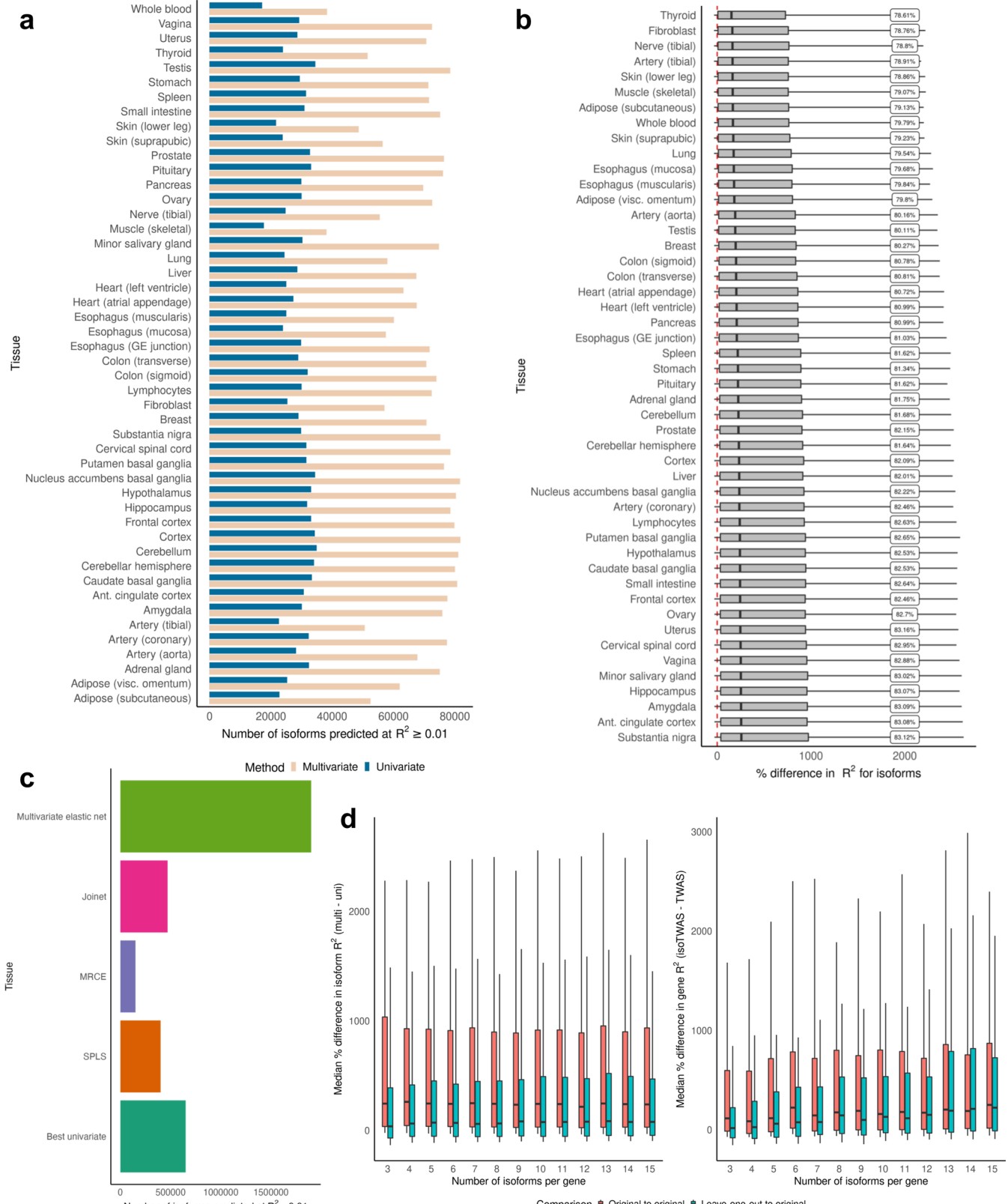

**Extended Data Fig. 3 | Isoform prediction comparison across 48 GTEx tissues.**
**(a)** Number of multivariate (cream) and univariate (blue) models predicting isoform expression at CV R² > 0.01 (X axis). **(b)** Percent difference in CV R² (X axis) of prediction of isoform expression models using multivariate models versus univariate models. The label shows the proportion of isoforms with improved performance using multivariate models (n = 139-803 biologically independent sample, see Supplementary Table 1). **(c)** Number of isoforms with CV R² > 0.01 (Y axis) using the baseline univariate model (teal, best univariate) and 4 multivariate

models. **(d)** On left, median percent difference in R² of predicting original isoform expression using multivariate versus univariate models (left) and gene expression using isoTWAS versus TWAS models (right) across increasing number of isoforms per gene, colored by models trained in the original dataset and the leave-one-out dataset (n = 139–255, see Supplementary Table 1). For (b,d), all boxplots represent the median, 25% and 75% quantiles, and whiskers correspond to the 10% and 90% quantiles.

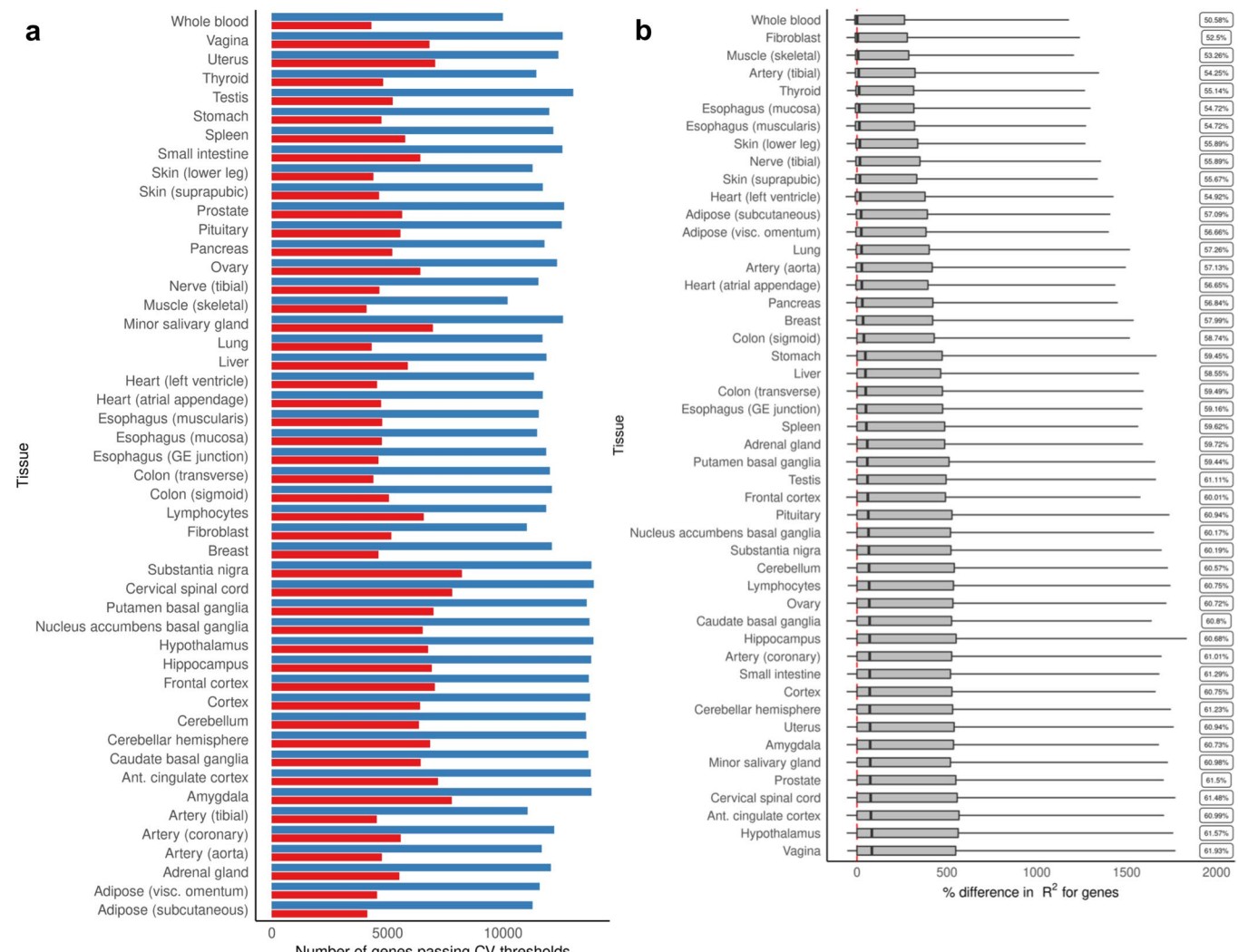

**Extended Data Fig. 4 | isoTWAS inclusion criterion and performance gains across 48 GTEx tissues. (a)** Number of genes that pass TWAS (blue) and isoTWAS (red) CV $R^2$ cutoffs to be available for testing in the trait-mapping step (X axis) **(b)** Percent difference in CV $R^2$ (X axis) of prediction of isoform expression models using multivariate models versus univariate models. The label shows the proportion of isoforms with improved performance using multivariate models. All boxplots represent the median, 25% and 75% quantiles, and whiskers correspond to the 10% and 90% quantiles.

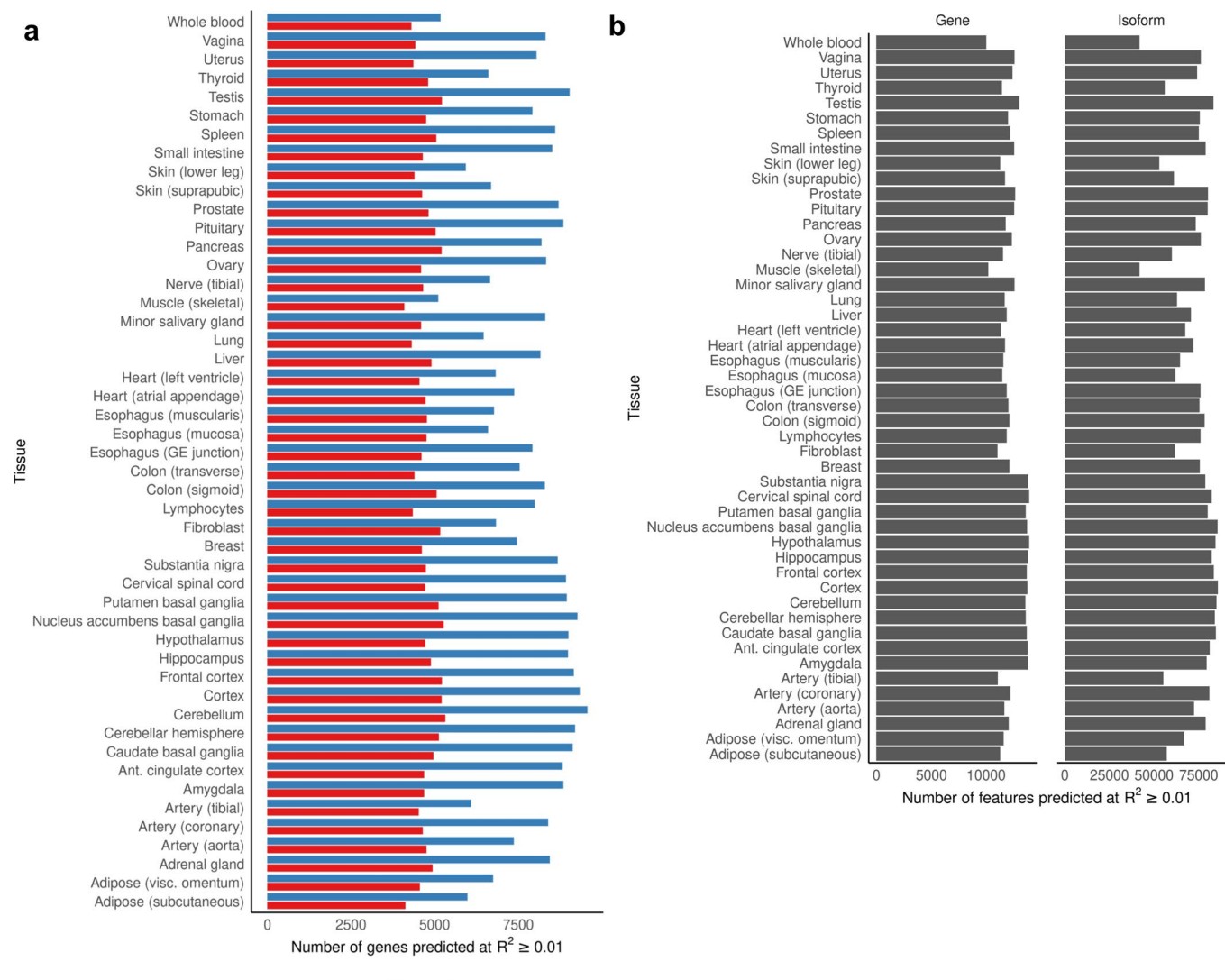

**Extended Data Fig. 5 | Gene prediction comparison across 48 GTEx tissues. (a)** Number of genes predicted at CV $R^2$ > 0.01 using TWAS (blue) and isoTWAS (red). **(b)** Number of genes (left) and isoforms (right) predicted at CV $R^2$ > 0.01 using isoTWAS across 48 GTEx tissues.

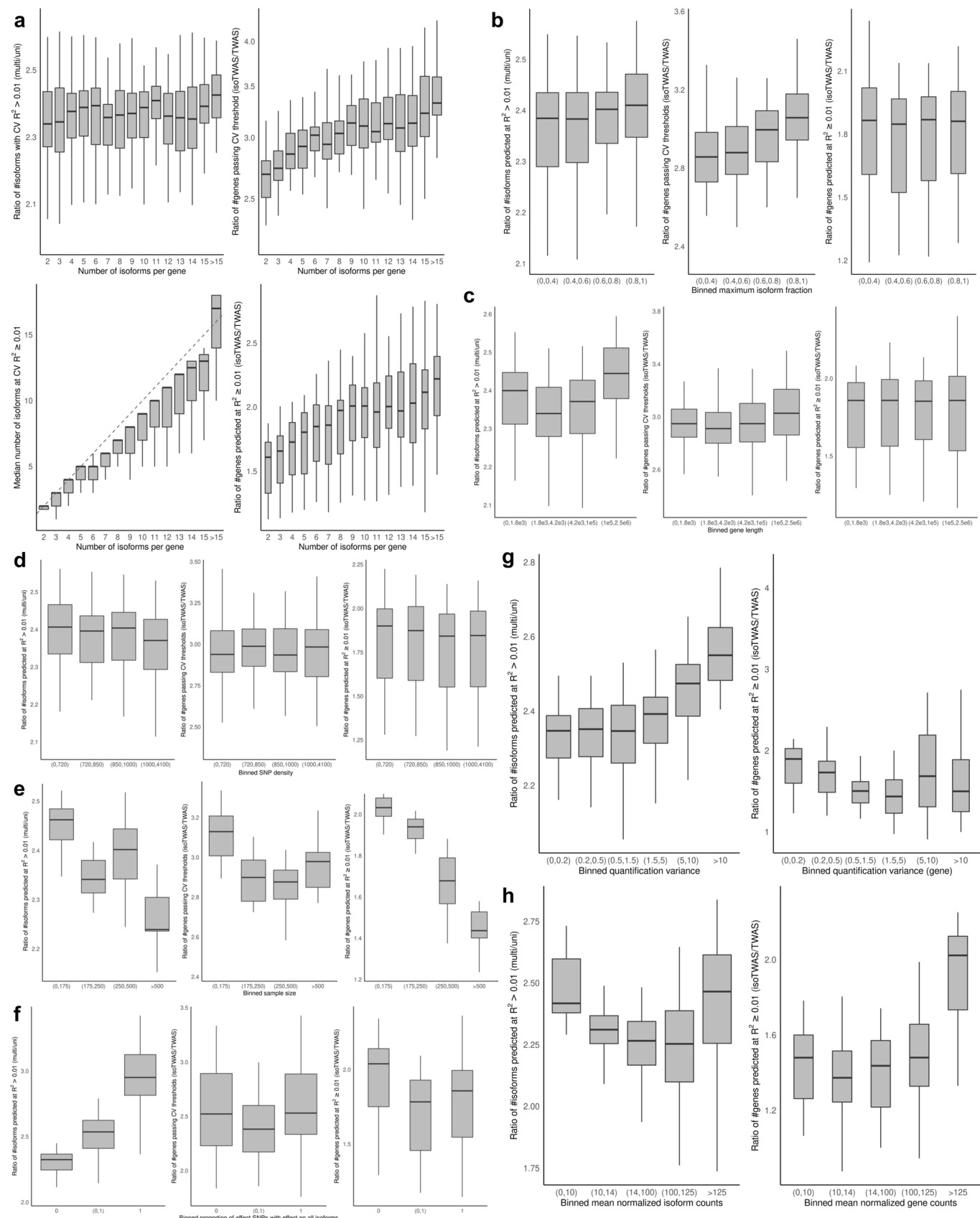

**Extended Data Fig. 6 | See next page for caption.**

**Extended Data Fig. 6 | isoTWAS performance across multiple factors. (a)** (top left) Ratio of number of isoforms predicted at $R^2 > 0.01$ using multivariate versus univariate prediction. (top right) Ratio of number of genes passing CV threshold using isoTWAS versus TWAS. (bottom left) Median number of isoforms predicted at CV $R^2 > 0.01$ in isoTWAS models across increasing number of isoforms per gene. The red line shows the line $Y = X + 1$. (bottom right) Ratio of number of genes with CV $R^2 > 0.01$ using isoTWAS versus TWAS. **(b-f)** Across bins for maximum isoform fraction (b), gene length (c), SNP density (d), sample size (e), proportion of shared isoTWAS model effect SNPs (f), (left) ratio of number of isoforms predicted at $R^2 > 0.01$ using multivariate versus univariate prediction, (middle) ratio of number of genes passing CV threshold using isoTWAS versus TWAS, and (right) ratio of number of genes with CV $R^2 > 0.01$ using isoTWAS versus TWAS. **(g-h)** Across bins for mean counts (g) and quantification variance (h) of isoforms and genes, (left) ratio of number of isoforms predicted at $R^2 > 0.01$ using multivariate versus univariate prediction and (right) Ratio of number of genes with CV $R^2 > 0.01$ using isoTWAS versus TWAS. For (a-h), all boxplots represent the median, 25% and 75% quantiles, and whiskers correspond to the 10% and 90% quantiles.

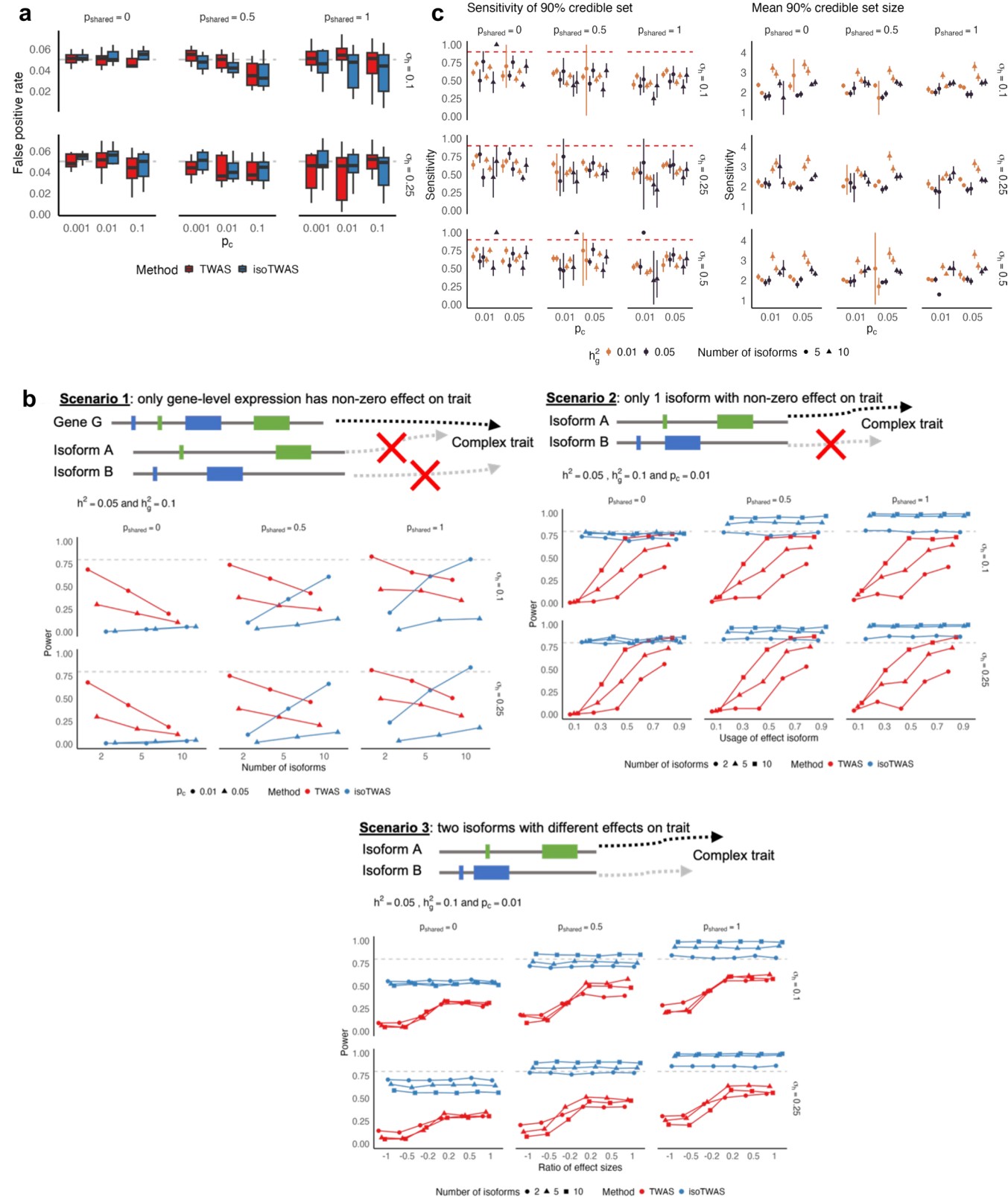

**Extended Data Fig. 7 | See next page for caption.**

**Extended Data Fig. 7 | Power comparison in simulation. (a)** Across 20 iterations of 1,000 simulations, boxplots of false positive rate to detect a gene-trait association using Cauchy-aggregated P values of isoform-trait associations (red) and gene-level TWAS (blue) from weighted burden tests. We calculate the false positive rate as the proportion of the 1,000 tests that give P > 0.05. All boxplots represent the median, 25% and 75% quantiles, and whiskers correspond to the 10% and 90% quantiles. **(b)** (Scenario 1) Power to detect gene-trait association (proportion of tests with $P < 2.5 \times 10^{-6}$ using weighted burden test, Y axis) across number of total isoforms per gene (X axis). Points are shaped by causal isoQTL proportion and colored by method. (Scenario 2) Power to detect gene-trait association across proportion of gene expression explained by effect isoform (X axis). (Scenario 3) Power to detect gene-trait association across ratio of effect sizes of 2 effect isoforms (X axis). All plots for (1–3) are facetted by proportion of shared isoQTLs (top margin) and proportion of expression heritability attributed to shared non-genetic effects across isoforms (right margin). For (2–3), points are shaped by number of isoforms per gene and colored by method. Here, expression heritability is set of 0.05, trait heritability is set to 0.1, and causal proportion of Scenarios 2–3 is set of 0.01. **(c)** Sensitivity and mean set size of 90% credible set using FOCUS to finemap isoform-trait associations for a single gene, across causal isoQTL proportion (X axis). Points are colored by trait heritability and shaped by the number of isoforms per gene. Plots are facetted by proportion of shared isoQTLs (top margin) and proportion of expression heritability attributed to shared non-genetic effects across isoforms (right margin). Line-ranges in show a 95% jackknife confidence interval (n = 1,000 independent simulations).

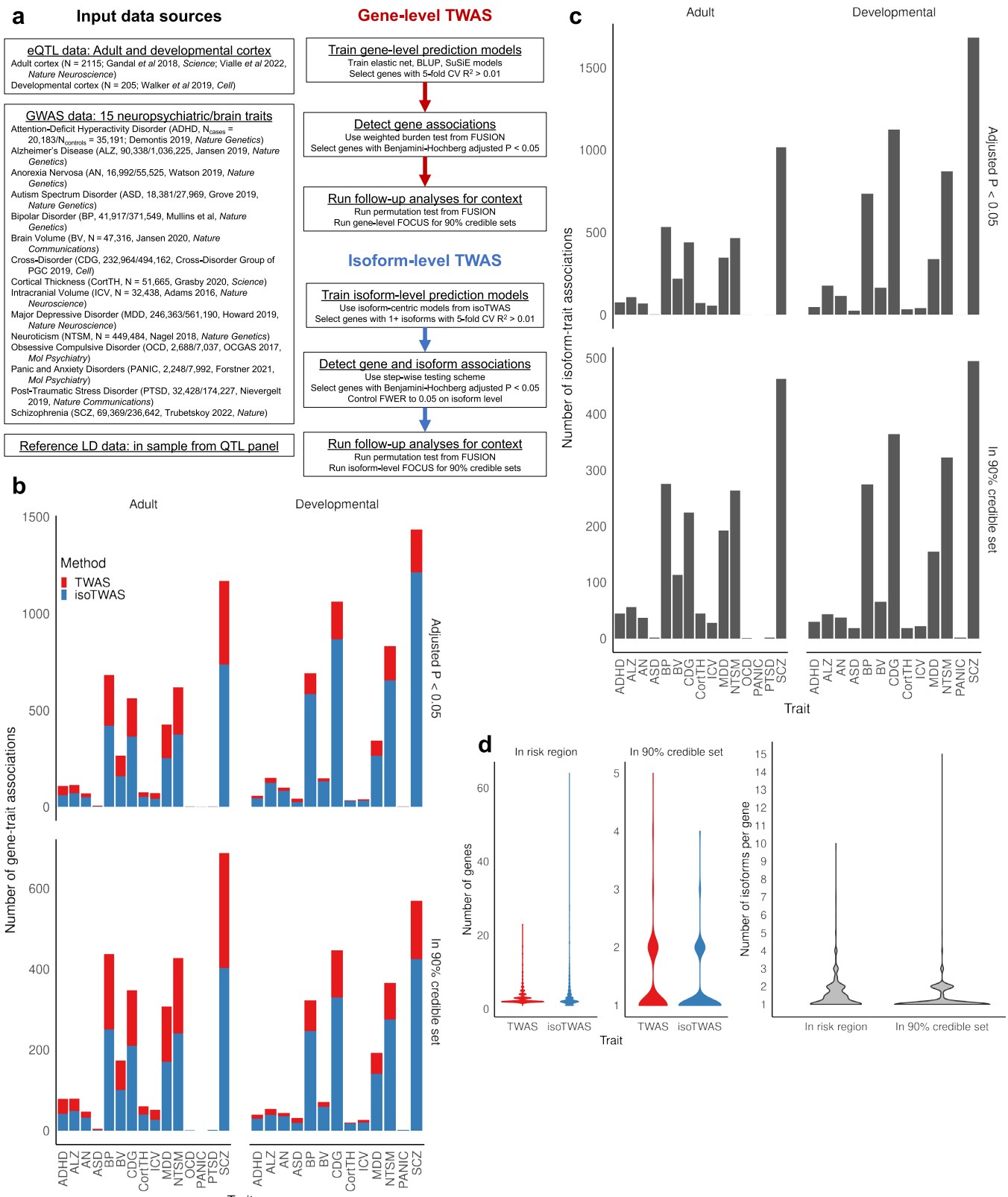

**Extended Data Fig. 8 | Discovery comparison in public data. (a)** Data sources for eQTL reference data, GWAS cohorts, and reference LD data are provided on the left (black). The full gene-level TWAS (red) and isoTWAS (blue) are summarized on the right. **(b)** Number of gene-trait associations (Y axis) using TWAS (red) and isoTWAS (blue) across trait (X axis), faceting by tissue (top margin) and threshold (right margin: adjusted weighted burden test P < 0.05 and permutation test P < 0.05, top; in 90% credible set using FOCUS fine-mapping, bottom). **(c)** Number of isoform-trait associations (Y axis) using isoTWAS across trait (X axis), faceting by tissue (top margin) and threshold (right margin: adjusted weighted burden test P < 0.05 and permutation test P < 0.05, top; in 90% credible set using FOCUS fine-mapping, bottom). **(d)** Distribution of number of genes (left) and isoforms (right) in risk region and in 90% credible set using TWAS and isoTWAS.

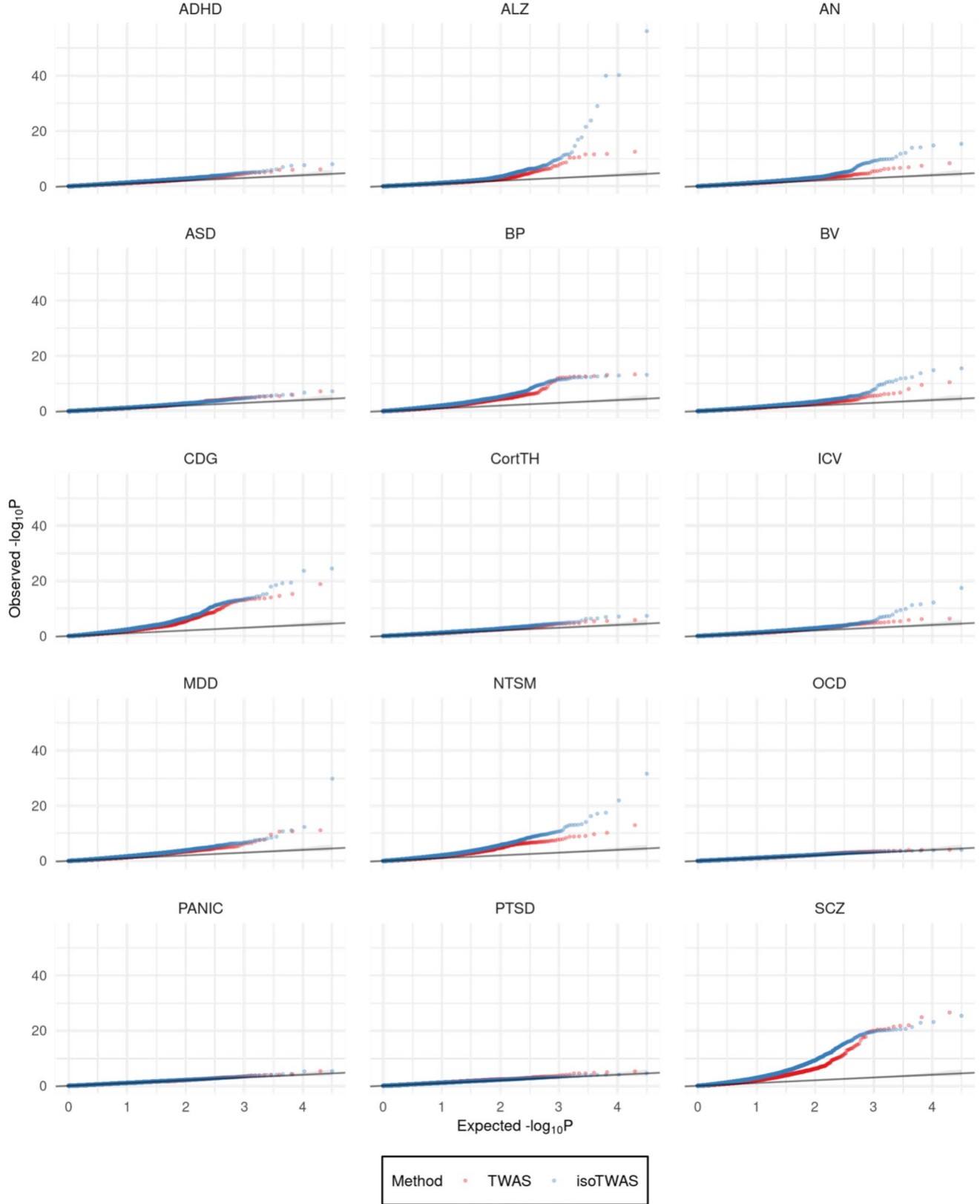

**Extended Data Fig. 9 | Test statistic inflation comparison in public data.** QQ-plots of gene-level P values using TWAS (red) and isoTWAS (blue) across 15 traits.

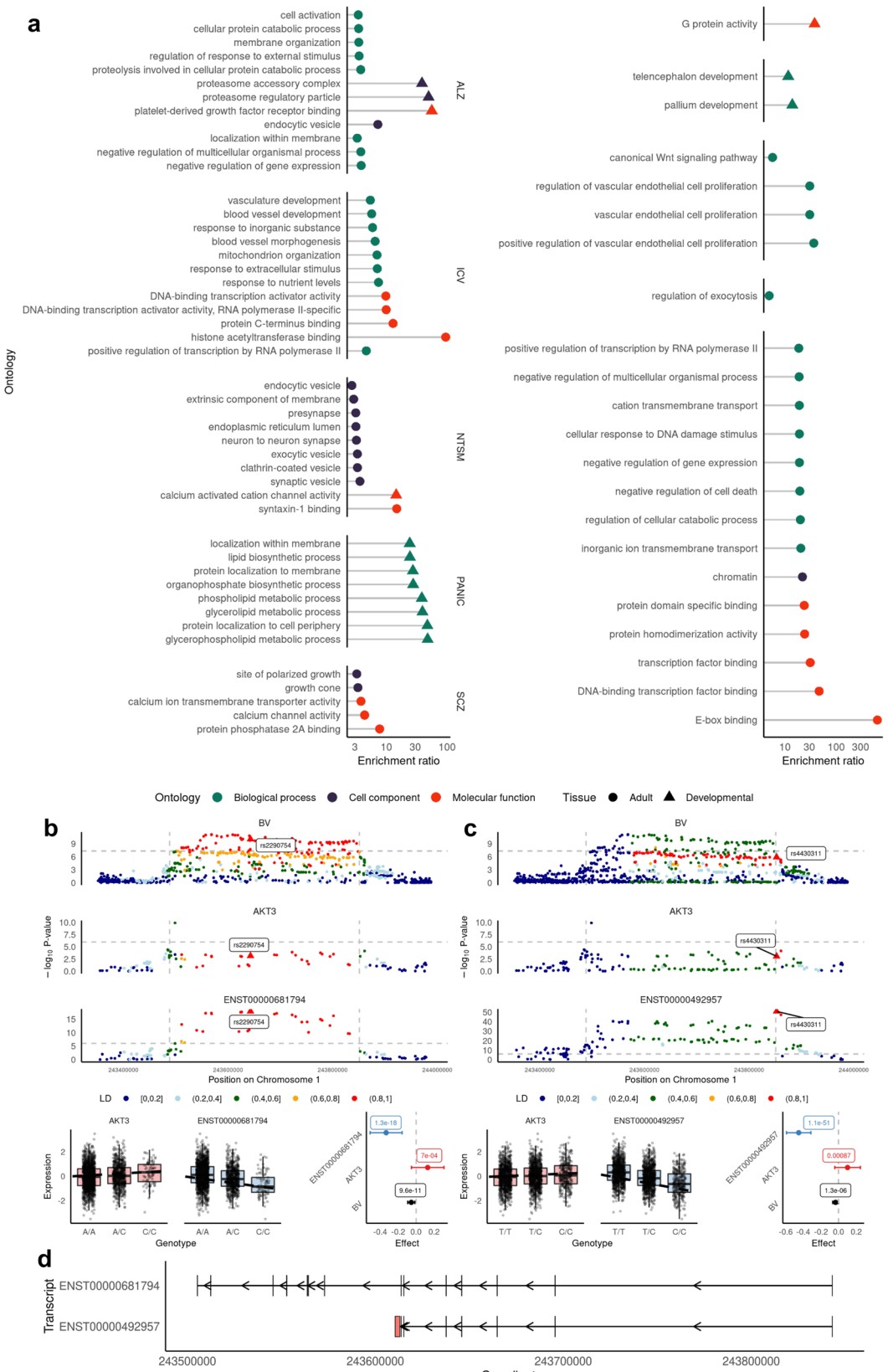

**Extended Data Fig. 10 | See next page for caption.**

**Extended Data Fig. 10 | Biological relevance of gene-trait associations detected by isoTWAS. (a)** Lollipop plot of enrichment ratio (X axis) of ontologies (Y axis) for isoTWAS-prioritized genes associated at adjusted weighted burden test P < 0.05 and permutation test P < 0.05. Points are shaped by tissue type (adult or developmental) and colored by ontology type (biological process, cell component, molecular function). **(b-d)** For ENST00000681794 association with BV **(b)** and ENST00000492957 with BV **(c)**, Manhattan plots of GWAS, eQTL, and isoQTL signal colored by LD (top), boxplots of gene (red) and isoform (blue) expression (Y axis) by genotype (X axis) (bottom left), and forest plot of lead isoQTL effect size using two-sided Wald-type t-test from linear regression and 95% confidence interval with isoform (blue), gene (red), and trait (black) (bottom right, n = 2,115 biologically independent samples). Vertical gray lines indicate the transcription start and end sites for each gene, and the horizontal gray line indicates $P = 5 \times 10^{-8}$ for GWAS and $10^{-6}$ for QTLs. All boxplots represent the median, 25% and 75% quantiles, and whiskers correspond to the 10% and 90% quantiles. **(d)** Comparison of exon and intron structure of ENST00000681794 and ENST00000492957, based on Gencode v38 reference.

# Reporting Summary

## Statistics

For all statistical analyses, confirm that the following items are present in the figure legend, table legend, main text, or Methods section.

| n/a | Confirmed | |
|---|---|---|
| ☐ | ☒ | The exact sample size ($n$) for each experimental group/condition, given as a discrete number and unit of measurement |
| ☐ | ☒ | A statement on whether measurements were taken from distinct samples or whether the same sample was measured repeatedly |
| ☐ | ☒ | The statistical test(s) used AND whether they are one- or two-sided *Only common tests should be described solely by name; describe more complex techniques in the Methods section.* |
| ☐ | ☒ | A description of all covariates tested |
| ☐ | ☒ | A description of any assumptions or corrections, such as tests of normality and adjustment for multiple comparisons |
| ☐ | ☒ | A full description of the statistical parameters including central tendency (e.g. means) or other basic estimates (e.g. regression coefficient) AND variation (e.g. standard deviation) or associated estimates of uncertainty (e.g. confidence intervals) |
| ☐ | ☒ | For null hypothesis testing, the test statistic (e.g. $F$, $t$, $r$) with confidence intervals, effect sizes, degrees of freedom and $P$ value noted *Give P values as exact values whenever suitable.* |
| ☐ | ☒ | For Bayesian analysis, information on the choice of priors and Markov chain Monte Carlo settings |
| ☐ | ☒ | For hierarchical and complex designs, identification of the appropriate level for tests and full reporting of outcomes |
| ☐ | ☒ | Estimates of effect sizes (e.g. Cohen's $d$, Pearson's $r$), indicating how they were calculated |

*Our web collection on statistics for biologists contains articles on many of the points above.*

## Software and code

Policy information about availability of computer code

| Data collection | No software was used for data collected. |
|---|---|
| Data analysis | The following software was used: Salmon v1.5.2, Salmon v1.8.0, tximeta v1.16.1, DESeq2 v1.38.3, fishpond v2.4.1, IsoformSwitchAnalyzeR v1.20.0, edgeR v3.40.2, minimac4, eagle v2.4, CrossMap.v.0.6.3, PLINK v1.90b6.21, bcftools v1.11, samtools v1.14, PicardTools v2.25.0, QTLtools v1.3.1. isoTWAS is available as an R package at https://github.com/bhattacharya-a-bt/isotwas (isoTWAS v1.0.0). Sample scripts for analyses are available at https://github.com/bhattacharya-a-bt/isotwas_manu_scripts. |

For manuscripts utilizing custom algorithms or software that are central to the research but not yet described in published literature, software must be made available to editors and reviewers. We strongly encourage code deposition in a community repository (e.g. GitHub). See the Nature Portfolio guidelines for submitting code & software for further information.

## Data

Policy information about availability of data

All manuscripts must include a data availability statement. This statement should provide the following information, where applicable:

- Accession codes, unique identifiers, or web links for publicly available datasets
- A description of any restrictions on data availability
- For clinical datasets or third party data, please ensure that the statement adheres to our policy

GTEx genetic, transcriptomic, and covariate data were obtained through dbGAP approval at accession number phs000424.v8.p2. Linkage disequilibrium reference

## Research involving human participants, their data, or biological material

Policy information about studies with human participants or human data. See also policy information about sex, gender (identity/presentation), and sexual orientation and race, ethnicity and racism.

| | |
|---|---|
| Reporting on sex and gender | Only publicly available data was used, as linked in the Data Availability statement. No new data was collected in this study. |
| Reporting on race, ethnicity, or other socially relevant groupings | N/A |
| Population characteristics | N/A |
| Recruitment | N/A |
| Ethics oversight | N/A |

Note that full information on the approval of the study protocol must also be provided in the manuscript.

# Field-specific reporting

Please select the one below that is the best fit for your research. If you are not sure, read the appropriate sections before making your selection.

☒ Life sciences   ☐ Behavioural & social sciences   ☐ Ecological, evolutionary & environmental sciences

For a reference copy of the document with all sections, see nature.com/documents/nr-reporting-summary-flat.pdf

# Life sciences study design

All studies must disclose on these points even when the disclosure is negative.

| | |
|---|---|
| Sample size | This study includes an integrative data analysis of data from the Genotype Tissue-Expression Project. Samples sizes for this dataset are reported in Aguet et al 2020, Science and in Supplementary Table S1. We also include data from the PsychENCODE Consortium and AMP-AD Consortium using prefrontal cortex RNA-seq and genotype data for two samples: adult (N = 2,115) and developmental frontal cortex (N = 205). Maximal sample sizes depended on the number of samples with both RNA-seq and genotype data; no power calculations were conducted to pre-determine sample size. We conducted gene- and isoform-level trait mapping for 15 neuropsychiatric traits: attention-deficit hyperactivity disorder (ADHD, Ncases = 20,183/Ncontrols = 35,191), Alzheimer's disease (ALZ, 90,338/1,036,225), anorexia nervosa (AN, 16,992/55,525), autism spectrum disorder (ASD, 18,381/27,969), bipolar disorder (BP, 41,917/371,549), brain volume (BV, N=47,316), cross-disorder (CDG, 232,964/494,162), cortical thickness (CortTH, N = 51,665), intracranial volume (ICV, N = 32,438), major depressive disorder (MDD, 246,363/561,190), neuroticism (NTSM, N = 449,484), obsessive compulsive disorder (OCD, 2,688/7,037), panic and anxiety disorders (PANIC, 2,248/7,992), post-traumatic stress disorder (PTSD, 32,428/174,227), and schizophrenia (SCZ, 69,369/236,642). These 15 traits represent complex brain-related traits that are studied by large consortium with sufficient sample sizes (Psychiatric Genomics Consortium, Complex Traits Genomics Lab, etc). These 15 traits represent a wide range of brain-related disorders and traits that provide a comprehensive analysis of the utility of isoform-level TWAS trait mapping. |
| Data exclusions | GTEx data was restricted to individuals of European genetic ancestry to ensure portability of genetic predictions. PsychENCODE and AMP-AD individuals were removed if their WGCNA network connectivity scores based on isoform-level expression was less than -3; these low scores indicate that these samples may be plagued by technical biases that may affect the estimation of genetic effects on gene- and isoform-level expression. These data exclusion factors were pre-determined. |
| Replication | There are no experimental findings to replicate. |
| Randomization | There are no experimental groups in this study. |
| Blinding | No data collection was conducted in this study. The data used in this study has been previously reported. As such, the investigators of this work were blinded to group allocation. |

# Reporting for specific materials, systems and methods

We require information from authors about some types of materials, experimental systems and methods used in many studies. Here, indicate whether each material, system or method listed is relevant to your study. If you are not sure if a list item applies to your research, read the appropriate section before selecting a response.

## Materials & experimental systems

| n/a | Involved in the study |
|-----|----------------------|
| ☒ ☐ | Antibodies |
| ☒ ☐ | Eukaryotic cell lines |
| ☒ ☐ | Palaeontology and archaeology |
| ☒ ☐ | Animals and other organisms |
| ☒ ☐ | Clinical data |
| ☒ ☐ | Dual use research of concern |
| ☒ ☐ | Plants |

## Methods

| n/a | Involved in the study |
|-----|----------------------|
| ☒ ☐ | ChIP-seq |
| ☒ ☐ | Flow cytometry |
| ☒ ☐ | MRI-based neuroimaging |

