## [Peer Review File · Nature Genetics]

Peer Review Information

Manuscript Title: Isoform-level transcriptome-wide association uncovers novel genetic risk mechanisms for neuropsychiatric disorders in the human brain

Corresponding author name(s): Dr Arjun Bhattacharya

Reviewer Comments & Decisions:

Decision Letter, initial version:
--

29th Nov 2022

Dear Dr Bhattacharya,

Your Article, "Isoform-level transcriptome-wide association uncovers extensive novel genetic risk mechanisms for neuropsychiatric disorders in the human brain" has now been seen by 2 referees. Please note that we did secure a further reviewer (#2), but they have not submitted their report; given the prolonged review time and the feedback supplied, we concluded there was enough for a decision at this point.

You will see from their comments copied below that while they find your work of considerable potential interest, they have raised quite substantial concerns that must be addressed. In light of these comments, we cannot accept the manuscript for publication, but would be very interested in considering a revised version that addresses these serious concerns.

Briefly, both reviewers acknowledge the potential interest in and utility of your iso-TWAS approach, but have substantial criticisms.

Reviewer #1 is unconvinced as to whether your "methodology improves on existing approaches". That said, they also make a number of comments with clear guidance for improvement that would help to address this high-level issue.

Reviewer #2 sounds more positive but, at the same time, somewhat underwhelmed ("moderately enthusiastic"). Again, they make a number of specific requests for improvement.

We think that Referee #1's comments on the technical basis of iso-TWAS are important and must be fully addressed - it seems unlikely, in our reading, that this reviewer will be supportive without doing so. Additionally, we would strongly encourage you to take Reviewer #2's comments on utility/biological insight to heart and agree that these aspects could be improved, such that the impact

of iso-TWAS is clearer to the reader.

We hope you will find the referees' comments useful as you decide how to proceed. If you wish to submit a substantially revised manuscript, please bear in mind that we will be reluctant to approach the referees again in the absence of major revisions.

To guide the scope of the revisions, the editors discuss the referee reports in detail within the team, including with the chief editor, with a view to identifying key priorities that should be addressed in revision and sometimes overruling referee requests that are deemed beyond the scope of the current study. We hope that you will find the prioritised set of referee points to be useful when revising your study. Please do not hesitate to get in touch if you would like to discuss these issues further.

If you choose to revise your manuscript taking into account all reviewer and editor comments, please highlight all changes in the manuscript text file. At this stage we will need you to upload a copy of the manuscript in MS Word .docx or similar editable format.

*2) If you have not done so already please begin to revise your manuscript so that it conforms to our Article format instructions, available [here](http://www.nature.com/ng/authors/article_types/index.html). Refer also to any guidelines provided in this letter.

[redacted]

If you wish to submit a suitably revised manuscript we would hope to receive it within 6 months. If you cannot send it within this time, please let us know. We will be happy to consider your revision so long as nothing similar has been accepted for publication at Nature Genetics or published elsewhere. Should your manuscript be substantially delayed without notifying us in advance and your article is eventually published, the received date would be that of the revised, not the original, version.

Thank you for the opportunity to review your work.

Sincerely,

Michael Fletcher, PhD
Senior Editor, Nature Genetics

ORCID: 0000-0003-1589-7087

Referee expertise:

Referee #1: statistical genetics, TWAS.

Referee #3: computational biology/genomics, machine learning, transcriptome analysis.

Reviewers' Comments:

Reviewer #1:

Remarks to the Author:

The study presents isoTWAS, a multivariate framework for integration of genetic variation, isoform-level expression, and trait association. The study addresses an "unmet need," namely, how to optimally use the transcriptome towards discovery of trait associations, with a particular focus on brain-related traits. The framework is potentially useful and the publicly available models can enhance the study's utility. However, there are a number of major gaps in the methodology, analysis, and presentation, which should be addressed. Overall, these gaps raise crucial questions as to whether the proposed methodology improves on existing approaches.

1. The authors report that the isoform-centric TWAS framework, at the imputation stage, is uncovering heritable variation for many more genes than the conventional TWAS framework. For which genes? Genes with primarily low expression? High expression? Genes with short transcripts? Genes with higher density of SNPs? Genes with putatively heritable variation identified through isoform-centric TWAS should be characterized to rule out the possibility of confounding. (See more below.)

2. I'm also concerned about the authors' approach for heritability estimation of gene expression, for example, to filter out genes (using GCTA-REML; lines 530-531). Gene expression, in contrast to complex traits (which are driven by a large number of genetic variants, each with modest effect size, in a polygenic architecture), is driven by a generally smaller number of variants with large effects. The modeling assumptions in the heritability estimation as applied to gene or isoform expression should therefore depend on the specific genetic architecture of the molecular trait and may substantially alter the conclusions from the presented analyses. It's not clear how similar or different the genetic architecture is for the two types of expression traits.

3. Are the genes in #1 enriched for genes with a large number of "detected" isoforms? Surely, if a gene consists of a single isoform, there should be no difference between the isoform-centric and conventional TWAS framework. It would appear then that the successful "detection" of these isoforms is a key component of the theoretical/empirical justification for the framework. What is the impact on performance when an isoform fails to be detected? This should be quantified.

4. The isoform quantification accuracy's impact on downstream analyses must be investigated and quantified. The isoform-level quantification is typically much more difficult than gene-expression quantification.

5. Gene complexity can have a substantial impact on the conclusions that can be reliably drawn. One scenario is a gene with a large number of very similar isoforms. Here, isoform quantification will be extremely difficult and may suffer from low accuracy since mapping the reads to the correct isoform source will be extremely challenging. The case of a gene with a large number of isoforms also presents its own challenges and may suffer from low quantification accuracy. These scenarios can impact the imputation of isoform expression (ie., garbage in, garbage out). (Note that the authors' simulation framework leverages empirical data from only 2 genes [lines 581-588] and appears to be overly simplistic.)

6. An isoform-centric framework comes with its own challenges. Since, as the authors point out, the isoforms for a gene can be highly correlated, the question of the causal isoform for a trait is of critical importance. This problem is very similar to the problem of proximal, correlated genes within a locus making it difficult to identify the causal gene in the locus. It's not clear that the isoform-centric analysis necessarily leads to improved causal discovery rate.

7. The highly powered schizophrenia GWAS (relatively to the less powered GWASs for the other neuropsychiatric traits) appears to show a greater shift towards low p-values from conventional TWAS than from the isoform-centric TWAS (Figure S13 Q-Q plots). This would imply a substantially improved ability to detect gene-trait associations from the conventional TWAS, contradicting one of the major conclusions of the study.

8. The scenario of different genetic effects on phenotype across isoforms is an interesting one, but the

study has not adequately investigated how widespread this phenomenon is. Thus, it's also not clear how informative the study's simulation framework is.

Minor

1. Figure legends for figure 2c and 2d should be switched.

Reviewer #3:

Remarks to the Author:

Summary of the key results: introducing isoform level TWAS and applies to two major datasets.

Originality and significance: Novel, and timely. The significance of the results in practice seems to be moderate or not shown convincingly.

Data & methodology: The analysis is done well.

Appropriate use of statistics and treatment of uncertainties: The statistics are done well.

Conclusions: conclusions are valid but oversold.

Suggested improvements: See below.

References: the authors appropriately credit previous work

Clarity and context: See below.

The study presents iso-TWAS, an isoform level extension of TWAS. Expanding TWAS to isoform level analysis is a natural generalization of the method and is timely. I think the comparison of several multivariate methods for doing isoform-level TWAS is impressive. Overall, I am excited about the topic, and even though I'm only moderately enthusiastic about it (see below), I would be happy to see this timely work published as a first-in-class method for isoform-level TWAS.

The methods seem to be done generally well, but the overall added value of the analysis appears to be moderate in practice or at least not shown convincingly. From the biology standpoint, I find the results limited and the follow-up rather superficial, not living up to the paper's title. As for the presentation, the text needs 1) reorganization to improve the logical flow between simulation and real data, 2) polishing to improve the methods' clarity, and 3) toning down throughout to represent the impact more realistically.

Below are my remaining specific comments:

– This statement is confusing as the expression of the gene is generally the sum of length normalized isoform expressions like TPM:

Line 76- "In addition, we assume that the abundance of a gene (total gene expression) is the sum of the abundance of its isoforms (isoform expression) (Supplemental Figure S1), on the raw count scale."

Please clarify/justify.

- Edit text: "multivariate data object" is programming lingo.
- Figure 2 labels are mixed up, and it has typos (heritable).
- For figure 2 and other performance reports in the paper, what is the data set used? Considering that the genes are selected by different heritability tests this affects the stats reported. To me, a reasonable set could be either all genes, or all protein-coding genes, or genes that are shared between the two methods, or genes that are the union of the two methods.
- Standard univariate Susie the model outperforms previous regression models used for TWAS (elastic net and bsImm) and is widely used. Considering that the model selected here is a multivariate extension of Susie, it'd be appropriate to show that the improved accuracy is associated with isoform level modeling and not Susie; this would be a comparison between standard susie applied to -gene level- expressions vs multivariate susie applied to -isoform level- expression. Reading through the paper, it seems like one of the "multivariate" methods used is the naive use of univariate methods on all isoforms independently, which that does include three different methods, including susie. It goes without saying the reviewer is confused! If the multivariate susie was indeed compared to how susie is usually used, please revise the text to avoid confusing the reader.
- 5% nominal p-value for h2 cut-off for the best isoform in a gene seems like a non-trivial choice, especially in genes with many isoforms. Please justify or correct this choice. Would it be appropriate to use some form of multiple hypothesis correction or extreme value modeling to avoid increased false positives in genes with many isoforms? The number of TWAS testable genes (e.g. Fig S6, or paragraph starting at line 139) is directly connected to how this is done.
- From the text it appears that less than half of the genes have more than one isoforms: "on average, four GENCODE v40-annotated isoforms per gene (... , median 1)" line 75. But looking at Figure S4 (and several other stats reported in the text) the multivariate model always outperforms the univariate model in >80% of the genes. Are these results limited to genes with multiple isoforms? If so please clarify the text and report appropriately.
- It seems to me that the gains by the multivariate model are stronger in low-sample tissues and fade away in larger sampled tissues (Figs S4 and S7). This is a little unexpected to me in the case of gene-level predictions at least. As the multivariate model is more complex model than a gene-level model which should be underpowered in smaller datasets and better performing in larger datasets. Please clarify or explore this issue.
- In Fig 2C and other places in the paper, the number of heritable isoforms is compared to the number of heritable genes. This is a little misleading because handling multiple isoforms in a single gene in such an analysis would be non-trivial. For instance, if multiple isoforms in a gene are in high correlation, they'll be counted multiple times. One naive approach would be to "impute" GRex for all isoform in a gene by its gene-level prediction for the univariate model, and then use that for comparison at an isoform level. If the authors have already addressed this issue in some other way, please clarify the text to reflect it.
- Paragraph starting on line 131, is confusing and out of the flow of the paper (e.g. the reference to

comparison among different multivariate methods).

– The claim about brain-specific effects starting on line 147 is not convincing. The supporting figures Fig 2C and Fig S9 show a clear pattern for the sample size-driven effect. The authors mention a correction in the F-test but there are not enough details.

- This is minor. In figure S17-S19 it is strange that the isoform level model outperforms the gene level model in cases where the eQTL is shared across isoforms. I assume this is because the noise added to individual isoforms in the simulation is independent. Unless I'm getting this wrong, this issue limits the scope of the simulation results to comparison among different multivariate methods only.

– Considering that brain-specific genes have a tendency to be intolerant of LoF variants, the statement on line 277 "suggesting that these genes are particularly intolerant to protein-truncating variation and that isoTWAS detects biological-relevant signal due to its isoform-specific focus." should be backed up by an appropriate test.

Pejman Mohammadi

Author Rebuttal to Initial comments

Reviewer #1:

Remarks to the Author:

The study presents isoTWAS, a multivariate framework for integration of genetic variation, isoform-level expression, and trait association. The study addresses an "unmet need," namely, how to optimally use the transcriptome towards discovery of trait associations, with a particular focus on brain-related traits. The framework is potentially useful and the publicly available models can enhance the study's utility. However, there are a number of major gaps in the methodology, analysis, and presentation, which should be addressed. Overall, these gaps raise crucial questions as to whether the proposed methodology improves on existing approaches.

We thank the reviewer for recognizing the importance of the overall goal of this work, addressing a critical unmet need in the field for optimally leveraging the transcriptome for gene-trait association mapping, as well as for acknowledging the usefulness of our framework and utility of the publicly available models we have trained. We understand the reviewer raises several important questions with respect to how our methodology and analyses improve upon existing approaches, which we address in the individual responses below.

1. The authors report that the isoform-centric TWAS framework, at the imputation stage, is uncovering heritable variation for many more genes than the conventional TWAS framework. For which genes? Genes with primarily low expression? High expression? Genes with short transcripts? Genes with higher density of SNPs? Genes with putatively heritable variation identified through isoform-centric TWAS should be characterized to rule out the possibility of confounding. (See more below.)

See below.

2. I'm also concerned about the authors' approach for heritability estimation of gene expression, for example, to filter out genes (using GCTA-REML; lines 530-531). Gene expression, in contrast to complex traits (which are driven by a large number of genetic variants, each with modest effect size, in a polygenic architecture), is driven by a generally smaller number of variants with large effects. The modeling assumptions in the heritability estimation as applied to gene or isoform expression should therefore depend on the specific genetic architecture of the molecular trait and may substantially alter the conclusions from the presented analyses. It's not clear how similar or different the genetic architecture is for the two types of expression traits.

We thank the reviewer for this suggestion of important comparisons of the prediction of isoform-centric models. We address these two comments simultaneously as follows:

First, we agree that the assumptions for standard GCTA heritability estimation of gene expression may not apply directly to isoform expression. In fact, Kim et al 2022 provides an extended multivariate variance components model for heritability estimation of isoform expression (DOI: 10.1101/2022.10.18.22281204).

Second, we re-trained isoform- and gene-specific predictive models without considering the heritability cutoff. This simplifies many of the filtering assumptions and broadens our analysis to all expressed genes in a given tissue. We have also expanded our sample sizes for the adult (N = 2115) brain cortex, now combining data from PsychENCODE with that from AMP-AD to further improve the performance of our predictive modeling. In addition, we restrict our SNP predictors to those included in HapMap3, which are the most well imputed across different datasets (DOI: 10.1038/nature02168). We have conducted multiple evaluations of the difference in cross-validation predictive \$R^2\$ with respect to three primary metrics considering all genes with multiple isoforms:

1. Prediction of individual isoforms using a multivariate approach that jointly models all isoforms of a gene (multivariate elastic net, multivariate regression with covariance estimation, sparse partial least squares, and multivariate elastic net with stacked generalization) compared to univariate models of individual isoforms. Here, we consider the difference in predictive R^2 (multivariate - univariate) and the number of isoforms that reach cross-validation $R^2 > 0.01$ using either approach.
2. Number of genes that pass cross-validation prediction cutoffs using isoTWAS (a gene with >1 isoform predicted at CV $R^2 > 0.01$) and TWAS (a gene predicted itself at CV $R^2 > 0.01$). We acknowledge that this metric can be biased towards isoTWAS, since genes with multiple isoforms have more of a chance at random to meet the criterion. However, independent of comparisons between isoTWAS and TWAS, this inclusion criterion for cross-validation prediction are standard to TWAS methods.
3. Prediction of total gene expression using isoTWAS models (combining isoforms of the same gene) and gene-specific TWAS models. Here, we consider the difference in predictive R^2 (isoTWAS - TWAS) and the number of genes that reach cross-validation $R^2 > 0.01$ using each method. The purpose of isoTWAS prediction is to predict individual isoforms of genes. We consider prediction of total gene expression as a corollary measure of how quantification error impacts prediction using isoform-specific models.

In general, we conclude that multivariate isoform-centric models provide a major improvement in prediction of individual isoforms compared to corresponding univariate models, which is the statistical unit of testing in isoTWAS. Based on predictions from multivariate models, we can study twice as many gene families. In addition, prediction of total gene expression does not drop using these isoform-centric models and, in fact, shows improvements in prediction compared to gene-centric TWAS models.

In addition, we suspected that multiple factors could influence these predictions, including the factors introduced by the reviewer. We summarize these in the Results section and expand on them in a Supplemental Note.

In the **revised Results section**, we write:

Line 183: “We considered 3 main criteria to evaluate the performance of both the multivariate and isoform-centric approaches of isoTWAS: (1) the number of isoforms whose expression can be imputed using multivariate/univariate models with cross-

validation (CV) $R^2 > 0.01$; (2) the number of unique genes with at least one isoform that can be imputed at CV $R^2 > 0.01$; and (3) the number of unique genes in which total gene expression can be imputed at CV $R^2 > 0.01$ using isoTWAS (summed) or TWAS models.”

The new Figure 3 details these metrics for brain tissues, with Supplemental Figures S5-S9 showing these metrics for all 48 tissues in GTEx. We now write in the Results section:

Line 189: “At the isoform level (criterion 1), through joint multivariate modeling of isoform expression, we trained 2.3-2.5-fold more models at cross-validation (CV) $R^2 > 0.01$ across the 48 tissues, compared to traditional univariate approaches (**Figure 3a, Supplemental Figure S5**). Using these multivariate models, we improve prediction for 79-82% of isoforms with a median increase of ~1.8-2.4 fold increase in adjusted R^2 (**Supplemental Figure S6, Supplemental Table S2**). Concordant with simulations, we found that the multivariate elastic net overwhelmingly outperformed other multivariate (and univariate) methods, indicating that leveraging the shared genetic architecture between isoforms of the same gene greatly aids in prediction of each individual isoform (**Supplemental Figure S7, Supplemental Table S2**). Notably, we observed that multivariate models were particularly important for brain tissues compared with non-brain tissues in GTEx, which showed significantly improved performance compared with univariate models (**Figure 3b**; $P = 0.011$ from OLS regression of median percent increase in CV R^2 for multivariate/univariate models against tissue type, adjusted for sample size), suggesting more shared isoQTL architecture in brain tissues than others which can be leveraged by isoTWAS for improved prediction. These gains in prediction accuracy directly translate into increased power in the trait association step⁵².

At the gene level (criteria 2 and 3), isoTWAS also increased the number of genes with testable models in the trait mapping step and improved prediction of total gene expression. The number of unique genes with at least 1 isoTWAS model at CV $R^2 > 0.01$ (inclusion criterion for isoTWAS trait mapping) was 1.9-2.5 times larger than the number of unique genes with TWAS models achieving CV $R^2 > 0.01$ for gene expression prediction (**Figure 3c, Supplemental Figure S8, Supplemental Table S2**). For a given gene, isoTWAS models (summed) outperformed TWAS models in prediction of total gene expression by a median of 25-70% in cross validation (**Supplemental Figure S9**) with a 50-80% increase in the number of genes that are predicted at CV $R^2 > 0.01$ (**Figure 3d, Supplemental Figure S10**). We replicated these gains in total gene expression prediction using an independent, out of sample QTL reference panel of adult

cortex from PsychENCODE/AMP-AD (Methods). Multivariate isoTWAS models outperformed univariate TWAS models in predicting total gene expression with a 15.2% median percent increase in adjusted R^2 when training in GTEx and testing in PsychENCODE/AMP-AD, and 23.9% vice versa; **Figure 3e, Supplemental Table S3**). As predictive performance is positively related to power to detect trait associations⁵¹, both the increased number and accuracy of trainable imputation models using isoTWAS's multivariate predictive framework have strong implications on increased discovery in trait mapping⁵¹."

Line 222: "As genes can differ widely with respect to the number and expression patterns of their constituent isoforms, as well as by other potentially relevant features such as gene length and SNP density, we next sought to characterize the impact of these factors of isoTWAS performance using GTEx models as evaluated by the 3 criteria outlined above (**Methods, Supplemental Note, Supplemental Figures S12-19, Supplemental Data 3-4**). Overall, we observed an increase in performance of isoTWAS multivariate modeling of both isoforms and genes with increasing number of isoforms per gene, although there was less conclusive of a pattern with increasing dominant isoform fraction⁵² (**Supplemental Figures S12-13, Supplemental Note, Methods**). We also noticed trends in the performance gain using isoTWAS multivariate modeling with respect to both isoform and gene expression prediction across gene length, SNP density at the gene locus, and sample size (**Supplemental Figures S14-16, Supplemental Note**). Finally, as the proportion of non-zero effect SNPs in the isoTWAS model that are shared across isoforms increased (**Supplemental Note**), we found an increasing trend in the gain in prediction of gene expression using isoTWAS compared to TWAS models (**Supplemental Figure S17**), reflecting a similar observation from simulation.

Lastly, we investigated how the robustness of isoform abundance estimation from short-read RNA-seq impacted performance gains of isoTWAS, compared with TWAS. Isoform abundance was initially quantified from probabilistic point estimates using Salmon, guided by Gencode annotations. We then assessed the performance of isoTWAS across loci binned by quantification variance measured across 50 inferential replicates from Salmon³². In general, we found that for isoform-level prediction, multivariate modeling in isoTWAS substantially outperformed univariate approaches as quantification variance increased. However, comparing isoTWAS isoform-centric and TWAS gene-centric models, there were no discernable trends in prediction of gene expression as the mean count and/or quantification variance of genes increased (**Supplemental Figure S18-19, Supplemental Note**)."

In the *newly added Supplemental Note*, we write:

“We evaluated the prediction of multivariate isoform-centric prediction models across a variety of factors that may influence the genetic architecture of isoform regulation at a locus or the inference of the regulation. In general, we computed three ratios: (1) the isoform prediction ratio, or the ratio of number of isoforms that are predicted at $CV R^2 > 0.01$ using multivariate and univariate models, (2) the inclusion criterion ratio, or the ratio of the number of genes that meet inclusion criteria for isoTWAS (gene with 1+ isoforms predicted at $CV R^2 > 0.01$) compared to TWAS (gene is predicted at $CV R^2 > 0.01$), and (3) the gene prediction ratio, or the ratio of the number of genes that are predicted at $CV R^2 > 0.01$ using isoform-centric isoTWAS models compared to gene-centric TWAS models. We plot boxplots of these ratios across the 48 GTEx tissues, and vary these factors across bins. In general, we note that, despite trends in these ratios across these factors, the ratios are always above 1, reinforcing the gains the prediction afforded by the multivariate isoform-centric prediction in isoTWAS.

1. *Number of expressed isoforms per gene.* **Supplemental Figure S12a** shows that the isoform prediction ratio stays relatively even as the number of isoforms per gene increases. The inclusion criterion ratio increases as the number of isoforms per gene increases but only until approximately 10 isoforms per gene. For genes with >10 isoforms per gene, the inclusion criterion ratio remains relatively even (**Supplemental Figure S12b**). There is a clear increasing trend in the median number of well-predicted isoforms per gene ($CV R^2 > 0.01$) as the number of expression isoforms per gene increases, suggesting that the increased number of isoforms per gene provides more information about shared genetic architecture between isoforms that can be leveraged for improved prediction (**Supplemental Figure S12c**). Lastly, in **Supplemental Figure S12d** we see a similar increasing trend in the gene prediction ratio as the number of isoforms per gene increases (increase in the ratio until approximately 10 isoforms per gene and a leveling off after).
2. *Maximum isoform fraction per gene.* We computed, using the raw counts of isoforms, the isoform fraction of each isoform of a gene using the `isoformtoIsoformFraction()` function in the Bioconductor package `IsoformSwitchAnalyzeR`¹. This function computes the fraction of each isoform’s expression to the total gene expression. We then found the isoform with the maximum isoform fraction for each gene and termed this fraction the maximum isoform fraction for the gene. In general, genes with large maximum isoform fraction are dominated by a single isoform, whereas genes with a small maximum isoform fraction have multiple isoforms with similar levels of expression. **Supplemental Figure S13** shows that, as maximum isoform fraction increases, there is a slight increase in the isoform prediction ratio, a larger increase in the inclusion criterion ratio, and no general trend in the gene prediction ratio.

3. *Gene length*. We computed the length of each gene as the difference in the end and start positions of the gene, as annotated in Ensembl v109. **Supplemental Figure S14** shows that, as gene length increases, there is a slight increase in both the isoform prediction ratio and the inclusion criterion ratio, and no general trend in the gene prediction ratio.
4. *SNP density*. We computed the number of SNPs that are within 1 Mb of the gene body, calling this value the SNP density of the gene locus. The SNP density represents the number of SNPs that comprise the design matrix in both the isoform-centric isoTWAS and gene-centric TWAS prediction models. **Supplemental Figure S15** shows that, as SNP density increases, there is a slight decrease in the isoform prediction ratio but no general trend in the inclusion criterion ratio and gene prediction ratio.
5. *Sample size*. **Supplemental Figure S16** shows that, as sample size increases, there is a decrease in the gene prediction ratio. This may reflect that, with larger sample sizes, gene-level expression QTLs may start to reflect the more subtle isoform-level expression QTLs, leading to a decrease in this ratio. We do note that, even at the largest sample sizes in GTEx, this gene prediction ratio is greater than 1. In datasets of small sample size (<175 samples), the isoform prediction ratio and inclusion criterion ratio are largest. These ratios decrease in datasets of larger sample size, but the ratio does not decrease consistently (e.g., isoform prediction ratio is higher in datasets of 250-500 samples compared to 175-200 and inclusion criterion ratio remain relatively similar as sample size increases beyond 175).
6. *Proportion of shared isoTWAS model effect SNPs*. For each isoform's predictive model, we determined which SNPs have large effects. First, we standardized the effect sizes in the model to mean 0 and unit variance. We found the SNPs whose effect sizes deviated significantly (Benjamini-Hochberg adjusted $P < 0.05$) and called them the isoform's effect SNPs. For each gene, we then computed the proportion of effect SNPs that were shared across all isoforms of the same gene. **Supplemental Figure S17** shows a clear increasing trend in the isoform prediction ratios, indicating that multivariate modelling can leverage shared isoform QTL architecture to improve marginal prediction of each isoform's expression. The inclusion criterion ratio remains relatively even as this proportion increases. The expression prediction ratio shows a decreasing trend as the proportion of shared isoTWAS effect SNPs increases, reflecting results from simulation (**Figure 2**).
7. *Mean normalized counts*. Here, using the isoform and gene counts normalized to library size and gene length, we compute each isoform and gene's mean normalized count across samples, using the countsFromAbundance = 'lengthScaledTPM' option from tximport. Since the bins of mean normalized counts for gene and isoform expression do not map one-to-one as for the previous factors, we only compute and plot the isoform prediction and gene prediction ratios. As the mean normalized counts for isoform expression increases, we see a decreasing trend in the isoform prediction ratio, with a slight increase in the bin of isoforms with large mean normalized counts. We see no clear trend in the gene expression prediction as the mean normalized gene counts increase, though we see a similar increase in the largest bin (**Supplemental Figure S18**).

8. *Quantification variance across inferential replicates of genes and isoforms.* Here, using the raw isoform and gene counts, we compute the quantification variance across inferential replicates from Salmon². We obtain the 50 inferential replicated from Salmon and import this using the Bioconductor package tximport³. We then computed the quantification variance for each isoform and gene using the computeInfRV() function from the Bioconductor package fishpond⁴. Briefly, this function first computes a matrix of variance (samples by features) across the 50 inferential replicates. Then, it computes the inferential quantification variance as the difference of the variance matrix and the mean counts matrix (Salmon Expectation-Maximization point estimates), standardized by the mean counts matrix. We collapse this inferential quantification variance to a per-feature measure by taking the mean of each row in the inferential quantification variance matrix. Again, since the bins of quantification variance for gene and isoform expression do not map one-to-one as for the previous factors, we only compute and plot the isoform prediction and gene prediction ratios. We find, for isoforms with low quantification variance (variance < 1.5), the isoform prediction ratio stays relatively even but increases as isoform quantification variance exceeds 1.5. However, there is no general trend with gene prediction ratio as gene quantification variance increases (**Supplemental Figure S19**). Leveraging this quantification variance to improve prediction is an interesting and worthwhile methodological opportunity that is discussed in the Discussion section.”

Supplemental Figures S12-19 are provided below.

Figure S12: Performance of isoTWAS across number of isoforms per gene across 48 GTEx tissues. (a) Ratio of number of isoforms predicted at $R^2 > 0.01$ using multivariate versus univariate prediction. (b) Ratio of number of genes passing CV threshold using isoTWAS versus TWAS. (c) Median number of isoforms predicted at CV $R^2 > 0.01$ in isoTWAS models across increasing number of isoforms per gene. The red line shows the line $Y = X + 1$. (d) Ratio of number of genes with CV $R^2 > 0.01$ using isoTWAS versus TWAS.

Figure S13: Performance of isoTWAS across increasing maximum isoform fraction across 48 GTEx tissues. **(a)** Ratio of number of isoforms predicted at $R^2 > 0.01$ using multivariate versus univariate prediction. **(b)** Ratio of number of genes passing CV threshold using isoTWAS versus TWAS. **(c)** Ratio of number of genes with CV $R^2 > 0.01$ using isoTWAS versus TWAS.

Figure S14: Performance of isoTWAS across increasing gene length across 48 GTEx tissues. **(a)** Ratio of number of isoforms predicted at $R^2 > 0.01$ using multivariate versus univariate prediction. **(b)** Ratio of number of genes passing CV threshold using isoTWAS versus TWAS. **(c)** Ratio of number of genes with CV $R^2 > 0.01$ using isoTWAS versus TWAS. $R^2 > 0.01$ using isoTWAS versus TWAS.

Figure S15: Performance of isoTWAS across increasing SNP density across 48 GTEx tissues. (a) Ratio of number of isoforms predicted at $R^2 > 0.01$ using multivariate versus univariate prediction. (b) Ratio of number of genes passing CV threshold using isoTWAS versus TWAS. (c) Ratio of number of genes with CV $R^2 > 0.01$ using isoTWAS versus TWAS.

Figure S16: Performance of isoTAS across increasing sample size across 48 GTEx tissues. **(a)** Ratio of number of isoforms predicted at $R^2 > 0.01$ using multivariate versus univariate prediction. **(b)** Ratio of number of genes passing CV threshold using isoTAS versus TWAS. **(c)** Ratio of number of genes with CV $R^2 > 0.01$ using isoTAS versus TWAS.

Figure S17: Performance of isoTWAS across proportion of SNPs with a non-zero effect on all isoforms of the gene, across 48 GTEx tissues. **(a)** Ratio of number of isoforms predicted at $R^2 > 0.01$ using multivariate versus univariate prediction. **(b)** Ratio of number of genes passing CV threshold using isoTWAS versus TWAS. **(c)** Ratio of number of genes with CV $R^2 > 0.01$ using isoTWAS versus TWAS.

Figure S18: Performance of isoTWAS across increasing mean counts of isoforms and genes across 48 GTEx tissues. (a) Ratio of number of isoforms predicted at $R^2 > 0.01$ using multivariate versus univariate prediction across increasing mean counts of isoforms. (b) Ratio of number of genes with $CV R^2 > 0.01$ using isoTWAS versus TWAS across increasing mean counts of genes.

Figure S19: Performance of isoTWAS across increasing quantification variance of isoforms and genes across 48 GTEx tissues. (a) Ratio of number of isoforms predicted at $R^2 > 0.01$ using multivariate versus univariate prediction across increasing quantification variance of isoforms. (b) Ratio of number of genes with CV $R^2 > 0.01$ using isoTWAS versus TWAS across increasing quantification variance of genes.

3. Are the genes in #1 enriched for genes with a large number of "detected" isoforms? Surely, if a gene consists of a single isoform, there should be no difference between the isoform-centric and conventional TWAS framework. It would appear then that the successful "detection" of these isoforms is a key component of the theoretical/empirical justification for the framework. What is the impact on performance when an isoform fails to be detected? This should be quantified.

The reviewer raises an interesting question, and we thank the reviewer for this suggestion for further evaluation. We have added an *in silico* experiment using GTEx data to assess the effect of successful detection of isoforms on TWAS and isoTWAS prediction. In short, for ~9000 randomly selected genes with 3-16 isoforms across the 13 brain tissues in GTEx, we randomly selected the isoform with the largest isoform fraction and removed it prior to aggregating expression to the gene-level. We then compared prediction of original gene expression (without this isoform removed) using TWAS and isoTWAS models trained in this synthetic leave-one-isoform-out dataset. In summary, we find two main results: (1) the advantage of using multivariate versus univariate models decreases when the dominant isoform is failed to be detected but this drop in performance is consistent across increasing numbers of isoforms per gene, and (2) both isoTWAS and TWAS prediction of gene-level expression suffers, but for genes with larger numbers of isoforms, the drop in performance of isoTWAS is negligible. Nevertheless, we find that, even when these isoforms are failed to be detected, multivariate isoform-centric models outperform univariate isoform-centric models in predicting isoform expression, and isoTWAS models outperform TWAS models in predicting gene expression. This analysis led to a **new Supplemental Figure S20**.

We discuss this additional evaluation briefly in the **revised Results section**:

Line 247: "Finally, we evaluated the impact of reference transcriptome annotation fidelity by generating a synthetic dataset quantified using a reference annotation masking the dominant isoforms for a set of genes. As expected, performance of both isoTWAS and TWAS models declined when isoforms failed to be detected in expression quantification (**Supplemental Note, Supplemental Figure S20**)."

We expand further in the *newly added Supplemental Note*.

“We consider a corollary experiment to assess how well isoTWAS prediction models impute gene and isoform expression when isoforms are failed to be detected. Across the 13 brain tissues in GTEx, we selected a random set of ~9000 genes in the following manner:

- We stratified our gene sets by the number of isoforms per gene and subset to genes with between 3 to 16 isoforms.
- We then randomly selected 50 genes from each group of genes with a certain number of isoforms per gene.

Then, we generated a synthetic leave-one-isoform-out (LOO) dataset, where the dominant isoform of each gene is missing. Using the raw transcript-level salmon quantifications, we removed each gene’s dominant isoform and summarized gene expression using tximeta’s summarizeToGene() function. Then, we trained isoTWAS and TWAS models in the synthetic datasets and imputed isoform (using isoTWAS) and gene (using isoTWAS and TWAS) expression in the original GTEx datasets. We compared these leave-one-out predictions to predictions from models trained in the original GTEx datasets.

Supplemental Figure S20a plots the distribution of the percent difference in isoform expression (using multivariate compared to univariate models), when using the original and synthetic leave-one-out training datasets. We find that the advantage of the multivariate models over univariate models greatly decreases with using this leave-one-out dataset compared to the true original, true gene expression measures. In addition, this drop in performance is consistent as we increase the number of isoforms per gene in the original dataset. This result is unsurprising as, in the leave-one-out dataset, though each isoform’s expression remains equal to its expression in the original dataset, the correlation structure is altered. The multivariate models depend on shared genetic architecture to best predict each isoform’s expression marginally.

Supplemental Figure 20b plots the distribution of the percent difference in gene expression (using isoTWAS compared to TWAS models), when using the original and synthetic leave-one-out training datasets. We find that the advantage of the isoTWAS models over TWAS models decreases using this leave-one-out dataset compared to the true original, but only for genes with smaller numbers of isoforms per gene. As the number of isoforms per gene in the original dataset increases, this drop in performance decreases. In general, these results suggest that, when a large portion of the total gene's expression is removed from the dataset, both isoform- and gene-centric expression prediction is negatively affected. Taken together, these results underscore that proper gene and isoform annotations are important when building a genetic predictor, and motivates the need to continue developing more comprehensive and tissue-specific transcriptome annotations. We discuss this further in the Discussion section."

4. The isoform quantification accuracy's impact on downstream analyses must be investigated and quantified. The isoform-level quantification is typically much more difficult than gene-expression quantification.

We thank the reviewer for this suggestion, as it contextualizes our method's suggestions for future work. We investigate the impact of quantification accuracy on prediction of both gene and isoform expression prediction across the 48 GTEx tissues. We find no discernable trends in prediction of gene expression using isoTWAS versus TWAS models as the quantification variance across inferential replicates in gene expression increases. However, we find that the predictive advantage of multivariate versus univariate models for isoform expression increases with increased quantification variance, with no. This latter result raises an interesting methodological opportunity to

leverage the inferential replicates to boost prediction, improving our paper's methodological discussion. We briefly discuss this in the *revised Results section*:

Line 239: "Lastly, we investigated how the robustness of isoform abundance estimation from short-read RNA-seq impacted performance gains of isoTWAS, compared with TWAS. Isoform abundance was initially quantified from probabilistic point estimates using Salmon, guided by Gencode annotations. We then assessed the performance of isoTWAS across loci binned by quantification variance measured across 50 inferential replicates from Salmon³². In general, we found that for isoform-level prediction, multivariate modeling in isoTWAS substantially outperformed univariate approaches as quantification variance increased. However, comparing isoTWAS isoform-centric and TWAS gene-centric models, there were no discernable trends in prediction of gene expression as the mean count and/or quantification variance of genes increased (**Supplemental Figure S18-19, Supplemental Note**)."

We expand further in the *Supplemental Note*:

"Here, using the raw isoform and gene counts, we compute the quantification variance across inferential replicates from Salmon². We obtain the 50 inferential replicates from Salmon and import this using the Bioconductor package tximport³. We then computed the quantification variance for each isoform and gene using the computeInfRV() function from the Bioconductor package fishpond⁴. Briefly, this function first computes a matrix of variance (samples by features) across the 50 inferential replicates. Then, it computes the inferential quantification variance as the difference of the variance matrix and the mean counts matrix (Salmon Expectation-Maximization point estimates), standardized by the mean counts matrix. We collapse this inferential quantification variance to a per-feature measure by taking the mean of each row in the inferential quantification variance matrix. Again, since the bins of quantification variance for gene and isoform expression do not map one-to-one as for the previous factors, we only compute and plot the isoform prediction and gene prediction ratios. We find, for isoforms with low quantification variance (variance < 1.5), the isoform prediction ratio stays relatively even but increases as isoform quantification variance exceeds 1.5. However, there is no general trend with gene prediction ratio as gene quantification variance increases (**Supplemental Figure S19**). Leveraging this quantification variance to improve prediction is an interesting and worthwhile methodological opportunity that is discussed in the Discussion section."

We note that we discuss this point as a limitation and possible methodological extension of our method in the original manuscript. We have expanded on this to point to our analysis. In the *revised Discussion*, we write:

LINE 460: “First, we note that isoform-level expression quantifications are maximum-likelihood estimates, due to the inherent limitations of short-read RNA-seq. These estimates are generally guided by existing transcriptome annotations (e.g., GENCODE) and thus are dependent on the completeness and accuracy of these genomic annotations.”

Line 477: “Second, while inferential replicates from RNA-seq quantification can provide measures of technical variation, they are not incorporated into the predictive models. Our analyses of prediction across inferential replicates suggest a methodological opportunity: leveraging these inferential replicates as a measure of quantification error may help in estimating the robustness of isoform prediction and, potentially, of the precision of these SNP effects. A more flexible predictive model that estimates standard errors for SNP effects by model-averaging across the replicate datasets may help with trait mapping by providing a prediction interval for both isoform- and gene-level imputed expression.”

5. Gene complexity can have a substantial impact on the conclusions that can be reliably drawn. One scenario is a gene with a large number of very similar isoforms. Here, isoform quantification will be extremely difficult and may suffer from low accuracy since mapping the reads to the correct isoform source will be extremely challenging. The case of a gene with a large number of isoforms also presents its own challenges and may suffer from low quantification accuracy. These scenarios can impact the imputation of isoform expression (ie., garbage in, garbage out). (Note that the authors' simulation framework leverages empirical data from only 2 genes [lines 581-588] and appears to be overly simplistic.)

We thank the reviewer for this important consideration. As we mention in our response to the reviewer's Remarks 1 and 2, we have assessed isoform and gene expression prediction across a large number of factors, one of which is the number of isoforms per gene. In this evaluation, we find that there is an increase in prediction of gene expression using isoTWAS versus TWAS models when the number of isoforms per gene increases; we did not observe a large increase in advantage of multivariate models over univariate models. As such, we believe that the advantages of isoTWAS over TWAS in prediction hold for genes with large numbers of isoforms. We have also assessed, as mentioned in our response to Remarks 1, 2, and 4, how these predictions vary with respect to quantification variance across inferential replicates of isoform and gene expression. In this evaluation, we find that there is no increase or decrease in prediction of gene expression using isoTWAS versus TWAS models when quantification variance increases; we observe a large increase in advantage of multivariate models over univariate models in prediction of isoform expression when quantification variance increases. Please refer to our responses to these remarks for more details.

To address the similarity of isoforms, we have also introduced a more complex simulation framework that has two parameters that govern the similarity between isoforms: a parameter p_s that determines the proportion of causal effects that are shared across isoforms, and a parameter σ_h that tunes the non *cis*-genetic effects on isoform expression that is shared between isoforms and between samples. These simulations show that, in general, as p_s increases, there is a drop in the advantage of isoTWAS models over TWAS models in predicting gene expression. When σ_h increases, the advantage of isoTWAS models over TWAS models in predicting gene expression is maintained. In our GTEx evaluation, we also empirically assess how prediction of isoform and gene expression is affected when we find shared genetic effects across isoforms of the same gene; we reinforce this slight drop in the advantage isoTWAS over TWAS models in predicting gene expression when the isoTWAS models have increasing proportions of shared effects. Interestingly, we find a large increase in the advantage of multivariate versus univariate models in predicting isoform expression as this proportion

increases. This last result reinforces the utility of considering isoforms jointly to increase marginal prediction of individual isoforms.

In the *revised Results section*, we write:

Line 161: “To systematically evaluate the performance of TWAS versus isoTWAS models on prediction of total gene expression across a variety of genetic architectures, we conducted an extensive set of simulations across 22 different gene loci using European-ancestry data from the 1000 Genomes Project⁵¹ (**Methods, Figure 2a**). At each gene locus, we controlled gene expression heritability and simulated 2-10 distinct isoforms, varying the proportion of causal isoQTLs (p_{causal}) and their sharing between isoforms (p_{shared}). We then trained cross-validated, multivariate predictive models of isoform expression (isoTWAS) or univariate models of gene expression (TWAS). For isoTWAS, of the specific multivariate prediction models tested, multivariate elastic net⁴³ demonstrated the greatest CV prediction of isoform expression across most simulation settings (**Figure 2b, Supplemental Figure S3, Supplemental Data 1**). For gene expression prediction, the optimal isoTWAS models (in sum) outperformed the optimal TWAS model, particularly at sparser isoQTL architectures, with median absolute increase in adjusted R^2 of 0.6-3.5% (**Figure 2c, Supplemental Figure S4, Supplemental Data 2**). Performance gains decreased somewhat with denser isoQTL architectures, although real data is consistent with 0.1-1% sparsity (i.e., 1-10 causal e- and isoQTLs per gene or isoform)³⁷. In simulations, we found that isoTWAS prediction of gene expression also increases as the proportion of shared non-zero effect SNPs across isoforms decreases (**Figure 2b-c, Supplemental Figure S4, Supplemental Data 2**).”

Line 232: “Finally, as the proportion of non-zero effect SNPs in the isoTWAS model that are shared across isoforms increased (**Supplemental Note**), we found an increasing trend in the gain in prediction of gene expression using isoTWAS compared to TWAS models (**Supplemental Figure S17**), reflecting a similar observation from simulation. Interestingly, as this proportion increased, we found an increase in the gain in prediction of isoform expression, reinforcing the utility of multivariate modelling in marginal prediction of isoform expression.”

In the *revised Methods section*, we now detail the more complex simulation framework:

Line 611: “We adopt techniques from Mancuso et al’s *twas_sim* protocol to simulate multivariate isoform expression based on randomly simulated genotypes and environmental random noise. First, for n samples, we generate a matrix of genotype dosages for the SNPs within 1 Megabase of 22 different genes (1 per chromosome) using an LD reference panel from European samples from 1000 Genomes Project.

Next, we generate a matrix of SNP-isoform effects across different causal SNP proportions p_c , numbers of isoforms t , and p_s proportion of the SNP-isoform effects being shared across isoforms of the same gene. We then add two matrices of random noise U and ϵ . The first noise matrix U represents non *cis*-genetic effects on isoforms that are correlated between samples and isoforms; we control the proportion of variance explained in isoform expression attributed to U using a parameter σ_h . The second matrix ϵ is a matrix of random noise that is independent for each isoform, such that $\epsilon_i \sim N(0, \sigma_e^2 I)$ where $\sigma_e^2 = 1 - \sigma_h - h_g^2$. We generate 10,000 simulations for each configuration of the simulation parameters, varying $n \in \{200, 500\}$, $p_c \in \{0.001, 0.01, 0.05\}$, $h_g^2 \in \{0.05, 0.10, 0.25\}$, $p_s \in \{0, 0.5, 1\}$, and $\sigma_h \in \{0.1, 0.25\}$. Full mathematical details are provided in **Supplemental Methods** and summarized in **Figure 2**.”

6. An isoform-centric framework comes with its own challenges. Since, as the authors point out, the isoforms for a gene can be highly correlated, the question of the causal isoform for a trait is of critical importance. This problem is very similar to the problem of proximal, correlated genes within a locus making it difficult to identify the causal gene in the locus. It's not clear that the isoform-centric analysis necessarily leads to improved causal discovery rate.

We fully agree with the reviewer about the challenges in fine-mapping using isoform-centric predictions. We prioritize an isoform-trait association in three steps. First, we detect an isoform-trait association by first adjusting false discovery across the ACAT-aggregated gene-level P-values. Second, we apply a permutation test whereby the SNP-to-isoform effects are permuted 10,000 times to generate a null distribution; this permutation test assesses how much signal is added by isoform expression, given the GWAS architecture of the locus, and controls for large LD blocks. Third, for isoforms that overlap within a 1 Mb window, we apply isoform-level probabilistic fine-mapping using Mancuso et al's FOCUS framework. The FOCUS framework accounts for correlations between imputed expression of these overlapping genes due to LD or pleiotropic SNP effects to generate a credible set of isoforms that best explains the risk region. We emphasize that the overall goal of the isoTWAS framework is to detect associations between genetically imputed isoform expression and a complex trait. In this revision, we explored the ability of FOCUS to identify the effect isoform at a risk region through simulations and the number of genes and isoforms at a risk region and credible sets in the empirical settings.

Motivated by reviewer comments, we greatly enhance the simulation framework to explore isoTWAS potential to identify the causal transcript. We extend our simulation framework to include a fine-mapping assessment. Here, we simulate a gene with 5 or 10 isoforms and a single effect isoform across a variety of simulation parameters for the isoform QTL architecture. We computed the sensitivity of 90% credible sets of isoforms (proportion of credible sets that contain the effect isoform) and the number of isoforms in the 90% credible set. As discussed in Mancuso et al, high SNP pleiotropy renders statistical fine-mapping difficult, especially when the risk region includes multiple transcripts. In our analysis, we find a similar result: the sensitivity of the 90% credible sets are within 60-75%. In general, in Supplemental Figure 23 (pasted below), we find that the sensitivity of 90% credible sets decreases and the mean set size increases as the proportion of shared isoQTLs increases, likely because of difficulties in fine-mapping when QTL horizontal pleiotropy is extremely high.

We write in the *revised Results section*:

Line 272: “Finally, we assessed the performance of probabilistic fine-mapping in identifying the true effect isoform in our simulation framework of genes with 5 or 10 isoforms (**Methods, Supplemental Figure S23, Supplemental Data 9**). In general, the sensitivity of 90% credible sets (proportion of credible sets that contained the true effect isoform) was under-calibrated, likely due to difficulties in fine-mapping when QTL horizontal pleiotropy is extremely high⁵³. We found that the sensitivity of 90% credible sets decreased and the mean set size increased with increasing proportion of shared isoQTLs.”

In the empirical setting, across 15 traits and models from 2 tissues (adult and developmental cortex), we find that the difference in the number of genes prioritized in the risk region using isoTWAS and TWAS models is relatively small (3.90 with isoTWAS and 3.15 with TWAS, with $P < 0.05$ due to the large number of total risk regions). The mean number of genes in the 90% credible set is generally even (1.25 using isoTWAS and 1.33 using TWAS, with $P < 0.05$ due to the large number of total credible sets). In

addition, we find that, in general, isoTWAS risk regions contain 1.54 isoforms per gene and isoTWAS credible sets contain 1.27 isoforms per gene. The distribution of the number of genes and isoforms per gene in risk regions and 90% credible sets are plotted in Supplemental Figure S28, pasted below.

In the *revised Results section*, we write:

Line 345: “We also empirically compared probabilistic fine-mapping⁵⁵ of results from isoTWAS and gene-level TWAS (**Methods**). Here, we conducted fine-mapping on significant trait-associated genes/isoforms (adjusted $P < 0.05$ and permutation $P < 0.05$) and are within 1 Mb of one another; we term a locus with overlapping genes/isoforms a risk region. Overall, the mean number of genes in a risk region using TWAS was 3.15 compared to 3.90 using isoTWAS (**Supplemental Figure 28a**); the mean number of genes in a 90% credible set using TWAS was 1.33 compared to 1.25 using isoTWAS (**Supplemental Figure 28a**). On average, there were 1.54 isoforms per gene in a risk region and 1.27 isoforms per gene in a 90% credible set (**Supplemental Figure 28b**). Isoform-centric modeling presents unique challenges for fine-mapping due to potentially high levels of horizontal pleiotropy, and remains an important and open question for the field. Nevertheless, isoTWAS identified a relatively comparable number of genes in risk regions compared with TWAS, and the combination of conservative two-step trait mapping, permutation testing, and probabilistic fine-mapping were critical for maintaining a narrow credible set size.”

Taken together, these simulation and empirical results indicate that FOCUS fine-mapping using gene- and isoform-centric models are similar as the number of isoforms per gene are close to 1. However, as our simulations suggest, if multiple isoforms of the same gene with a large proportion of shared isoQTLs persist past two-step trait mapping and permutation testing, 90% credible sets have reduced sensitivity. This presents an interesting opportunity for future methods development to account for QTL pleiotropy in fine-mapping approaches as a complement to isoTWAS. We write in the **revised Discussion section**:

Line 488: “Lastly, as we show, this framework can suffer from reduced power, inflated false positives, and reduced sensitivity in fine-mapping in the presence of SNP horizontal pleiotropy, where the genetic variants in the isoform expression model affect the trait, independent of isoform expression, or when multiple SNPs affect expression of isoforms^{95,96}. For pathways that are not observed or accounted for in the reference expression panel and GWAS, accounting for horizontal pleiotropy may improve trait mapping. We motivate further methodological extensions of probabilistic fine-mapping to reconcile pleiotropy for SNPs shared across models for multiple isoforms at the same genetic locus, as summary-statistic based methods that control for horizontal pleiotropy are not yet effective⁹⁷.”

7. The highly powered schizophrenia GWAS (relatively to the less powered GWASs for the other neuropsychiatric traits) appears to show a greater shift towards low p-values from conventional TWAS than from the isoform-centric TWAS (Figure S13 Q-Q plots). This would imply a substantially improved ability to detect gene-trait associations from the conventional TWAS, contradicting one of the major conclusions of the study.

We thank the reviewer for this comment. In this revision, we remake the QQ-plots with our updated models and now also remove the MHC locus (Supplemental Figure S27, pasted below) for associations for all 15 traits and find a shift towards smaller P-values for isoTWAS associations compared to TWAS associations. This shift reflects the 10-20% increase in converted χ^2 test statistics (squared Z-scores) seen in Figure 5e.

We write in the *revised Results section*:

Line 333: “To investigate whether this increase in trait mapping discovery reflected true biological signal, rather than test statistic inflation due to the increased number of tests (~4-fold increase in number of tests), we next compared the null distributions across

methods for results across the 15 traits (**Supplemental Figure S27**). As the genomic inflation factor is not a reliable measure in TWAS settings⁷⁹, we estimated inflation in gene-level test statistics using an empirical Bayes approach (**Methods**). Collapsing across all 15 traits, there were no significant differences between TWAS and isoTWAS in the 95% credible intervals for test statistic inflation (**Figure 5d**). Using a heuristic to estimate increases in effective sample size (**Methods**), we observed an approximate increase in effective sample size of 10-20% when using isoTWAS compared to TWAS (**Figure 5e, Supplemental Table S9**). These analyses indicate that isoTWAS discovery is both well-calibrated to the null and facilitates increased discovery in real data compared to gene-level TWAS.”

8. The scenario of different genetic effects on phenotype across isoforms is an interesting one, but the study has not adequately investigated how widespread this phenomenon is. Thus, it's also not clear how informative the study's simulation framework is.

This is an important question that the reviewer asks. For isoTWAS associations at adjusted $P < 0.05$ and permutation $P < 0.05$, we find 1,335 genes with multiple isoforms that are associated with the trait. Of these genes, 661 genes exhibited isoform-trait associations in different directions.

We write in the **revised Results section**:

Line 300: "In addition, of the 1,335 genes with multiple isoform-trait associations, 661 gene exhibited distinct isoform-level associations in different directions"

Minor

1. Figure legends for figure 2c and 2d should be switched.

We thank the reviewer for this comment. The new Figure 3 (corresponding to the old Figure 2) is reworked.

Reviewer #3:

Remarks to the Author:

Summary of the key results: introducing isoform level TWAS and applies to two major datasets.

Originality and significance: Novel, and timely. The significance of the results in practice seems to be moderate or not shown convincingly.

Data & methodology: The analysis is done well.

Appropriate use of statistics and treatment of uncertainties: The statistics are done well.

Conclusions: conclusions are valid but oversold.

Suggested improvements: See below.

References: the authors appropriately credit previous work

Clarity and context: See below.

The study presents iso-TWAS, an isoform level extension of TWAS. Expanding TWAS to isoform level analysis is a natural generalization of the method and is timely. I think the comparison of several multivariate methods for doing isoform-level TWAS is impressive. Overall, I am excited about the topic, and even though I'm only moderately enthusiastic about it (see below), I would be happy to see this timely work published as a first-in-class method for isoform-level TWAS.

The methods seem to be done generally well, but the overall added value of the analysis appears to be moderate in practice or at least not shown convincingly. From the biology standpoint, I find the results limited and the follow-up rather superficial, not living up to the

paper's title. As for the presentation, the text needs 1) reorganization to improve the logical flow between simulation and real data, 2) polishing to improve the methods' clarity, and 3) toning down throughout to represent the impact more realistically.

We thank the reviewer for recognizing the value of our work and appreciating that we are presenting a first attempt at a new approach to model genetic effects on the transcriptome and integrate these with complex trait genetics. Upon the reviewer's many helpful suggestions and comments, we have worked to reorganize the paper to improve the flow, clarity, and interpretation. We have increased the complexity of our simulation framework, included an exhaustive evaluation of prediction metrics in GTEx, and increased our sample sizes for the adult brain cortex reference panel to compare discovery using isoTWAS and gene-level TWAS. We address the reviewer's comments point-by-point below.

Below are my remaining specific comments:

– This statement is confusing as the expression of the gene is generally the sum of length normalized isoform expressions like TPM:

Line 76- "In addition, we assume that the abundance of a gene (total gene expression) is the sum of the abundance of its isoforms (isoform expression) (Supplemental Figure S1), on the raw count scale." Please clarify/justify.

We thank the reviewer for pointing out this unclear statement in our manuscript. As we use salmon and tximeta to quantify and import transcript-level expression and sum to gene-level expression, we are, as the reviewer says, measuring gene expression as the sum of transcript-level measures of transcripts per million (TPM). We have corrected this in the manuscript. Specifically in the Results section, we write:

LINE 115: "In addition, we assume that the abundance of a gene is measured as the is the sum of the abundance of its isoforms, computed as transcripts per million, or TPM (Supplemental Figure S1)^{32,33,41,42}."

- Edit text: "multivariate data object" is programming lingo.

The reviewer is correct that this terminology is awkward. We have corrected it to "matrix" in the main text:

LINE 112: "We extend the traditional gene-level TWAS approach by jointly modeling the expression of distinct transcript-isoforms of a given gene as a matrix while accounting for the pair-wise correlations between these isoforms^{21,37,39,40}."

– Figure 2 labels are mixed up, and it has typos (heritable).

We thank the reviewer for this comment. The new Figure 3 (corresponding to the old Figure 2) is reworked.

- For figure 2 and other performance reports in the paper, what is the data set used? Considering that the genes are selected by different heritability tests this affects the stats reported. To me, a reasonable set could be either all genes, or all protein-coding genes, or genes that are shared between the two methods, or genes that are the union of the two methods.

The reviewer is correct in asking us to redefine the feature set of genes that are compared across gene- and isoform-level TWAS, especially for the evaluations for prediction. As the other reviewer points out, it may be the case that heritability estimation for gene and isoform expression may require different methods with different assumptions. In this revision, we have dropped the filtering by heritability. The evaluations of prediction are now conducted for all genes with >1 expressed isoform in a given tissue-specific dataset (48 tissues from GTEx and 2 other pre-frontal cortex datasets).

Accordingly, we have re-trained isoform- and gene-specific predictive models without considering the heritability cutoff. In addition, we have expanded our sample sizes for the adult brain cortex reference panel (N = 2115), using data from PsychENCODE and now adding AMP-AD to increase our power. In addition, we restrict our SNP to those included in HapMap3, which are most well imputed across genotype arrays. We have conducted multiple evaluations of the difference in cross-validation predictive R^2 with respect to three primary metrics considering all genes with multiple isoforms:

1. Prediction of individual isoforms using a multivariate approach that jointly models all isoforms of a gene (multivariate elastic net, multivariate regression with covariance estimation, sparse partial least squares, and multivariate elastic net with stacked generalization) compared to univariate models of individual isoforms. Here, we consider the difference in predictive R^2 (multivariate – univariate; 5-fold cross validation) and the number of isoforms that reach cross-validation $R^2 > 0.01$ using either approach.
2. Number of genes that pass cross-validation prediction cutoffs using isoTWAS (a gene with >1 isoform predicted at CV $R^2 > 0.01$) and TWAS (a gene predicted itself at CV $R^2 > 0.01$). We acknowledge that this metric could be biased towards isoTWAS, since genes with multiple isoforms have more of a chance at random to meet the criterion. However, independent of comparisons between isoTWAS and TWAS, this inclusion criterion for cross-validation prediction are standard to TWAS methods.
3. Prediction of total gene expression using isoform-specific isoTWAS models (summed across isoforms for each gene) and gene-specific TWAS models. Here, we consider the difference in predictive R^2 (isoTWAS - TWAS) and the number of

genes that reach cross-validation $R^2 > 0.01$ using each method. The purpose of isoTWAS prediction is to predict individual isoforms of genes. This prediction of total gene expression gives a measure of how quantification error impacts prediction using isoform-specific models.

These changes in evaluation are reflected in the new Figure 3.

It is worth mentioning that the inclusion criterion for hypothesis testing still requires a filtering step ($CV R^2 > 0.01$). In practice, gene-level TWAS and isoform-level TWAS have different inclusion criteria based on this cross-validation R^2 since the testing units are different (genes vs. isoforms). In our simulations, we circumvent this filtration step. In our trait mapping analyses using 15 complex neuropsychiatric or brain-related traits, we consider associations using gene-level TWAS models with $CV R^2 > 0.01$ and isoform-level isoTWAS models with $CV R^2 > 0.01$.

- Standard univariate Susie the model outperforms previous regression models used for TWAS (elastic net and bsimm) and is widely used. Considering that the model selected here is a multivariate extension of Susie, it'd be appropriate to show that the improved accuracy is associated with isoform level modeling and not Susie; this would be a comparison between standard susie applied to -gene level- expressions vs multivariate susie applied to -isoform level- expression. Reading through the paper, it seems like one of the "multivariate" methods used is the naive use of univariate methods on all isoforms independently, which that does include three different methods, including susie. It goes without saying the reviewer is confused! If the mutivariate susie was indeed compared to how susie is usually used, please revise the text to avoid confusing the reader.

We thank the reviewer for this comment and apologize for the confusion. To clarify this important point, we run three sets of predictive models in this paper:

- (1) univariate prediction of total gene expression, equivalent to traditional gene level TWAS;
- (2) univariate prediction of isoform expression;
- (3) multivariate, joint modeling of isoform expression for a given gene.

The same univariate models are employed in (1) and (2) above and include elastic net, BLUP, and SuSIE, as currently implemented in the TWAS FUSION package. Therefore, to answer one of the questions above, we do already compare SuSIE applied to gene expression vs multivariate SuSIE applied to isoform level expression.

For (3), we clarify in the text now the main multivariate models that are used to predict the matrix of isoform expression jointly, which include multivariate regression with covariance estimation, sparse partial least squares, and multivariate elastic net with stacked generalization. We note here, that during the review process, we uncovered a small error in the software that mis-labelled the CV R^2 for multivariate SuSIE. Correcting this label swap revealed that the multivariate elastic net was actually our strongest performing model, rather than multivariate SuSIE. Ultimately, multivariate SuSIE did not manage to show strong enough performance in predicting isoform expression to justify its large computational time and required several iterations to define a proper prior covariance structure; we note that multivariate SuSIE is developed as a tool for fine-mapping associations and not prediction. We took this opportunity to place multivariate elastic net (rather than multivariate SuSIE) as the main default multivariate method in the isoTWAS package and add two multivariate modelling frameworks for further comparison: elastic net with stacked generalization (joinet) and sparse partial least squares (spl). Multivariate elastic net continued to outperform these additional methods.

In the **revised Results section**, we now write:

Line 126: “The isoTWAS framework contains three general steps (**Figure 1**). First, we build multivariate predictive models of isoform-level expression using well-powered functional genomics training datasets, including GTEx³⁷ and PsychENCODE^{24,27}. Here, we trained and systematically compared 4 multivariate predictive frameworks: (1) multivariate elastic net penalized regression⁴³, (2) multivariate LASSO penalized regression with simultaneous covariance estimation (MRCE)⁴⁴, (3) multivariate elastic net regression with stacked generalization (jointnet)⁴⁵, and (4) sparse partial least squares (SPLS)⁴⁶. As a baseline for comparison, we also modeled each individual isoform independently with univariate regularized regressions, as implemented in Gusev et al’s FUSION software^{4,43,47,48} (see **Methods** and **Supplemental Methods**). Models were trained to predict isoform expression using the set of *cis*-SNPs within 1 Megabase (Mb) of the gene body (**Methods, Figure 1b**). Model performance was assessed via 5-fold cross-validation, using McNemar’s adjusted R^2 between observed and predicted expression.”

In the **revised Methods section**, we reinforce that this univariate modeling of isoforms serves as a baseline for comparison for our 4 multivariate modeling choices. We write:

Line 563: “...the simplest implemented method is the univariate predictive modelling used in FUSION. We disregard the correlation structure between isoforms and train a univariate elastic net, estimation of the best linear unbiased predictor (BLUP) in a linear mixed model⁸⁰, and SuSiE predictive model for each isoform separately. The model with the largest adjusted R^2 out of these three models is outputted. This approach serves as a baseline measurement for prediction of each isoform independently.”

We also write in the Methods section:

LINE 541: “In settings where we are interested in predicting gene-level expression from these predicted isoforms, isoTWAS trains an elastic net penalized linear regression that predicts gene-level expression from genetically-predicted isoform-level expression; this model training is conducted across the same 5 folds to prevent data leakage.”

– 5% nominal p-value for h^2 cut-off for the best isoform in a gene seems like a non-trivial choice, especially in genes with many isoforms. Please justify or correct this choice. Would it be appropriate to use some form of multiple hypothesis correction or extreme value modeling to avoid increased false positives in genes with many isoforms? The number of TWAS testable genes (e.g. Fig S6, or paragraph starting at line 139) is directly connected to how this is done.

We thank the reviewer for this comment about the choice of heritability correct in asking us to redefine the feature set of genes that are compared across gene- and isoform-level TWAS, especially for the evaluations for prediction. As the other reviewer points out, it may be the case that heritability estimation for gene and isoform expression may require different methods with different assumptions. In this revision, we have dropped the filtering by heritability and have considered all genes in this comparison. Dropping this filtering step does not change the conclusions of our empirical evaluations of isoTWAS.

– From the text it appears that less than half of the genes have more than one isoforms: "on average, four GENCODE v40-annotated isoforms per gene (... , median 1)" line 75. But looking at Figure S4 (and several other stats reported in the text) the multivariate model always outperforms the univariate model in >80% of the genes. Are these results limited to genes with multiple isoforms? If so please clarify the text and report appropriately.

We thank the reviewer for asking for this clarification, as it aids with interpretability of the results. We restrict our analyses to genes with multiple expressed isoforms. In the Results section, we now write:

LINE 179: "Next, we assessed predictive performance in real data from 48 tissues (13 brain) with sufficient sample sizes ($N > 100$) in GTEx for all genes with multiple expressed isoforms (**Supplemental Table S1; Methods**)."

– It seems to me that the gains by the multivariate model are stronger in low-sample tissues and fade away in larger sampled tissues (Figs S4 and S7). This is a little unexpected to me in the case of gene-level predictions at least. As the multivariate model is more complex model than a gene-level model which should be underpowered in smaller datasets and better performing in larger datasets. Please clarify or explore this issue.

The reviewer raises an interesting question about the relationship between predictive performance and sample size that we explore in the revision. We evaluated multiple factors that could influence these predictions in GTEx data, including sample size. In general, we do find that for prediction of total gene expression, the performance gains of isoTWAS compared with traditional TWAS become attenuated as sample size increases. The ratio of the number of genes that pass inclusion criteria using isoTWAS and TWAS decreases for datasets of >175 samples, but stays relatively even thereafter as sample size increases. Consistent with our expectations, the ratio of the number of well-predicted genes using isoTWAS versus TWAS declines with increased sample size, as we would expect both isoTWAS and TWAS models to converge towards capturing the “true” underlying eQTL architecture for a given gene. Nevertheless, even for datasets of large sample size, we find overall considerable improvements in the number of isoforms and number of genes that can be predicted using multivariate compared to univariate models and isoTWAS compared to TWAS models, respectively. We summarize in the revised Results section:

Line 229: “We also noticed trends in the performance gain using isoTWAS multivariate modeling with respect to both isoform and gene expression prediction across gene length, SNP density at the gene locus, and sample size (**Supplemental Figures S14-16, Supplemental Note**).”

We write, in the **Supplemental Note**:

“For these comparison, we computed three ratios: (1) the isoform prediction ratio, or the ratio of number of isoforms that are predicted at $CV R^2 > 0.01$ using multivariate compared with univariate models, (2) the inclusion criterion ratio, or the ratio of the number of genes that meet inclusion criteria for isoTWAS (gene with 1+ isoforms predicted at $CV R^2 > 0.01$) compared to TWAS (gene is predicted at $CV R^2 > 0.01$), and (3) the gene prediction ratio, or the ratio of the number of genes that can be predicted at $CV R^2 > 0.01$ using isoform-centric isoTWAS models (summed per gene) compared to gene-centric TWAS models. We show boxplots of these ratios across the 48 GTEx

tissues, and vary these factors across bins. In general, we note that these ratios are always above 1, reinforcing the overall gains in prediction afforded by the multivariate isoform-centric modeling framework of isoTWAS.

Supplemental Figure S16 shows that, as sample size increases, there is a decrease in the gene prediction ratio. This may reflect that, with larger sample sizes, gene-level expression QTLs may start to reflect the more subtle isoform-level expression QTLs, leading to a decrease in this ratio. We do note that, even at the largest sample sizes in GTEx, this gene prediction ratio is greater than 1. In datasets of small sample size (<175 samples), the isoform prediction ratio and inclusion criterion ratio are largest. These ratios decrease in datasets of larger sample size, but the ratio does not decrease consistently (e.g., isoform prediction ratio is higher in datasets of 250-500 samples compared to 175-200 and inclusion criterion ratio remain relatively similar as sample size increases beyond 175).”

– In Fig 2C and other places in the paper, the number of heritable isoforms is compared to the number of heritable genes. This is a little misleading because handling multiple isoforms in a single gene in such an analysis would be non-trivial. For instance, if multiple isoforms in a gene are in high correlation, they'll be counted multiple times. One naive approach would be to "impute" GRex for all isoform in a gene by its gene-level prediction for the univariate model, and then use that for comparison at an isoform level. If the authors have already addressed this issue in some other way, please clarify the text to reflect it.

We thank the reviewer for explaining how this comparison was confusing in the original manuscript. As we explain above, we have decided to drop the heritability filtering step and now compare all genes. This decision allows for a uniform comparison between predictions from isoTWAS and TWAS. Nevertheless, the overall conclusions of the advantage of isoTWAS over TWAS in prediction and discovery of trait associations do not change.

- Paragraph starting on line 131, is confusing and out of the flow of the paper (e.g. the reference to comparison among different multivariate methods).

We thank the reviewer for this comment that helped us re-organize the paper. This paragraph is now integrated more concisely with better flow in the Results section. We now write in the **revised Results section**:

Line 186: “At the isoform level (criterion 1), through joint multivariate modeling of isoform expression, we trained 2.3-2.5-fold more models at cross-validation (CV) $R^2 > 0.01$ across the 48 tissues, compared to traditional univariate approaches (**Figure 3a**, **Supplemental Figure S5**). Using these multivariate models, we improve prediction for 79-82% of isoforms with a median increase of ~1.8-2.4 fold increase in adjusted R^2 (**Supplemental Figure S6**, **Supplemental Table S2**). Concordant with simulations, we found that the multivariate elastic net overwhelmingly outperformed other multivariate (and univariate) methods, indicating that leveraging the shared genetic architecture between isoforms of the same gene greatly aids in prediction of each individual isoform (**Supplemental Figure S7**, **Supplemental Table S2**). Notably, we observed that multivariate models were particularly important for brain tissues compared with non-brain tissues in GTEx, which showed significantly improved performance compared with univariate models (**Figure 3b**; $P = 0.011$ from OLS regression of median percent increase in CV R^2 for multivariate/univariate models against tissue type, adjusted for sample size), suggesting more shared isoQTL architecture in brain tissues than others which can be leveraged by isoTWAS for improved prediction. These gains in prediction accuracy directly translate into increased power in the trait association step⁵².”

– The claim about brain-specific effects starting on line 147 is not convincing. The supporting figures Fig 2C and Fig S9 show a clear pattern for the sample size-driven effect. The authors mention a correction in the F-test but there are not enough details.

The reviewer is correct that the statistical test supporting this claim was not detailed enough in the original manuscript. As we have dropped the heritability threshold in the revised manuscript, we have chosen not to include this comparison of heritability of isoforms and genes across tissues in the revision. We did observe that multivariate models improved isoform prediction for brain tissues compared with non-brain tissues, as quantified by the median percent increase in CV R^2 for multivariate versus univariate models. However, as the reviewer notes, brain tissues also tend to have the smallest sample sizes in GTEx, which could explain this result. We conducted an OLS regression of the median percent increase against tissue type (brain and other) and sample size and found a decrease in other tissues with $P = 0.011$, shown in Figure 3b and pasted below. We write in the **revised Results section**:

Line 196: “Notably, we observed that multivariate models were particularly important for brain tissues compared with non-brain tissues in GTEx, which showed significantly improved performance compared with univariate models (**Figure 3b**; $P = 0.011$ from OLS regression of median percent increase in CV R^2 for multivariate/univariate models against tissue type, adjusted for sample size), suggesting more shared isoQTL architecture in brain tissues than others which can be leveraged by isoTWAS for improved prediction.”

- This is minor. In figure S17-S19 it is strange that the isoform level model outperforms the gene level model in cases where the eQTL is shared across isoforms. I assume this is because the noise added to individual isoforms in the simulation is independent. Unless I'm getting this wrong, this issue limits the scope of the simulation results to comparison among different multivariate methods only.

The reviewer is correct in their intuition about the simulation framework. In this revision, we have added complexity to the simulation framework to address this issue. We now add a term for non *cis*-genetic effects on isoform expression that is shared between isoforms and between samples. With these changes to the simulation parameters, we see the result that the reviewer expected in Figure 2 below: the difference in prediction drops as this sharing parameter increases. We also note here that we examined this

relationship empirically in our evaluation of expression prediction, as well, finding a similar result.

In the **revised Methods section**, we now write, and expand with more details in the Supplemental Methods:

Line 611: “We adopt techniques from Mancuso et al’s *twas_sim* protocol to simulate multivariate isoform expression based on randomly simulated genotypes and environmental random noise. First, for n samples, we generate a matrix of genotype dosages for the SNPs within 1 Megabase of 22 different genes (1 per chromosome) using an LD reference panel from European samples from 1000 Genomes Project.

Next, we generate a matrix of SNP-isoform effects across different causal SNP proportions p_c , numbers of isoforms t , and p_s proportion of the SNP-isoform effects being shared across isoforms of the same gene. We then add two matrices of random noise U and ϵ . The first noise matrix U represents non *cis*-genetic effects on isoforms that are correlated between samples and isoforms; we control the proportion of variance explained in isoform expression attributed to U using a parameter σ_h . The second matrix ϵ is a matrix of random noise that is independent for each isoform, such that $\epsilon_i \sim N(0, \sigma_e^2 I)$ where $\sigma_e^2 = 1 - \sigma_h - h_g^2$. We generate 10,000 simulations for each configuration of the simulation parameters, varying $n \in \{200, 500\}$, $p_c \in \{0.001, 0.01, 0.05\}$, $h_g^2 \in \{0.05, 0.10, 0.25\}$, $p_s \in \{0, 0.5, 1\}$, and $\sigma_h \in \{0.1, 0.25\}$. Full mathematical details are provided in **Supplemental Methods** and summarized in **Figure 2**.”

We write in the **revised Results section** briefly:

Line 232: “Finally, as the proportion of non-zero effect SNPs in the isoTWAS model that are shared across isoforms increased (**Supplemental Note**), we found an increasing trend in the gain in prediction of gene expression using isoTWAS compared to TWAS models (**Supplemental Figure S17**), reflecting a similar observation from simulation. Interestingly, as this proportion increased, we found an increase in the gain in prediction of isoform expression, reinforcing the utility of multivariate modelling in marginal prediction of isoform expression.

We expand further in the newly added Supplemental Note and paste Supplemental Figure S17 below:

“*Proportion of shared isoTWAS model effect SNPs.* For each isoform’s predictive model, we determined which SNPs have large effects. First, we standardized the effect sizes in the model to mean 0 and unit variance. We found the SNPs whose effect sizes deviated significantly (Benjamini-Hochberg adjusted $P < 0.05$) and called them the isoform’s effect SNPs. For each gene, we then computed the proportion of effect SNPs that were shared across all isoforms of the same gene. **Supplemental Figure S17** shows a clear increasing trend in the isoform prediction ratios, indicating that multivariate modelling can leverage shared isoform QTL architecture to improve marginal prediction of each

isoform’s expression. The inclusion criterion ratio remains relatively even as this proportion increases. The expression prediction ratio shows a decreasing trend as the proportion of shared isoTWAS effect SNPs increases, reflecting results from simulation (Figure 2).

– Considering that brain-specific genes have a tendency to be intolerant of LoF variants, the statement on line 277 "suggesting that these genes are particularly intolerant to protein-truncating variation and that isoTwas detects biological-relevant signal due to its isoform-specific focus." should be backed up by an appropriate test.

The reviewer is correct in pointing out that we need to add a statistic to justify our claim that isoTwas uncovers more genes that are intolerant of loss of function. Here, we use a Fisher's exact test to assess whether isoTwas identifies a larger number of high pLI genes ($pLI > 0.90$, as defined by gnomAD), compared to TWAS, despite a different number of genes that are tested with each method. This statistical test shows that isoTwas-prioritized genes are more enriched for high pLI genes compared to TWAS-prioritized genes, even accounting for differences in the number of total genes discovered in trait associations with each method.

In the *revised Results section*, we write:

Line 305: "Finally, to explore whether these isoTwas-specific associations were capturing true disease signal, we compared the rate at which each method prioritized constrained genes (probability of loss-of-function intolerance, $pLI \geq 0.9$; **Supplemental Tables S5-S8**), which are known to be substantially enriched for disease associations⁷⁵. Across adult and developmental panels for 15 traits, respectively, isoTwas prioritized 724 and 385 constrained genes compared with 106 and 200 with TWAS, a significant increase (adult: $P=0.048$, developmental: $P = 1.23 \times 10^{-5}$, Fisher's Exact test). Altogether, these results emphasize that isoTwas not only recovers the vast majority of TWAS associations but also greatly increases discovery of candidate GWAS mechanisms, particularly for genes intolerant to protein-truncating variation⁷⁶."

Decision Letter, first revision:

19th Aug 2023

Dear Arjun,

Thank you for submitting your revised manuscript "Isoform-level transcriptome-wide association uncovers extensive novel genetic risk mechanisms for neuropsychiatric disorders in the human brain" (NG-A60952R). It has now been seen by the original referees and their comments are below. The reviewers find that the paper has improved in revision, and therefore we'll be happy in principle to publish it in Nature Genetics, pending minor revisions to satisfy the referees' final requests and to comply with our editorial and formatting guidelines.

Sincerely,

Michael Fletcher, PhD
Senior Editor, Nature Genetics

ORCID: 0000-0003-1589-7087

Reviewer #1 (Remarks to the Author):

I am impressed by the amount of detailed work and the thoughtfulness with which the authors have addressed each of my concerns. The new analyses have further strengthened an already significant project. I would be happy to see the study published in Nature Genetics. The authors have, in my view, generated some new insights into optimally leveraging the transcriptome and laid the basis for some future methods development.

Reviewer #3 (Remarks to the Author):

I want to thank the authors for their review response—no further comments.

Author Rebuttal, first revision:

Decision Letter, second revision:

Final Decision Letter:

5th Oct 2023

Dear Dr. Bhattacharya,

I am delighted to say that your manuscript "Isoform-level transcriptome-wide association uncovers genetic risk mechanisms for neuropsychiatric disorders in the human brain" has been accepted for publication in an upcoming issue of Nature Genetics.

Your paper will be published online after we receive your corrections and will appear in print in the next available issue. You can find out your date of online publication by contacting the Nature Press Office (press@nature.com) after sending your e-proof corrections. Now is the time to inform your Public Relations or Press Office about your paper, as they might be interested in promoting its publication. This will allow them time to prepare an accurate and satisfactory press release. Include your manuscript tracking number (NG-A60952R1) and the name of the journal, which they will need when they contact our Press Office.

Please note that Nature Genetics is a Transformative Journal (TJ). Authors may publish their research with us through the traditional subscription access route or make their paper immediately

open access through payment of an article-processing charge (APC). Authors will not be required to make a final decision about access to their article until it has been accepted. [Find out more about Transformative Journals](https://www.springernature.com/gp/open-research/transformative-journals)

Authors may need to take specific actions to achieve [compliance with funder and institutional open access mandates](https://www.springernature.com/gp/open-research/funding/policy-compliance-faqs). If your research is supported by a funder that requires immediate open access (e.g. according to [Plan S principles](https://www.springernature.com/gp/open-research/plan-s-compliance)) then you should select the gold OA route, and we will direct you to the compliant route where possible. For authors selecting the subscription publication route, the journal's standard licensing terms will need to be accepted, including [those licensing terms will supersede any other terms that the author or any third party may assert apply to any version of the manuscript](https://www.nature.com/nature-portfolio/editorial-policies/self-archiving-and-license-to-publish).

If you have not already done so, we invite you to upload the step-by-step protocols used in this manuscript to the Protocols Exchange, part of our on-line web resource, natureprotocols.com. If you complete the upload by the time you receive your manuscript proofs, we can insert links in your article that lead directly to the protocol details. Your protocol will be made freely available upon publication of your paper. By participating in natureprotocols.com, you are enabling researchers to more readily reproduce or adapt the methodology you use. [Natureprotocols.com](http://natureprotocols.com) is fully searchable, providing your protocols and paper with increased utility and visibility. Please submit your protocol to <https://protocolexchange.researchsquare.com/>. After entering your nature.com username and

password you will need to enter your manuscript number (NG-A60952R1). Further information can be found at <https://www.nature.com/nature-portfolio/editorial-policies/reporting-standards#protocols>

Sincerely,
Chiara

Chiara Anania, PhD
Associate Editor
Nature Genetics
<https://orcid.org/0000-0003-1549-4157>